# Occam's Razor for SSL: Memory-Efficient Parametric Instance Discrimination

**Eric Gan**[*]                                                                    *egan8@ucla.edu*
*Computer Science Department, University of California Los Angeles*

**Patrik Reizinger**[*]                                        *patrik.reizinger@tuebingen.mpg.de*
*Max Planck Institute for Intelligent Systems, Tübingen AI Center, ELLIS Institute, Tübingen, Germany*

**Alice Bizeul**[*]                                                      *alice.bizeul@inf.ethz.ch*
*Department of Computer Science & ETH AI Center, ETH Zürich*

**Attila Juhos**                                                  *attila.juhos@tuebingen.mpg.de*
*Max Planck Institute for Intelligent Systems, Tübingen AI Center, ELLIS Institute, Tübingen, Germany*

**Mark Ibrahim**                                                      *marksibrahim@meta.com*
*FAIR, Meta*

**David Klindt**                                                            *klindt@cshl.edu*
*Cold Spring Harbor Laboratory*

**Randall Balestriero**                                                  *rbalestr@brown.edu*
*Computer Science Department, Brown University*

**Wieland Brendel**                                            *wieland.brendel@tuebingen.mpg.de*
*Max Planck Institute for Intelligent Systems, Tübingen AI Center, ELLIS Institute, Tübingen, Germany*

**Baharan Mirzasoleiman**                                                *baharan@cs.ucla.edu*
*Computer Science Department, University of California Los Angeles*

**Reviewed on OpenReview:** *https://openreview.net/forum?id=GFNTbsVFlP*

## Abstract

Self-supervised learning (SSL) is the prevalent paradigm for representation learning often relying on pairwise similarity between multiple augmented views of each example. Numerous learning methods with various complexities such as gradient stopping, negative sampling, projectors, additional regularization terms, were introduced in the past years. These methods can be effective, but they require careful hyperparameter tuning, have increased computational and memory requirements and struggle with latent dimensionality collapse. Furthermore, complexities such as gradient stopping make them hard to analyse theoretically and confound the essential components of SSL. We introduce a simple parametric instance discrimination method, called Datum IndEx as its Target (DIET). DIET has a single computational branch, without explicit negative sampling, gradient stopping or other hyperparameters. We empirically demonstrate that DIET (1) can be implemented in a memory-efficient way; (2) achieves competitive performance with state-of-the-art SSL methods on small-scale datasets; and (3) is robust to hyperparameters such as batch size. We uncover tight connections to Spectral Contrastive Learning in the lazy training regime, leading to practical insights about the role of feature normalization. Compared to SimCLR or VICReg, DIET also has higher-rank embeddings on CIFAR100 and TinyImageNet, suggesting that DIET captures more latent information.

---

[*]Equal contribution.

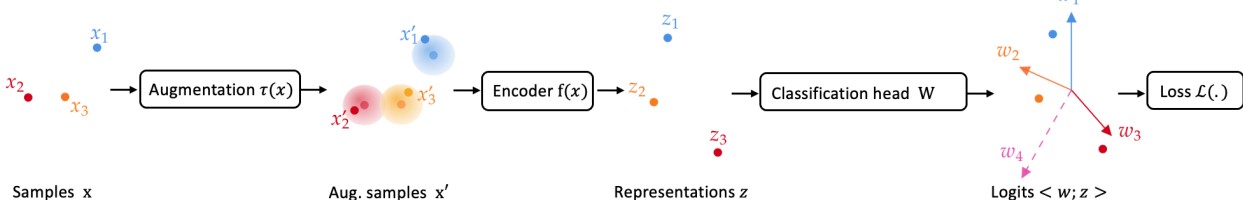

Figure 1: **Overview of the Datum IndEx as its Target (DIET) method**: The typical pipeline for SSL involves selecting a data augmentation strategy, a model architecture, and defining a loss function along with its corresponding hyperparameters. State-of-the-art SSL methods often involve complex design choices across all three aspects. In contrast, DIET simplifies this process: the DIET pipeline has only one computational branch, does not require explicit negative sampling, regularization or elaborate techniques such as stop gradients or parameter averaging.

## 1 Introduction

Self-supervised representation learning (SSL) has become a powerful method for training neural networks without relying on labeled data (Chen et al., 2020; Misra & Maaten, 2020). What makes self-supervised learning (SSL) possible is solving an auxiliary unsupervised task, enabling to pretrain models on large unlabeled datasets. The different principles include reconstruction-based methods such as MAEs (He et al., 2022a), as well as contrastive (Chen et al., 2020; HaoChen et al., 2021b; Radford et al., 2021; Khosla et al., 2020) and non-contrastive methods (Bardes et al., 2021; Chen & He, 2021; Caron et al., 2021a; Oquab et al., 2024; Zbontar et al., 2021). Contrastive Learning (CL) relies on mapping similar samples (called positive pairs) close to each other in latent space, while embedding dissimilar samples (called negative samples) far from each other. Non-contrastive methods do not have negative pairs, they avoid a collapsed representation via regularization terms. This principle led to many successful applications of SSL in the tasks of semantic classification (Chen et al., 2020), image segmentation (Caron et al., 2021a), and monocular depth estimation (Fu et al., 2018), as well as across diverse data domains, ranging from medical imaging (Eslami et al., 2021) to remote sensing (Tao et al., 2020).

The increasing interest in SSL has led to the emergence of a plethora of methods, each introducing its own variation of the core principles. SSL relies on asymmetric computational branches, predictor and projector networks, stop gradients, and many other techniques. Though these might address specific challenges, the field faces many problems. Problems include dimensionality collapse (Jing et al., 2021; von Kügelgen et al., 2021), when some latent factor is not captured, high compute and memory requirements due to large batch sizes and storing augmented samples (Chen et al., 2020), not to mention the need to carefully tune various hyperparameters. This additional complexity not only made it harder to navigate the SSL landscape in practice, it also confounds the truly essential components of SSL. Furthermore, highly complex methods, even though they might improve performance, are less amenable to theoretical analysis. We aim to build up the simplest SSL pipeline to uncover the essential components of self-supervised representation learning. We propose DIET, a parametric instance discrimination (PID) method that solves a classification task based on the sample index, i.e., it learns to distinguish each pair of samples. DIET has a single computational branch, does not require explicit negative sampling, is robust to hyperparameters, and has advantageous learning dynamics with higher-rank embeddings compared to SimCLR (Chen et al., 2020) or VICReg (Bardes et al., 2021). The inherent limitation of instance discrimination is that the classifier head linearly grows with the dataset size, limiting its scalability. Our insight from the DIET loss is that we can accurately approximate the gradients without keeping the whole classifier in memory, significantly improving efficiency. Additionally, we exploit the structure of stateful optimizers such as Adam (Kingma & Ba, 2014) for further gains—we call this version Scaled DIET (s-DIET). We provide theoretical insights on a parameterized feature model, pinpointing the positive effect of feature normalization and demonstrate the feasibility of (scaled) DIET. Empirically, DIET offers a simple yet state-of-the-art alternative to existing SSL baselines on small-scale datasets, providing practical value for practitioners working with specialized, limited data. The modifications introduced in s-DIET enable DIET to scale more effectively to larger settings such as CIFAR-100 and TinyImageNet, while preserving its advantages on small-scale datasets. Our **contributions** are:

- We propose Datum IndEx as its Target (DIET) as a simple parametric instance discrimination (PID) method for self-supervised representation learning (§ 3);
- We provide theoretical insights into the behavior of DIET in comparison to existing SSL methods and pinpoint the advantages of feature normalization (§ 4);
- We show how instance discrimination can be scaled up in form of Scaled DIET (s-DIET) (§ 5);
- We provide extensive empirical evidence that DIET is competitive on downstream classification with SOTA on small datasets, and it has a higher-rank embedding (§ 6).

## 2 Background: Why Self Supervised Learning Needs Occam's Razor

**The SSL model zoo.** SimCLR (Chen et al., 2020) measures the distance between latent representations with cosine similarity, which is scaled by a tunable temperature parameter. DINO (Caron et al., 2021b) utilizes a student-teacher Vision Transformer (ViT) architecture (Dosovitskiy et al., 2020) to minimize the cross-entropy between the student and teacher probability distributions across $K$ classes. SWaV (Caron et al., 2020) incorporates clustering to assign labels to representations, ensuring consistent cluster assignments between data points and their transformed counterparts. MAE (He et al., 2021) uses masking as data augmentation, encouraging the model to learn representations by reconstructing the masked-out information. CLIP (Radford et al., 2021) relies on caption-image pairs as a self-supervised signal. Although the training pipeline for SSL is consistent overall (Morningstar et al., 2024, cf. Fig. 1), approaches differ regarding data augmentations, model architecture, and the loss function.

**SSL is over-specialized.** SSL development was mostly driven by industry, thus, focused on large-scale natural images and sounds (Radford et al., 2021; Oquab et al., 2024; Siméoni et al., 2025). This led to a point where methods are architecture- and dataset-specific (He et al., 2022b; Assran et al., 2023; Oquab et al., 2024). This overspecialization imposes a high barrier of adaptation:

 (i) **Uninformative loss** w.r.t. the DNN's quality (Reed et al., 2021; Garrido et al., 2022): as the last few layers (the projector) are discarded after training, the loss is not necessarily indicative of performance;
 (ii) **Too many hyperparameters:** for loss, projector, and augmentations, with hard-to-predict effect on performance (Grill et al., 2020a; Tian et al., 2021; He & Ozay, 2022);
 (iii) **Lack of hyperparameter transferability** across datasets and architectures (Zhai et al., 2019; Cosentino et al., 2022);
 (iv) **Heavy code refactoring,** compared to supervised models, e.g., for generating positive pairs, handling asymmetric computational branches, and parameter moving averages (Grill et al., 2020b; Caron et al., 2021b).

This makes SSL implementation more costly than supervised learning, often requiring distributed training and long training schedules that reduce the accessibility and inclusivity of SSL research (Crowell, 2023).

## 3 Datum IndEx as its Target (DIET): a simple SSL method

**Motivation: A Simple SSL Method.** Recent SSL methods rely on various design choices and techniques to structure learned representations while avoiding representation collapse, including regularization terms, specialized architectures, and specific data transformations, leading to a multitude of sensitive hyperparameters that require careful tuning. This is a barrier to practical adaptation and theoretical study. In contrast, instance discrimination formulates SSL as cross-entropy maximization over instance labels, i.e., "learning by distinguishing individual data points within a dataset." As we will show, this provides a simpler SSL pipeline, has incentives to capture more information, and is more amenable to theoretical study.

**Intuitition.** SSL often requires large batch size to provide accurate entropy estimates in high dimensions to maximize the uniformity loss (Wang & Isola, 2020; Zhai et al., 2023). The batchwise perspective is not only limited by GPU memory, but also by how it changes the underlying problem. Contrastive objectives such as InfoNCE/SimCLR can be thought of as solving an underlying classification problem, discriminating the positive pair from all negative samples, given the anchor sample. This formulation only constrains relationships between data samples in the batch. This makes it possible that (negative) samples not in a batch have very similar representations to the anchor or positive sample—unless the batch size equals the dataset size. On the other hand, instance discrimination requires distinguishing each pair of data samples, eliminating the above failure case. Intuitively, this incentivizes capturing more information, as, e.g., two

images of similar dogs need to be distinguished, which might be possible by picking up subtle variations such as in fur color.

## 3.1 The DIET method

Instance discrimination focuses on distinguishing individual samples within a dataset by treating each sample as its own class. Alexey et al. (2015) introduced parametric instance discrimination (PID), where they constructed surrogate classes via an elaborate gradient-norm–based strategy and designed class prototypes as trainable parameters, whereas Wu et al. (2018) introduced a non-parametric alternative, where prototypes are selected from a memory bank of previously observed samples. We adopt PID with a nonlinear backbone (encoder) $f_{\boldsymbol{\theta}}$ with parameters $\boldsymbol{\theta}$ and a linear projection $\boldsymbol{W}_H$. We use the sample (datum) index as the classification label and call our method Datum IndEx as its Target (DIET). We optimize the cross-entropy between the probability distribution of a sample's predicted and ground truth indices:

$$\mathcal{L}_{\mathrm{DIET}}(\boldsymbol{x}_n) = \mathrm{XEnt}(\boldsymbol{W}_H f_{\boldsymbol{\theta}}(\boldsymbol{x}_n), y_n = n), \tag{1}$$

where $\boldsymbol{x}_n \in \mathbb{R}^D$ denotes inputs, $y_n$ the corresponding label, which equals, $n$, i.e., the dataset index. The encoder $f_{\boldsymbol{\theta}}$ maps inputs to latents $\boldsymbol{z}_n = f_{\boldsymbol{\theta}}(\boldsymbol{x}_n)$. The learnable projection matrix $\boldsymbol{W}_H$ then maps $\boldsymbol{z}_n$ to logits corresponding to the dataset indices.

**The simplicity of DIET** boils down to (cf. Appx. C.1 for pseudocode):
   (i) **No *explicit* negative sampling:** DIET relies on $\boldsymbol{W}_H$ to ensure distinct representations for each sample instead of negative sampling.
  (ii) **One computational branch:** as the index contains the information that augmented views of the same sample belong together, there is no need for two computational branches;
 (iii) **No specialized solutions:** no stop gradients, asymmetric computational graphs, exponential moving averages and hyperparameters make DIET simple.

The astute reader might notice one limiting factor in DIET: $\boldsymbol{W}_H \in \mathbb{R}^{N \times d}$ grows proportionally to the dataset size $N$. However, this limitation can be overcome, as we show in § 5.

## 4 Theoretical analysis

Our main argument for DIET in § 3 was its simple pipeline. However, these simplifications also make the connection to other popular SSL methods non-obvious. Thus, we present a theoretical analysis to show that under some assumptions, the instance discrimination loss of DIET has the same minimizer as the popular InfoNCE method (Chen et al., 2020; Zimmermann et al., 2021a). Our formulation also suggests that if we replace the cross entropy loss function in (1) with a Mean Squared Error (MSE) loss, then this MSE-DIET loss has the same minimizers as the Spectral Contrastive Learning (SCL) loss (HaoChen et al., 2021b). *These results indicate that for the theoretical guarantees of these methods, not all bells and whistles are necessary, as the much simpler DIET algorithm can learn the same representations.* Last, our formulation also enables us to better understand theoretically why feature normalization can yield better representations (§ 4.2).

## 4.1 A Framework Connecting Pairwise SSL Losses and Instance Discrimination

Given a model $f : \mathbb{R}^d \to \mathbb{R}^m$ and dataset $\mathcal{D}$, many SSL losses in the literature are defined based on the pairwise similarity between embeddings $\boldsymbol{z}_j = f(\boldsymbol{x}_j)$ of samples from the dataset (Chen et al., 2020; Chen & He, 2021; Grill et al., 2020a)—often instantiated as the cosine similarity, which for unit-norm vectors is equivalent to the inner product $\boldsymbol{z}_1^\top \boldsymbol{z}_2$. We call such losses *pairwise similarity losses* and denote them by $\mathcal{L}_{ps}(\mathcal{D}; f) = l_{ps}(\{\boldsymbol{z}_1^\top \boldsymbol{z}_2\})$. This is in contrast to *instancewise losses* $\mathcal{L}_{in}(\mathcal{D}, \mathcal{Y}; f) = \frac{1}{|\mathcal{D}|} \sum_{(\boldsymbol{x}, y)} l_{in}(f(\boldsymbol{x}), y)$ defined as the average of a loss applied on each sample against labels $\mathcal{Y}$, which are more common in the broader machine learning literature.

While pairwise losses avoid an explicit dependence on labels, their specialized construction has made SSL difficult to analyze theoretically, and prevents the direct application of tools from the broader literature aimed at addressing instancewise losses. We bridge this gap by developing a connection between models trained with pairwise loss and models with a single additional linear projection trained with an instancewise loss and labels $\mathcal{Y}$ that are simply the datum index. Relying on the invariance of the inner product in the SCL loss up to orthogonal transformations, empirical and theoretical results about the linearity of feature spaces learned

by neural networks (Roeder et al., 2020; Reizinger et al., 2024; Park et al., 2023), and the invariance of linear probe performance to invertible linear transformations (HaoChen et al., 2021a), we assume that the inner products are preserved by the projector, i.e. that the projector is column-orthogonal.

**Definition 1.** *Let $\mathcal{H}$ be a hypothesis class and $\mathcal{D} = \{\boldsymbol{x}_i\}$ a dataset. For pairwise loss function $\mathcal{L}_{ps}$, define the optimization program*

$$\min_{f \in \mathcal{H}} \mathcal{L}_{ps}(\mathcal{D}; f) \tag{2}$$

*We call instancewise loss $\mathcal{L}_{in}$ with instance labels $\mathcal{Y} = \{y_i\}_{i \in \mathcal{D}}$ an* instancewise equivalent *of $\mathcal{L}_{ps}$ if, for model $\boldsymbol{W}_H f$ constructed by appending linear layer $\boldsymbol{W}_H \in \mathbb{R}^{m \times p}$ to the base model $f$, the optimization program*

$$\min_{f \in \mathcal{H}, \boldsymbol{W}_H \in \mathbb{R}^{m \times p}} \mathcal{L}_{in}(\mathcal{D}, \mathcal{Y}; \boldsymbol{W}_H f) \tag{3}$$

*satisfies the following*
- *If $(f, \boldsymbol{W}_H)$ is a minimizer of 3 and $\boldsymbol{W}_H$ is column-orthogonal, then $f$ is a minimizer of 2.*
- *If $f$ is a minimizer of 2, then there exists $\boldsymbol{W}_H$ such that $\boldsymbol{W}_H$ is column-orthogonal and $(f, \boldsymbol{W}_H)$ is a minimizer of 3.*

The motivation behind the above definition is to reframe a pairwise loss as an instancewise loss by simply adding a linear projector, potentially opening new avenues of theoretical and empirical analysis. Our main theoretical result is to show that some commonly studied pairwise SSL losses have natural instancewise equivalents.

**InfoNCE Loss.** The InfoNCE objective is the most well-studied pairwise loss and the basis of the popular SimCLR method (Chen et al., 2020; Zimmermann et al., 2021a). Since SimCLR uses unit-normalized representations, which have been shown to provide better performance, we consider the hypothesis class of functions that produce representations on the unit hypersphere $\mathcal{H} = \{f : \mathbb{R}^d \to \mathbb{S}^{|\mathcal{D}|-1}\}$. Under this setting, we find that the instancewise equivalent of the InfoNCE loss is the DIET loss from § 3!

**Theorem 1.** *For the hypothesis class of unit-normalized embedding functions, DIET is an instancewise equivalent of the InfoNCE loss.*

**Spectral Contrastive Learning** One of the most well-understood pairwise losses in theoretical analysis is the spectral contrastive loss (SCL) (HaoChen et al., 2021a), where $\delta$ is the Kronecker delta. [1]

$$\mathcal{L}_{\text{SCL}} = -\mathop{\mathbb{E}}_{(\boldsymbol{x}_1, y_1), (\boldsymbol{x}_2, y_2) \sim \mathcal{D}} \left[ \delta_{y_1, y_2} f(\boldsymbol{x}_1)^\top f(\boldsymbol{x}_2) \right] + \frac{1}{2} \mathop{\mathbb{E}}_{(\boldsymbol{x}_1, y_1), (\boldsymbol{x}_2, y_2) \sim \mathcal{D}} \left[ (f(\boldsymbol{x}_1)^\top f(\boldsymbol{x}_2))^2 \right]. \tag{4}$$

Just as SCL can be viewed as a simplification of the InfoNCE objective by dropping the softmax objective, we call also define an MSE-DIET loss with one-hot encoded labels.

$$\mathcal{L}_{\text{DIET}}^{\text{MSE}} = \frac{1}{2} \mathbb{E}_{\boldsymbol{x}_i \sim \mathcal{D}} \left[ \|\boldsymbol{W}_H f(\boldsymbol{x}_i) - \boldsymbol{e}_i\|^2 \right]. \tag{5}$$

We assume $f$ is a parametric feature model $f(\boldsymbol{x}) = \boldsymbol{W}\phi(\boldsymbol{x})$ constructed by composing a fixed, potentially high-dimensional and non-linear feature map $\phi$ with a learnable linear operator $\boldsymbol{W} \in \mathbb{R}^{m \times N}$ with $m \leq N$. This setting captures the lazy training or neural tangent kernel (NTK) regime of neural networks which is common in theoretical analysis and accurate for neural networks in the infinite width limit (Jacot et al., 2018). In this setting, we again find a simple instancewise equivalent.

**Theorem 2.** *For the hypothesis class of parametric feature models, MSE-DIET is an instancewise equivalent of the SCL loss.*

The proofs are presented in Appx. A.2. These results indicate that, at least in a simplified setting, simple instancewise losses can accomplish the same as the more complex pairwise losses. With DIET and MSE-DIET being instancewise equivalents to the well known SSL losses, this justifies the exploration of parametric instance discrimination as an alternative approach to SSL. From a wider perspective, this creates a rigorous connection between the theory on SSL losses, which has been largely independent and self-contained, and the broader machine learning literature, which focuses primarily on instancewise losses. We provide empirical evidence of the given equivalences in § 6 with additional validation in Appx. D.7.

---

[1] Equation (4) differs from some previous definitions by a few constant factors. This does not affect any of the analysis, cf. Appx. A.1.2

Table 1: **DIET trained on small datasets achieves similar accuracy to Imagenet pre-trained SSL for numerous small-scale datasets.** Benchmarks are taken from †:Yang et al. (2022), +:Ericsson et al. (2021)

| Arch. | Pretrain | Method | *Aircraft* | *DTD* | *Pets* | *Flower* | *CUB-200* | *Food101* | *Cars* |
|---|---|---|---|---|---|---|---|---|---|
| *Resnet18* | IN100† | SimCLR | 24.19 | 54.35 | 46.46 | 75.00 | 16.73 | - | - |
| | - | DIET | 37.29 | 50.62 | 64.06 | 72.01 | 33.03 | 62.00 | 42.55 |
| *Resnet50* | | SimCLR | 44.90 | 74.20 | 83.33 | 90.87 | 42.74 | 67.47 | 43.73 |
| | | SimCLRv2 | 46.38 | 76.38 | 84.72 | 92.90 | 52.78 | 73.08 | 50.37 |
| | | MoCov2 | 41.79 | 73.88 | 84.00 | 90.07 | 43.84 | 71.63 | 39.87 |
| | | BYOL | 53.87 | 76.91 | 89.10 | 94.50 | 52.14 | 73.01 | 56.40 |
| | IN-1k+ | VICReg | 53.41 | 76.12 | 89.45 | 93.72 | 62.37 | 75.59 | 61.51 |
| | | SimSiam | 5.97 | 53.03 | 62.17 | 57.93 | 15.34 | 35.45 | 0.85 |
| | | DeepClusterv2 | 54.49 | 78.62 | 89.36 | 94.72 | 59.06 | 77.94 | 58.60 |
| | | Swav | 54.04 | 77.02 | 87.60 | 94.62 | 54.14 | 76.62 | 54.06 |
| | - | DIET | 44.81 | 51.75 | 67.08 | 73.32 | 41.03 | 71.58 | 55.82 |
| *SwinTiny* | - | DIET | 33.15 | 51.88 | 58.06 | 70.78 | 32.11 | 8.86 | 47.12 |
| *Convnext-S* | - | DIET | 43.13 | 9.52 | 61.72 | 67.72 | 31.44 | 69.84 | 40.63 |

## 4.2 Learning More Features via Normalization

Normalizing features to the hypersphere is common SSL (Zimmermann et al., 2021b). We seek to understand how normalization affects the features learned by DIET. For this, we analyze feature learning with content and style latents (von Kügelgen et al., 2021) in a variant of sparse coding that is common in feature learning (Wen & Li, 2021; Zou et al., 2021; Chen et al., 2023; Xue et al., 2023)—cf. Appx. A.1.4 for details.

We model the interaction of content and style concepts in a simple additive model. Let $\mathcal{C} = \{1, \ldots, C\}$ label a set of latent concepts. To each $c \in \mathcal{C}$ we assign a *content* (low noise) feature $\boldsymbol{u}_c$ and a *style* (high noise) feature $\boldsymbol{v}_c$. For a cat, a content feature could be ear shape (almost always a pointed one), while a style feature could be fur color (often highly varying between breeds), with $\boldsymbol{v}_c, \boldsymbol{u}_c \in \mathbb{R}^d$. We assume all $\boldsymbol{u}_i$ and $\boldsymbol{v}_i$ are orthonormal, the feature noises $\epsilon_{\boldsymbol{u}}, \epsilon_{\boldsymbol{v}}$ are drawn from symmetric, zero-mean distributions $p(\epsilon_{\boldsymbol{u}}), p(\epsilon_{\boldsymbol{v}})$ with variances $\sigma_{\boldsymbol{u}}^2 < \sigma_{\boldsymbol{v}}^2$ and a bounded support such that for $\nu_{\boldsymbol{u}}, \nu_{\boldsymbol{v}} < 1$, $|\boldsymbol{u}| \leq \nu_{\boldsymbol{u}}$ and $|\boldsymbol{v}| \leq \nu_{\boldsymbol{v}}$, while the background noise $\boldsymbol{\xi}$ is drawn from a Gaussian distribution scaled by some parameter $\varphi$.

$$\phi(\boldsymbol{x}) = (1 + \epsilon_{\boldsymbol{u}})\boldsymbol{u}_c + (1 + \epsilon_{\boldsymbol{v}})\boldsymbol{v}_c + \boldsymbol{\xi};,$$

where $c \in \mathcal{C}$. We define data augmentation $A$ to replace the noise components $\epsilon_{\boldsymbol{u}}, \epsilon_{\boldsymbol{v}}, \boldsymbol{\xi}$ with a different realization from the same distribution.

By studying the standard and normalized MSE DIET losses, we can prove that only normalized DIET captures both features (proof is in Appx. A.3):

**Theorem 3.** *If $\boldsymbol{W}$ minimizes $\mathcal{L}_{\mathrm{DIET}}^{\mathrm{MSE}}$ and $\boldsymbol{W}_N$ minimizes $\mathcal{L}_{\mathrm{DIET-NORM}}^{\mathrm{MSE}}$ then*

$$\frac{\|\boldsymbol{W}\boldsymbol{v}_c\|}{\|\boldsymbol{W}\boldsymbol{u}_c\|} = \frac{\sigma_{\boldsymbol{u}}^2}{\sigma_{\boldsymbol{v}}^2} + o(1); \qquad \frac{1 - \nu_u}{1 + \nu_v} \leq \frac{\|\boldsymbol{W}_N\boldsymbol{v}_c\|}{\|\boldsymbol{W}_N\boldsymbol{u}_c\|} \leq \frac{1 + \nu_u}{1 - \nu_v}$$

Thm. 3 shows that the smaller variance (content) feature $\boldsymbol{u}$ implies a small alignment between the weight matrix $\boldsymbol{W}$ and the style feature $\boldsymbol{v}_c$ at the optimum of $\mathcal{L}_{\mathrm{DIET}}^{\mathrm{MSE}}$. Thus, DIET may fail to learn style features if a content feature is present—in line with a similar result for contrastive learning from von Kügelgen et al. (2021). Thm. 3 characterizes the alignment quantitatively, supplementing the qualitative non-identifiability result of von Kügelgen et al. (2021). Informally, in the unnormalized model, the larger noise from the style feature introduces a larger loss, so to minimize the loss the model ends up focusing primarily on the lower noise content feature. In contrast, normalization introduces an additional dependency between the directions, which has the effect of balancing the learning between the directions. We show that normalized DIET learns both features approximately equally so long as the noise does not significantly corrupt the features. For example, if the noise ratio is bounded by $\nu_u, \nu_v \leq \frac{1}{2}$, then the alignment with the style feature and the content feature will differ by at most a factor of 3. We validate these findings in § 7.

Table 2: **DIET achieves higher validation accuracy on medical datasets than SSL with standard hyper-parameters.** The supervised model is pretrained on ImageNet1k and a linear probe is trained on top of fixed representations.

| Architecture | Pretraining | Method | *DermaMNIST* | *BloodMNIST* | *PathMNIST* |
|---|---|---|---|---|---|
| *Resnet18* | - | SimCLR | 66.88 | 14.56 | 11.80 |
| | - | MoCov2 | 66.88 | 53.70 | 18.97 |
| | - | BYOL | 65.89 | 80.56 | **65.68** |
| | - | VICReg | 66.78 | 47.18 | 11.31 |
| | - | SimSiam | 66.88 | 43.23 | 17.17 |
| | - | DIET | **73.92** | **89.24** | 44.53 |
| | - | s-DIET | **76.71** | **98.16** | **84.78** |
| | IN-1k$^+$ | Supervised | **74.06** | **88.13** | **59.37** |

# 5 Making DIET Memory Efficient

## 5.1 Batch Cross Entropy

DIET's classifier head $\boldsymbol{W}_H$ scales with the number of examples, limiting scalability due to memory requirements. For our insight, we consider the gradients w.r.t. $\boldsymbol{w}_k$ (the derivation is in Appx. A.4)

$$\nabla_{\boldsymbol{w}_k}\mathcal{L}_{\text{DIET}} = -\frac{1}{B}\sum_{i=1}^{B}(y_{i,k} - p(k|\boldsymbol{z}_i))\boldsymbol{z}_i, \tag{6}$$

where $y_{i.k}$ is the $i^{th}$ component of the true, one-hot label of sample $k$, and $p(k|\boldsymbol{z}_i)$ is the predicted class probability distribution, given the embedding of sample $i$ and only the logits corresponding to samples in the batch appear. Thus, for a batch $\boldsymbol{X}_{\mathcal{I}}$ with $B$ elements and with indices $\mathcal{I} \subseteq [N]$, let $\boldsymbol{W}_H[\mathcal{I}] \in \mathbb{R}^{B \times d}$ collect the $i^{\text{th}}$ row of $\boldsymbol{W}_H$, for all $i \in \mathcal{I}$. We hypothesize that this subsampling provides a reasonable approximation of the gradients of all parameters $\theta$:

$$\nabla_\theta \text{XEnt}_N(\boldsymbol{W}_H f_\theta(\boldsymbol{X}_{\mathcal{I}}), \mathcal{I}) \approx \nabla_\theta \text{XEnt}_B(\boldsymbol{W}_H[\mathcal{I}]f_\theta(\boldsymbol{X}_{\mathcal{I}})). \tag{7}$$

Instead of calculating standard cross entropy on the $N$-dimensional outputs of $\boldsymbol{W}_H$, we select its $B$ rows corresponding to the indices in the batch, reassign samples a distinct label from $\{0, \ldots, B-1\}$ and calculate a $B$-dimensional *batch cross entropy*. This can be interpreted as batchwise instance discrimination. For empirical validation, cf. § 7.3, for an illustration, Fig. 2.

**Memory savings: an illustration on ImageNet.** Batch cross entropy requires only that $B$ rows of $\boldsymbol{W}_H$ are in memory, which in the standard $\dim x \gg \dim z$ case provides a small memory cost compared to loading the data. For example, 256 ImageNet (Deng et al., 2009) images with $224 \times 224$ resolution require 150 MB per batch, as opposed to only 2 MB for the $2048-$dimensional classifier.

## 5.2 Multi batch crossentropy

Although our experiments were built around the batch cross entropy loss, there is a generalization, where we decouple the batch size from the number of rows of $\boldsymbol{W}_H$ loaded. Instead of only loading the rows of $\boldsymbol{W}_H$ that correspond to index set $\mathcal{I}$, we would actually load a number of heads corresponding to $m \cdot B$ samples from an index set $\mathcal{I}_m$, where $m$ is chosen arbitrarily large. In this case, we would make sure that the upcoming $m$ batches, each with $B$ elements, cover exactly, without overlap or hiatus, the training datapoints from $\mathcal{I}_m$.

Figure 2: **Batch cross entropy:** The index set $\mathcal{I}$ collects all sample indices that are in the current batch. We then use $\mathcal{I}$ to select which rows of $\boldsymbol{W}_H$ to load into memory, decreasing the memory footprint

# 6 Experiments

**Setup.** We perform experiments on a toy, a synthetic, and 4 real-world datasets: CIFAR-10, CIFAR-100 (Krizhevsky et al., 2009), ImageNet-100 (Tian et al., 2020), and TinyImageNet (Le & Yang, 2015). For our models, we study the ResNet family of architectures, specifically ResNet-18 and ResNet-50 (He et al., 2016), and vision transformers (ViT) (Dosovitskiy et al., 2020), specifically ViT-B/16. ResNet-18, ResNet-50, and Vit-B/16 have embedding dimensions 512, 2048, and 768, respectively. We use a three-layer ReLU MLP as a projection head during the training of s-DIET. We fix the default label smoothing to 0.8 and the data augmentation pipeline to a combination of cropping, flipping, color jitter and gaussian blurring. Details on data augmentation are presented in Alg. 3. We use an AdamW optimizer with a $10^{-3}$ learning rate and 0.05 weight decay with a cosine learning rate decay. We fix the batch size to 256 for all experiments and train DIET and s-DIET for 5000 epochs. Our baselines were trained until convergence using the same data augmentation as for DIET. All baseline hyper-parameters were kept to

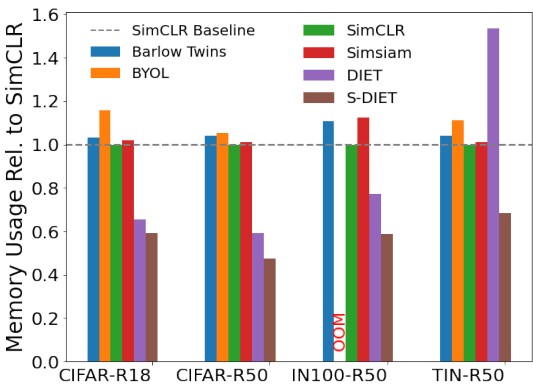

Figure 3: **GPU Memory Comparison** relative to SimCLR with a batch size of 256. OOM indicates out-of-memory on an Nvidia A40 GPU. DIET requires 2x, SimCLR up to 2.2x more memory than s-DIET. Absolute values are deferred to Tab. 15

the default values proposed by the original works. After training, we evaluate our representations by training a linear classifier on top of frozen representations to perform semantic classification on the validation set.

## 6.1 An edge on small datasets

**Transfer learning.** We investigate how a trained-from-scratch DIET performs on small datasets that are commonly handled by SSL through transfer learning: Aircraft (Maji et al., 2013), DTD (Cimpoi et al., 2014), Pets (Parkhi et al., 2012), Flowers (Nilsback & Zisserman, 2008), CUB200 (Wah et al., 2011), Food101 (Bossard et al., 2014), Cars (Krause et al., 2013). These datasets have much fewer samples than Imagenet but their image distribution is often much less diverse, e.g., focusing only on aircraft. The current SOTA is to pretrain one's favorite SSL method on a larger dataset such as Imagenet100 or Imagenet-1k and to fine-tune on the target dataset. But pretraining over large, uncurated datasets can introduce risks such as data poisoning or bias amplification, which are critical to avoid in high-stakes scenarios (Zhang et al., 2024). Perhaps surprisingly, **DIET provides an alternative without pre-training i.e., by training directly on the small dataset**—this can be leveraged in scenarios where tight control over the data is required, e.g., to avoid data poisoning. Tab. 1 shows that DIET matches or surpasses SimCLR pre-trained on ImageNet-1K across three of the evaluated transfer datasets. When compared to SimCLR trained on IN100, DIET consistently outperforms it—often substantially. These findings suggest that DIET can serve as a simpler yet competitive alternative to more complex and less interpretable methods in small-scale settings.

**DIET is SOTA beyond Natural Images.** Medical datasets generally have very few samples as such data is notoriously hard to collect. Furthermore, pre-training on ImageNet is less sensible as the data distributions differ significantly. Thus we compare SSL methods (DIET, SimCLR, MoCov2, VICReg) trained from scratch on three datasets from the MedMNISTv2 medical imaging benchmark (Yang et al., 2023) (i) PathMNIST ($90,000-7,180$ train/test split); (ii) DermaMNIST ($10,015-2,005$ split); and (iii) BloodMNIST ($17,092-3,421$ split). For SimCLR, MoCov2, VICReg, we use the default hyperparameters from Susmelj et al. (2020) which yield good performance ($>80\%$) on CIFAR10, a comparably small dataset of $60,000$ images. All algorithms achieve high training accuracy via a linear probe, but the baseline SSL methods do not generalize well to the test sets (Tab. 2). By contrast, DIET achieves much higher performance. For an ablation for DIET with ViT, see Appx. C.8, and training curves in Fig. 16, showing that DIET's hyperparameters transfer. DIET also has a speed advantage: for ResNet18, DIET is 1.75x faster than SimCLR (and 1.72x faster than VICReg). These findings provide strong practical guidance for SSL practitioners: **On small-scale in-the-wild datasets—often characterized by distribution shifts from standard image benchmarks—DIET serves as a simple yet effective alternative for achieving state-of-the-art performance.** BYOL is a strong baseline, and on the largest of the medical examples we consider, PathMNIST, it even outperforms supervised learning.

Table 3: **s-DIET achieves higher accuracy than existing CL methods by up to 2.72%.** Linear probe accuracy of s-DIET against DIET and various SSL baselines on CIFAR-10, CIFAR-100, ImageNet-100, and TinyImageNet. s-DIET obtains state-of-the-art with limited GPU memory.

| Method | CIFAR-10 | | CIFAR-100 | | ImageNet-100 | TinyImageNet |
| | ResNet-18 | ResNet-50 | ResNet-18 | ResNet-50 | ResNet-50 | ResNet-50 |
| --- | --- | --- | --- | --- | --- | --- |
| SimCLR | 90.00 | 91.64 | 63.56 | 67.90 | 79.68 | 46.32 |
| MoCov2 | 81.30 | 82.53 | 63.75 | 68.10 | 70.90 | 38.81 |
| BYOL | 90.76 | 92.32 | 65.26 | 68.10 | (OOM) | 40.72 |
| VICReg | 91.15 | 92.67 | 66.76 | 70.11 | - | 43.05 |
| Simsiam | 90.78 | 92.42 | 65.66 | 69.62 | 80.12 | 40.48 |
| DIET | 54.64 | 89.70 | 62.93 | 68.96 | 73.50 | 51.66 |
| s-DIET | **91.48** | **93.08** | **66.88** | **72.34** | **80.16** | **52.52** |

## 6.2 Scaling DIET to Large-Scale Natural Datasets

While DIET achieves near state-of-the-art performance on smaller datasets like CIFAR-10/-100 (50,000 samples), its original formulation begins to show limitations when scaled to more challenging datasets such as ImageNet-100 and TinyImageNet (Tab. 3). On ImageNet-100, DIET struggles to match SOTA, while on TinyImageNet DIET is no longer memory efficient due to the larger number of samples which directly impact the size of $W_H$ (Fig. 3). In these scenarios, we use batch cross entropy (§ 5) to improve the memory efficiency of DIET, while adding representation normalization to improve the feature learning ability of DIET (§ 4.2).

A three-layer MLP projection head is added during training and removed at evaluation, following prior findings that this improves the learning efficacy of self-supervised methods (Bordes et al., 2022; Xue et al., 2024). The full S-DIET algorithm is summarized in Apdx. D.1. This scaled version of DIET (s-DIET) achieves a balance betweeen the simplicity of DIET and practical performance: s-DIET is more than 2x more memory efficient than DIET on TinyImageNet and up to 2.2x more memory efficient than other SSL methods, while outperforming SSL baselines by up to 2.72% points. The compromise is that s-DIET is slower to converge (§ 6.2). Improving the convergence rate of s-DIET is an interesting direction for future research. For an ablation with a ViT backbone, cf. Tab. 21. In Tab. 3, we find that s-DIET matches and outperforms popular SSL methods for ImageNet-100

Table 4: **Training time** of s-DIET versus SimCLR on CIFAR 10/100, in hours, on a single NVIDIA A5000 GPU. Although more memory efficient, s-DIET compromises on training time.

| Method | Model | Training Time |
| --- | --- | --- |
| s-DIET | ResNet-18 | 16.2 |
| SimCLR | ResNet-18 | 3.9 |
| s-DIET | ResNet-50 | 51.0 |
| SimCLR | ResNet-50 | 11.5 |

and TinyImageNet respectively. For even larger datasets such as ImageNet-1k, DIET is completely infeasible due to the size of $W_H$, so s-DIET must be used to avoid memory inefficiencies. However, we find that the slow convergence rate of s-DIET is a limiting factor, especially in resource limited scenarios: s-DIET achieves 52.01% linear probe accuracy after 500 epochs, whereas SOTA SSL methods can achieve around 65% accuracy using the same number of epochs.

## 7 Ablations

In the previous sections, we evaluate DIET and its extended variant, s-DIET, on standard semantic classification benchmarks. We now shift focus to a more general *unsupervised* evaluation of the learned representations.

### 7.1 Improved Feature Learning

**Toy Setting: clean and noisy features.** We use the setup presented in § 4.2 with 4 latent classes and noise parameters $\sigma_u = 0.01, \sigma_v = 0.1$ (for details, cf. Appx. D.4) and compare the alignment between the weights and the clean and noisy feature of the first class. As illustrated in (Fig. 4), normalization enables the model to learn both features, whereas, without it, only the clean feature is learned.

Table 5: **The effect of normalization** on downstream classification accuracy in CIFAR-100.

| Normalization | Accuracy |
| --- | --- |
| Yes (s-DIET) | **66.88** |
| No (DIET) | 62.60 |

**Normalization Increases Embedding Rank.** Interestingly, we show that the normalization applied in s-DIET further improves the singular value spectrum of DIET embeddings, resulting in representations with

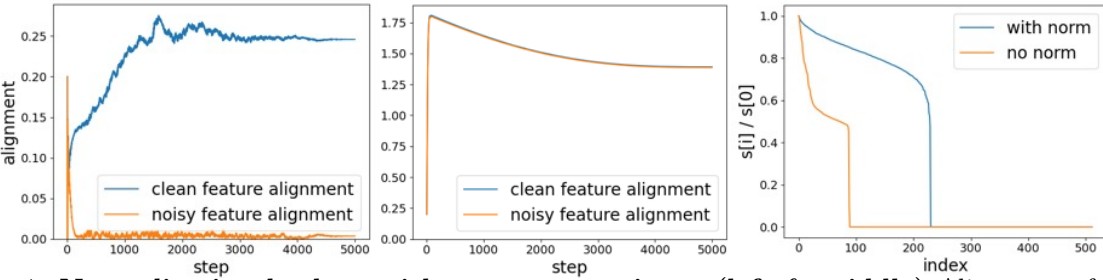

Figure 4: **Normalization leads to richer representations. (left & middle)** Alignment of weight matrix $\boldsymbol{W}$ with clean feature $\boldsymbol{u}_1$ and noisy feature $\boldsymbol{v}_1$ (calculated as $\|\boldsymbol{W}\boldsymbol{u}_1\|$ and $\|\boldsymbol{W}\boldsymbol{v}_1\|$, respectively) when using DIET (left) and normalized DIET (*right*). DIET only learns the clean feature but normalized DIET learns both features almost equally. **(right)** Singular values of representations for CIFAR-100 are sorted in decreasing order. Values are normalized by the largest singular value.

even higher rank (Tab. 7). We also confirm that these enhancements translate into richer representations, in line with prior findings (Garrido et al., 2022; Thilak et al., 2024), and they also translate into improved classification accuracy on CIFAR-100 (Tab. 5).

**Combined MNIST and CIFAR-10.** We construct a synthetic dataset modelling the data generation process from § 4.2 where each input example consists of a CIFAR-10 image and a MNIST image of the same label index concatenated along the channel dimension—akin to the design of Shah et al. (2020); Chen et al. (2021). We use weaker augmentations on the MNIST image, making the MNIST image the content and the CIFAR-10 image the style feature (for details, cf. Appx. D.5).

We train a ResNet-18 using DIET with and without normalization. During linear probe evaluation, we may mask the MINST digit to compare how well the models learned the CIFAR-10 image. We observe in the inset table that DIET quickly overfits the MNIST digit, even when it is masked, indicating that the CIFAR-10 features are not well learned. Normalization maintains high performance regardless of whether the MNIST digit is present, showing that the CIFAR-10 features are learned.

Table 6: **The effect of masking** on downstream classification accuracy on the combined MNIST and CIFAR-10 dataset.

| Normalization | No Masking | Masking |
|---|---|---|
| Yes (s-DIET) | 83.9 | **84.06** |
| No (DIET) | 13.76 | **43.56** |

Higher-rank inputs can improve downstream linear classifier performance (Cover, 1965), which inspired recent works to propose the rank-based RankMe (Garrido et al., 2022) and LiDAR (Thilak et al., 2024) metrics to evaluate the embedding rank in SSL. (Garrido et al., 2022; Thilak et al., 2024) find that these metrics strongly correlate with downstream accuracy.

**DIET learns high-rank embeddings.** As real-world datasets do not grant access to ground-truth features, we also adopt a proxy-based evaluation using rank-based metrics and track singular values of the learned representations, as in (Xue et al., 2022). In Tab. 7, we compare the RankMe (Garrido et al., 2022) and LiDAR (Thilak et al., 2024) scores on CIFAR-100 and TinyImageNet. We find that, despite its simplicity, DIET improves dimensional collapse, a challenge typically addressed with bells and whistles, resulting in capturing a richer set of features. In addition, we show that the singular values of DIET's embeddings converge faster to a narrow range, whereas SSL baselines converge slower and span a wider range, showing DIET's clear advantage (c.f., Fig. 11).

Table 7: **DIET produces high-rank embeddings:** DIET achieves substantially higher RankMe and LiDAR scores using ResNet18 architectures.

| Dataset | Method | RankMe (↑) | LiDAR (↑) |
|---|---|---|---|
| CIFAR100 | DIET | **499.58** | **479.21** |
| | SimCLR | 355.05 | 326.58 |
| | VICReg | 422.28 | 377.37 |
| | MoCov2 | 313.49 | 309.17 |
| | SimSiam | 441.52 | 308.88 |
| | BYOL | 261.42 | 222.38 |
| TinyImageNet | DIET | 318.38 | **414.88** |
| | SimCLR | 365.87 | 343.13 |
| | VICReg | **408.02** | 391.60 |
| | MoCov2 | 335.73 | 394.33 |
| | SimSiam | 419.42 | 294.21 |
| | BYOL | 238.04 | 208.03 |

## 7.2 Experimental Validation of Thm. 5

We train a model using SCL and MSE-DIET on the toy dataset described in § 4.2 and Appx. D.4. We then convert the MSE-DIET model into an equivalent model where the head is column-orthogonal as described in Lemma 1, and then we compare the procrustes distance between the embeddings produced by the SCL and equivalent DIET model. Figure 5 shows that the procrustes distance vanishes, indicating that the models learn the same embeddings up to an orthogonal transformation. This finding is especially interesting in light of our loss ablation in Appx. C and Tab. 8, where the MSE and cross entropy formulations of the DIET objective show substantially different classification performance. This finding is in line with the known trade-off between the richness of the representation and downstream performance (Rusak et al., 2024).

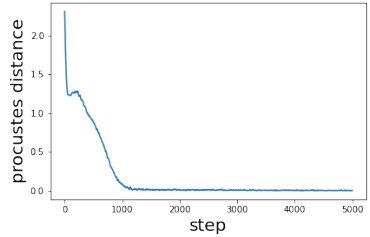

Figure 5: **Procrustes distance** between embeddings learned by MSE-DIET and SCL vanishes.

## 7.3 Sensitivity analysis

Finally, we explore the sensitivity of DIET to its remaining hyper-parameters.

**Loss Function.** We compare the performance of using mean-squared versus cross entropy loss on CIFAR-100 (Tab. 8). Although our experimental validation of Thm. 5 shows that the embeddings learned by the MSE-DIET and SCL objectives have a vanishing procrustes distance, this similarity does not necessarily transfer to similar downstream performance. Namely, we find that the cross-entropy loss provides better performance in practice. This is consistent with standard practice in supervised learning.

Table 8: **The effect of the loss** on downstream classification accuracy.

| Model | | MSE | CE |
|---|---|---|---|
| CIFAR100 | RN18 | 58.21 | **66.88** |
| | RN50 | 64.44 | **72.34** |

**Batch size.** We investigate the effect of batch size on TinyImagenet and report the accuracy in Tab. 9. Remarkably, the performance of DIET is stable across a broad range of batch sizes, and even as low as 16 causes a relative performance drop of only 5%. Similar conclusions are drawn for s-DIET (Fig. 8).

**Batch Cross Entropy.** To confirm that batch cross entropy closely approximates standard cross entropy, we calculate the cosine similarity of base model gradients (i.e., excluding the projection head or classifier head) of randomly initialized models on CIFAR-100 for the base model. Tab. 10 shows that the cosine similarity between the gradients of batch cross entropy and full cross entropy is nearly 1 across the board, with higher cosine similarity for larger models and batch sizes.

Table 9: **The effect of batch size** on downstream classification accuracy on TinyImagenet (3000 training epochs).

| batch size | 16 | 32 | 64 | 128 | 256 | 512 |
|---|---|---|---|---|---|---|
| RN18 | 37.9 | 42.7 | 43.4 | 43.3 | 43.7 | 43.7 |

**Label smoothing.** Finally, we investigate the effect of label smoothing (LS) on downstream performance and observe that applying LS with values between 0.4 and 0.8 significantly accelerates the convergence rate of DIET. As a result, label smoothing also enhances the performance of DIET as much as $\sim$5% points when training for a fixed number of epochs, as shown in Fig. 14.

**Projector network.** A projector network is often used in SSL to improve performance (Chen et al., 2020; He et al., 2022b). We observe that a 3-layer ReLU MLP as a projector also improves linear probe accuracy for s-DIET on CIFAR-100 (Tab. 11).

**Data Augmentations.** Previous works have emphasized the importance of data augmentation (DA) for the success of SSL (Balestriero et al., 2023; Morningstar et al., 2024; Ciernik et al., 2024). Thus, we consider three DA regimes: *Low* only includes random crops and horizontal flips; *Intermediate* further adds color jittering and grayscaling;

Table 10: **Cosine similarity** of gradients for CE and batch CE with ResNet models on CIFAR-100. Batch CE approximates CE.

| Batch Size | RN18 | RN50 |
|---|---|---|
| 64 | 0.9944 | 0.9960 |
| 128 | 0.9965 | 0.9980 |
| 256 | 0.9975 | 0.9990 |
| 512 | 0.9980 | 0.9995 |

Table 11: **The effect of the projector** network on linear probe accuracy on CIFAR-100.

| Model | Pre-project. | Post-project. |
|---|---|---|
| RN18 | **66.88** | 63.46 |
| RN50 | **72.34** | 67.60 |

and *High* further adds Gaussian blur and random erasing (Zhong et al., 2020)—for the exact setup, cf. Alg. 3. Tab. 12 shows that on TinyImagenet (for ablations, cf., Fig. 15 and Tab. 16) DIET greatly benefits from intermediate DA, however, the high regime does not have a large further improvement.

## 8  Discussion

Our work focuses on understanding the simplest set of components that make SSL work, for which we introduce Datum IndEx as its Target (DIET), a parametric instance discrimination (PID) method. DIET has only one computational branch and requires no explicit negative sampling or other specialized techniques such as stop gradients. In a simplified linear model, we provide theoretical insights about how feature normalization can help recover more features in the presence of content (lower variance) and style (high variance) features and investigate connections to the well known InfoNCE and SCL objectives from contrastive learning. To improve memory efficiency, we introduced a batched cross entropy strategy based on analyzing the gradients of DIET, providing a scalable version of the algorithm.

Table 12: **The effect of data augmentation** strength on downstream classification accuracy in CIFAR-100. Refer to the text for details

| DA | Low | Inter. | High |
|---|---|---|---|
| RN18 | 31.48 | 43.62 | 43.88 |
| RN50 | 40.24 | 48.80 | 50.81 |
| RN101 | 40.07 | 49.74 | 50.76 |

Through extensive evaluation, we show that DIET offers state-of-the-art results over other SSL methods on small-scale datasets. We also demonstrate DIET's memory efficiency on ImageNet-100 and TinyImageNet and find that DIET learns higher-rank embeddings, corroborating our insights about the role of feature normalization. As SSL continues to be adopted across a wider range of tasks and domains, DIET offers a simple yet effective approach for real-world applications, requiring minimal tuning while learning rich representations in small-scale settings.

### Acknowledgments

The authors thank the International Max Planck Research School for Intelligent Systems (IMPRS-IS) for supporting Patrik Reizinger and Attila Juhos. Patrik Reizinger acknowledges his membership in the European Laboratory for Learning and Intelligent Systems (ELLIS) PhD program. This work was supported by the German Federal Ministry of Education and Research (BMBF): Tübingen AI Center, FKZ: 01IS18039A. Wieland Brendel acknowledges financial support via an Emmy Noether Grant funded by the German Research Foundation (DFG) under grant no. BR 6382/1-1 and via the Open Philantropy Foundation funded by the Good Ventures Foundation. Wieland Brendel is a member of the Machine Learning Cluster of Excellence, EXC number 2064/1 – Project number 390727645. This research utilized compute resources at the Tübingen Machine Learning Cloud, DFG FKZ INST 37/1057-1 FUGG. Alice Bizeul's work is supported by an ETH AI Center Doctoral fellowship. Baharan Mirzasoleiman was partially supported by the National Science Foundation CAREER Award 2146492, the NSF-Simons AI Institute for Cosmic Origins (CosmicAI), Institute for Foundations of Machine Learning (IFML), and an Okawa Research Award.

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

# Supplementary Materials

The supplementary materials is providing the proofs of the main's paper formal results. We also provide as much background results and references as possible throughout to ensure that all the derivations are self-contained. Some of the below derivation do not belong to formal statements but are included to help the curious readers get additional insights into current SSL methods.

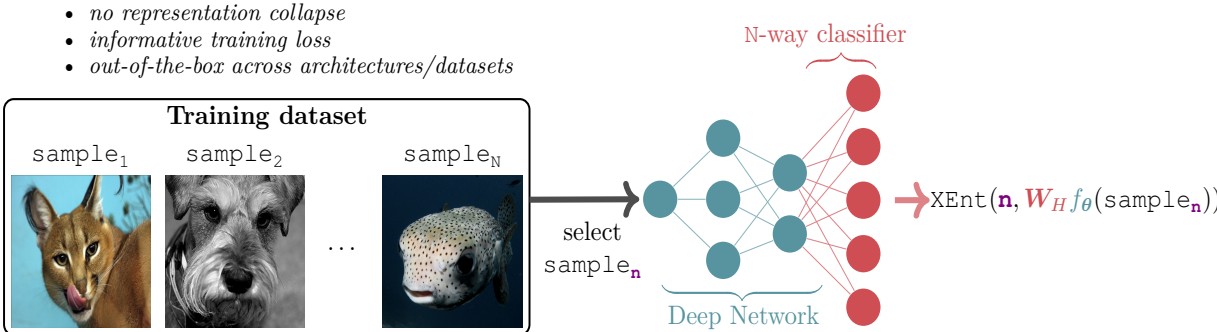

Figure 6: **DIET** uses the datum index (n) as the class-target –effectively turning unsupervised learning into a supervised learning problem. In our case, we employ the cross-entropy loss (X-Ent), no extra care needed to handle different dataset or architectures. As opposed to current SOTA, we do not rely on a projector nor positive views *i.e* no change needs to be done to any existing supervised pipeline to obtain DIET. As highlighted in Fig. 7, DIET's training loss is even informative of downstream test performances, and as ablated in Appx. C there is no degradation of performance with longer training, even for very small datasets (Tab. 1).

## A    Theoretical Analysis and Proofs

### A.1    Technical Setup

#### A.1.1    Notation and Setup

We use regular font for scalars, bold lowercase font for vectors, bold uppercase font for matrices.

We use $\|\cdot\|$ to represent the Euclidean norm for vectors and $\|\cdot\|_F$ to represent the Frobenius norm for matrices. The vector $\boldsymbol{e}_i$ represents the $i$-th standard basis vector. For a matrix $\boldsymbol{M}$, we write $\boldsymbol{M}^\dagger$ for the Moore-Penrose pseudoinverse of $\boldsymbol{M}$.

We say a matrix $\boldsymbol{M} \in \mathbb{R}^{m \times n}$ is an isometry if $\boldsymbol{M}^\top \boldsymbol{M} = \boldsymbol{I}_m$. Equivalently, $\langle \boldsymbol{M}\boldsymbol{v}_1, \boldsymbol{M}\boldsymbol{v}_2 \rangle = \langle \boldsymbol{v}_1, \boldsymbol{v}_2 \rangle$ for all $\boldsymbol{v}_1, \boldsymbol{v}_2 \in \mathbb{R}^n$. We say $\boldsymbol{M}$ is a partial isometry if $\boldsymbol{M}$ acts as an isometry on the orthogonal complement of its kernel.

For a matrix $\boldsymbol{M} \in \mathbb{R}^{m \times n}$ and a scalar function $g : \mathbb{R}^{m \times n} \to \mathbb{R}$, $\frac{\partial g}{\partial \boldsymbol{M}}$ consists of the partial derivatives of $g$ with respect to the entries of $\boldsymbol{M}$, namely

$$\frac{\partial g}{\partial \boldsymbol{M}} = \begin{bmatrix} \frac{\partial g}{\partial M_{11}} & \cdots & \frac{\partial g}{\partial M_{1n}} \\ \vdots & \ddots & \vdots \\ \frac{\partial g}{\partial M_{m1}} & \cdots & \frac{\partial g}{\partial M_{mn}} \end{bmatrix}$$

We use the Kronecker delta function $\delta_{i,j}$, which is defined as 1 if $i = j$ otherwise 0.

#### A.1.2    Definition of Spectral Contrastive Loss

Recall the given definition of the spectral contrastive loss

$$\mathcal{L}_{scl} = \underset{(\boldsymbol{x}_1,y_1),(\boldsymbol{x}_2,y_2)\sim\mathcal{D}}{\mathbb{E}} \left[ -\delta_{y_1,y_2} f(\boldsymbol{x}_1)^\top f(\boldsymbol{x}_2) \right] + \underset{(\boldsymbol{x}_1,y_1),(\boldsymbol{x}_2,y_2)\sim\mathcal{D}}{\mathbb{E}} \left[ (f(\boldsymbol{x}_1)^\top f(\boldsymbol{x}_2))^2 \right],$$

In Xue et al. (2023), the positive pair term in the contrastive loss was instead defined as

$$\mathbb{E}_{(x,y),(x,y')\sim\mathcal{D},y=y'}[-2f(\boldsymbol{x})^\top f(\boldsymbol{x}')]$$

so that

$$\mathcal{L}_{scl}^* = \mathop{\mathbb{E}}_{(x,y),(x,y')\sim\mathcal{D},y=y'}[-2f(\boldsymbol{x})^\top f(\boldsymbol{x}')] + \mathop{\mathbb{E}}_{(\boldsymbol{x}_1,y_1),(\boldsymbol{x}_2,y_2)\sim\mathcal{D}}\left[(f(\boldsymbol{x}_1)^\top f(\boldsymbol{x}_2))^2\right],$$

This only differs from the current definition by a some constant multiple $\alpha$, where $\alpha$ is the inverse of the probability that a randomly chosen pair is a positive pair. The reason for changing this normalization is that with the original formulation, the norm of the optimal weights and embeddings would grow with the number of classes. Quantitatively, it is not hard to check that

$$\mathcal{L}_{scl}^*(\alpha f) = \alpha^2 \mathcal{L}_{scl}(f)$$

That is, the loss landscape of the two loss functions is the same up to rescaling. It turns out this is the correct scaling factor to keep the norm of the optimal weights and embeddings bounded, with scale matching those produced by DIET.

### A.1.3   Isometric Classifier Head

**Assumption 4.** *The embedding dimension is at most the number of labels. That is, $m \le n$.*

If Assumption 4 is satisfied, then requiring that $\boldsymbol{W}_H$ be an isometry does not restrict the expressivity of the model class since any model can be converted into an equivalent one where $\boldsymbol{W}_H$ is an isometry:

**Lemma 1.** *Suppose Assumption 4 holds and $f$ is a linear model $f_{\boldsymbol{W}}(\boldsymbol{x}) = \boldsymbol{W}\boldsymbol{x}$ and $\boldsymbol{W}_H$ is the projection head. For any model $(\boldsymbol{W}_H, \boldsymbol{W})$, there exists another model $(\boldsymbol{W}_H', \boldsymbol{W}')$ such that the model outputs agree, i.e. $\boldsymbol{W}_H \boldsymbol{W} = \boldsymbol{W}_H' \boldsymbol{W}'$, and $\boldsymbol{W}_H'$ is an isometry.*

*Proof.* Let $\boldsymbol{W}_H = \boldsymbol{U}\boldsymbol{\Sigma}\boldsymbol{V}^\top$ be an SVD of $\boldsymbol{W}_H$, where $\boldsymbol{U} \in \mathbb{R}^{n\times n}, \boldsymbol{\Sigma} \in \mathbb{R}^{n\times m}, \boldsymbol{V} \in \mathbb{R}^{m\times m}$. Since rank $\boldsymbol{W}_H \le m \le n$, this decomposition can be truncated so that

$$\boldsymbol{W}_H = \boldsymbol{U}_1 \boldsymbol{\Sigma}_1 \boldsymbol{V}^\top$$

where $\boldsymbol{U}_1 \in \mathbb{R}^{n\times m}, \boldsymbol{\Sigma}_1 \in \mathbb{R}^{m\times m}$ and $\boldsymbol{U}_1^\top \boldsymbol{U}_1 = \boldsymbol{I}_m$. Then taking $\boldsymbol{W}_H' = \boldsymbol{U}_1$ and $f' = \boldsymbol{\Sigma}_1 \boldsymbol{V}^\top f$ works.   $\square$

### A.1.4   Theoretical Setting for Normalization Theory

In this section, we formally define the theoretical setup used in § 4.2.

Let $C \in \mathbb{Z}^+, \nu_u, \nu_v, \sigma_u, \sigma_v, \phi \in \mathbb{R}^+$ be constants. Let $\mathcal{C} = \{1, \ldots, C\}$ label a set of latent concepts. To each $c \in \mathcal{C}$ we assign a *content* (low noise) feature $\boldsymbol{u}_c$ and a *style* (high noise) feature $\boldsymbol{v}_c$. For a cat, a content feature could be ear shape (almost always a pointed one), while a style feature could be fur color (often highly varying between breeds), with $\boldsymbol{v}_c, \boldsymbol{u}_c \in \mathbb{R}^d$. We assume all $\boldsymbol{u}_i$ and $\boldsymbol{v}_i$ are orthonormal, the feature noises $\epsilon_{\boldsymbol{u}}, \epsilon_{\boldsymbol{v}}$ are drawn from symmetric, zero-mean distributions $p(\epsilon_{\boldsymbol{u}}), p(\epsilon_{\boldsymbol{v}})$ with variances $\sigma_{\boldsymbol{u}}^2 < \sigma_{\boldsymbol{v}}^2$ and a bounded support such that for $\nu_{\boldsymbol{u}}, \nu_{\boldsymbol{v}} < 1$, $|\boldsymbol{u}| \le \nu_{\boldsymbol{u}}$ and $|\boldsymbol{v}| \le \nu_{\boldsymbol{v}}$, while the background noise is drawn from a Gaussian distribution scaled by some parameter $\phi$.

We also make the following technical assumptions:

1. Balanced classes: The number of examples from each latent class are equal.

2. Isometric classifier head: $\boldsymbol{W}_H$ is a fixed isometry. As before, this allows us to study the structure of the embedding space induced by the loss function without worrying about the effect of $\boldsymbol{W}_H$.

3. Alignment: For all $i, h_i = \|\boldsymbol{W}_H^\top \boldsymbol{e}_i\| \ne 0$. If $h_i = 0$, then the model outputs would always be perpendicular to $\boldsymbol{e}_i$, so the normalized DIET loss on $\boldsymbol{x}_i$ would be a constant. Requiring $h_i \ne 0$ ensures that $\boldsymbol{x}_i$ can contribute to the learning.

4. Initialization: We initialize $\boldsymbol{W} = \boldsymbol{0}$, and train using gradient descent on the population loss.

5. Sparse concepts: $|C| = o(d)$.

While some of these theoretical assumptions are idealized, we demonstrate that similar behavior occurs in more general real-world settings in Section 7.

### A.1.5 Normalization of Zero

Note that normalizing the zero vector is not well-defined. This can be an issue in the setup of Theorem 3 because we initialize $\boldsymbol{W} = \boldsymbol{0}$. In PyTorch, this is handled by redefining $norm(\boldsymbol{x}) \leftarrow \frac{\boldsymbol{x}}{\max\{\|\boldsymbol{x}\|, \epsilon\}}$ for negligible $\epsilon$. We will take a similar approach, where we simply define $norm(\boldsymbol{0}) = \boldsymbol{0}$ and the Jacobian as $\boldsymbol{J}_{norm}(\boldsymbol{0}) = \boldsymbol{I}$. This can be seen as taking $\epsilon \to 0$ and rescaling the Jacobian at $\boldsymbol{0}$ so that it does not blow up. Note that in the standard formula for the Jacobian of the normalization function,

$$\boldsymbol{J} = \frac{1}{\|\boldsymbol{x}\|}\left(\boldsymbol{I} - \frac{1}{\|\boldsymbol{x}^2\|}\boldsymbol{x}\boldsymbol{x}^\top\right)$$

the same formula holds when $\boldsymbol{x} = \boldsymbol{0}$ if we drop the $\|x\|$ terms. In the following proofs, this is how we will interpret such formulas in case we need to normalize a zero vector.

### A.2 Proof of Theorems

### A.2.1 Proof of Thm. 1

*Proof.* A result from Lu & Steinerberger (2021) shows that the global minimizer of DIET is the simplex ETF configuration. Awasthi et al. (2022) showed that the global minimizer of the InfoNCE object is also the simplex ETF configuration. Since an orthogonal transformation preserves both norms and simplex ETF structure, the theorem follows. $\square$

### A.2.2 Proof of Thm. 2

The statement of Thm. 2 is equivalent to the following:

**Theorem 5.** *Suppose that Assumption 4 holds and $f$ is a parametric feature model $f(\boldsymbol{x}) = \boldsymbol{W}\phi(\boldsymbol{x})$. Then,*
- *If $(\boldsymbol{W}, \boldsymbol{W}_H)$ is a global minimizer of $\mathcal{L}_{\mathrm{DIET}}^{\mathrm{MSE}}$ and $\boldsymbol{W}_H$ is column-orthogonal, then $\boldsymbol{W}$ is a global minimizer of $\mathcal{L}_{\mathrm{SCL}}$.*
- *If $\boldsymbol{W}$ is a global minimizer of $\mathcal{L}_{\mathrm{SCL}}$, then there exists $\boldsymbol{W}_H$ such that $\boldsymbol{W}_H$ is column-orthogonal and $(\boldsymbol{W}, \boldsymbol{W}_H)$ is a global minimizer of $\mathcal{L}_{\mathrm{DIET}}^{\mathrm{MSE}}$.*

Denote by $N = |\mathcal{D}|$ be the size of the augmented dataset. We represent this dataset in matrix form

$$\mathcal{D} = (\boldsymbol{X}, \boldsymbol{Y}) \in \mathbb{R}^{d \times N} \times \mathbb{R}^{n \times N}$$

where every column of $\boldsymbol{X}$ is the representation of an augmented input in feature space and the corresponding column of $\boldsymbol{Y}$ is a one-hot encoding of the label.

Define the following useful matrices to characterize the structure of the data:

$$\boldsymbol{M} = \mathbb{E}_{(x,y)\sim\mathcal{D}}[\boldsymbol{x}\boldsymbol{x}^\top] = \frac{1}{N}\boldsymbol{X}\boldsymbol{X}^\top$$

$$\boldsymbol{M}_{pos} = \mathbb{E}_{(x_1,y_1),(x_2,y_2)\sim\mathcal{D}}[\boldsymbol{x}_1\boldsymbol{x}_2^\top \delta_{y_1,y_2}] = \frac{1}{N^2}\boldsymbol{X}\boldsymbol{Y}^\top\boldsymbol{Y}\boldsymbol{X}^\top$$

Here $\boldsymbol{M}$ is the expected outer product of all examples with themselves, and $\boldsymbol{M}_{pos}$ is the expected outer product between pairs of examples if they are in the same class (known as positive pairs).

We outline the proof as follows. First we leverage a result from Xue et al. (2023) which characterizes the critical points and global minima of the spectral contrastive loss in the same setting. We then prove a relationship between the critical points of MSE diet and the sepctral contrastive loss. Finally, we prove a relationship between the global minima of the two loss functions.

For the rest of this section, we will just write $\mathcal{L}_{diet}$ in place of $\mathcal{L}_{diet}^{mse}$.

The following is a statement and slightly simplified proof of the key theorem from Xue et al. (2023):

**Theorem 6.** *A linear function $f(\boldsymbol{x}) = \boldsymbol{W}\boldsymbol{x}$ is a critical point of $\mathcal{L}_{scl}$ iff there is a basis such that*

$$\boldsymbol{M}^\dagger\boldsymbol{M}_{pos} = diag(\lambda_1, \dots, \lambda_r, \lambda_{r+1}, \dots, \lambda_d)$$

$$\boldsymbol{W}^\top\boldsymbol{W}\boldsymbol{M} = diag(\lambda_1, \dots, \lambda_r, 0, \dots, 0)$$

$$\boldsymbol{W}^\top\boldsymbol{W}\boldsymbol{M}_{pos} = diag(\lambda_1^2, \dots, \lambda_r^2, 0, \dots, 0)$$

*with $\lambda_1, \ldots, \lambda_d \geq 0$ and we have $r \leq \mathrm{rank}\, \boldsymbol{W} \leq m$.*
*It is a global minimum of $\mathcal{L}_{scl}$ iff it satisfies*

$$\boldsymbol{W}^\top \boldsymbol{W} \boldsymbol{M} = [\boldsymbol{M}^\dagger \boldsymbol{M}_{pos}]_m$$

*Proof.* The first order condition for $\mathcal{L}_{scl}$

$$\frac{\partial \mathcal{L}_{scl}}{\partial \boldsymbol{W}} = -\boldsymbol{W} \boldsymbol{M}_{pos} + \boldsymbol{W} \boldsymbol{M} \boldsymbol{W}^\top \boldsymbol{W} \boldsymbol{M} = 0 \tag{8}$$

Since $\boldsymbol{M}$ and $\boldsymbol{M}_{pos}$ are positive semidefinite, $\boldsymbol{M}^\dagger \boldsymbol{M}_{pos}$ is diagonalizable. Therefore we can construct a basis $\{\boldsymbol{v_1}, \ldots, \boldsymbol{v_d}\}$ of eigenvectors of $\boldsymbol{M}^\dagger \boldsymbol{M}_{pos}$ with corresponding eigenvalues $\lambda_1, \ldots, \lambda_d$.
Now we have $\mathrm{im}\, \boldsymbol{M}_{pos} \subset \mathrm{im}\, \boldsymbol{M}$, which implies that $\boldsymbol{M}_{pos} = \boldsymbol{M} \boldsymbol{M}^\dagger \boldsymbol{M}_{pos}$. Then Equation 8 implies that

$$(\boldsymbol{W}^\top \boldsymbol{W} \boldsymbol{M})^2 \boldsymbol{v}_i = \boldsymbol{W}^\top \boldsymbol{W} \boldsymbol{M} (\boldsymbol{M}^\dagger \boldsymbol{M}_{pos}) \boldsymbol{v}_i = \lambda_i \boldsymbol{W}^\top \boldsymbol{W} \boldsymbol{M} \boldsymbol{v}_i$$

Thus either $\boldsymbol{W}^\top \boldsymbol{W} \boldsymbol{M} \boldsymbol{v}_i = \boldsymbol{0}$ or $\boldsymbol{W}^\top \boldsymbol{W} \boldsymbol{M} \boldsymbol{v}_i$ is an eigenvector of $\boldsymbol{W}^\top \boldsymbol{W} \boldsymbol{M}$ with eigenvalue $\lambda_i$. Since $\boldsymbol{W}^\top \boldsymbol{W} \boldsymbol{M}$ is diagonalizable, the latter implies that $\boldsymbol{v}_i$ is also an eigenvalue of $\boldsymbol{W}^\top \boldsymbol{W} \boldsymbol{M}$ with $\boldsymbol{W}^\top \boldsymbol{W} \boldsymbol{M} \boldsymbol{v}_i = \lambda_i \boldsymbol{v_i} = \boldsymbol{M}^\dagger \boldsymbol{M}_{pos} \boldsymbol{v_i}$
Thus, with possible reordering of the $\boldsymbol{v}_i$, we have a basis $\boldsymbol{v}_1, \ldots, \boldsymbol{v}_r, \ldots, \boldsymbol{v}_d$ such that in this basis

$$\boldsymbol{M}^\dagger \boldsymbol{M}_{pos} = diag(\lambda_1, \ldots, \lambda_r, \lambda_{r+1}, \ldots, \lambda_d)$$
$$\boldsymbol{W}^\top \boldsymbol{W} \boldsymbol{M} = diag(\lambda_1, \ldots, \lambda_r, 0, \ldots, 0)$$
$$\boldsymbol{W}^\top \boldsymbol{W} \boldsymbol{M}_{pos} = diag(\lambda_1^2, \ldots, \lambda_r^2, 0, \ldots, 0)$$

with $\lambda_1, \ldots, \lambda_d \geq 0$ and we have and $r \leq \mathrm{rank}\, \boldsymbol{W} \leq m$.
Note that if $\boldsymbol{W}$ admits the above form, then

$$\boldsymbol{W}^\top \boldsymbol{W} \boldsymbol{M}_{pos} = \boldsymbol{W}^\top \boldsymbol{W} \boldsymbol{M} \boldsymbol{W}^\top \boldsymbol{W} \boldsymbol{M}$$

which implies

$$\boldsymbol{W} \boldsymbol{M}_{pos} = \boldsymbol{W} \boldsymbol{M} \boldsymbol{W}^\top \boldsymbol{W} \boldsymbol{M}$$

hence all such $\boldsymbol{W}$ are critical points.
Then for all such $\boldsymbol{W}$,

$$\mathcal{L} = \mathrm{Tr}[-2\boldsymbol{W}^\top \boldsymbol{W} \boldsymbol{M}_{pos} + \boldsymbol{W}^\top \boldsymbol{W} \boldsymbol{M} \boldsymbol{W}^\top \boldsymbol{W} \boldsymbol{M}]$$
$$= -2\sum_{i=1}^r \lambda_i^2 + \sum_{i=1}^r \lambda_i^2$$
$$= -\sum_{i=1}^r \lambda_i^2$$

It is clear from the above expression that the minimum among critical points is achieved when $r$ is maximal and $\lambda_1, \ldots, \lambda_m$ are the largest eigenvalues. This happens if and only if

$$\boldsymbol{W}^\top \boldsymbol{W} \boldsymbol{M} = [\boldsymbol{M}^\dagger \boldsymbol{M}_{pos}]_m$$

It remains to check the behavior as $\|\boldsymbol{W}\|_F$ grows large. Equivalently, $\boldsymbol{W}^\top \boldsymbol{W}$ has a large eigenvalue $\lambda$. Let $\boldsymbol{w}$ be a corresponding eigenvector. If $\boldsymbol{w} \in \ker \boldsymbol{M}$, then $\boldsymbol{M} \boldsymbol{w} = \boldsymbol{M}_{pos} \boldsymbol{w} = 0$, so we see that the loss is unchanged. Otherwise, $\boldsymbol{w}$ has some nonzero alignment with $\mathrm{im}(\boldsymbol{W})$. But then $\mathrm{Tr}[\boldsymbol{W}^\top \boldsymbol{W} \boldsymbol{M} \boldsymbol{W}^\top \boldsymbol{W} \boldsymbol{M}]$ grows quadratically in $\lambda$, but $\mathrm{Tr}[-2\boldsymbol{W}^\top \boldsymbol{W} \boldsymbol{M}_{pos}]$ grows at most linearly in $\lambda$, hence the loss is large. We conclude that the previously found condition in fact specifies the global minimizers of $\mathcal{L}$. $\qquad\square$

The following lemma establishes a connection between the critical points of $\mathcal{L}_{diet}$ versus $\mathcal{L}_{scl}$.

**Lemma 2.** *The following are true:*

- *If $(\boldsymbol{W}, \boldsymbol{W}_H)$ is a critical point of $\mathcal{L}_{diet}$ and $\boldsymbol{W}_H$ is an isometry, then $\boldsymbol{W}$ is a critical point of $\mathcal{L}_{scl}$.*

- *If $\boldsymbol{W}$ is a critical point of $\mathcal{L}_{scl}$, then there exists a partial isometry $\boldsymbol{W}_H$ such that $(\boldsymbol{W}, \boldsymbol{W}_H)$ is a critical point of $\mathcal{L}_{diet}$.*

*Proof.* The first order condition for $\mathcal{L}_{diet}$ requires that

$$\frac{\partial \mathcal{L}_{diet}}{\partial \boldsymbol{W}} = \boldsymbol{W}_H^\top (\boldsymbol{W}_H \boldsymbol{W} \boldsymbol{X} - \boldsymbol{Y}) \boldsymbol{X}^\top = 0 \tag{9}$$

$$\frac{\partial \mathcal{L}_{diet}}{\partial \boldsymbol{W}_H} = (\boldsymbol{W}_H \boldsymbol{W} \boldsymbol{X} - \boldsymbol{Y}) \boldsymbol{X}^\top \boldsymbol{W}^\top = 0 \tag{10}$$

On the other hand, the first order condition for $\mathcal{L}_{scl}$ is

$$\boldsymbol{W} \boldsymbol{M}_{pos} = \boldsymbol{W} \boldsymbol{M} \boldsymbol{W}^\top \boldsymbol{W} \boldsymbol{M}.$$

Indeed, if $\boldsymbol{W}$ is a critical point of $\mathcal{L}_{diet}$, then Equation 9 implies

$$\boldsymbol{W} \boldsymbol{X} \boldsymbol{X}^\top = \boldsymbol{W}_H^\top \boldsymbol{Y} \boldsymbol{X}^\top \tag{11}$$

And Equation 10 gives

$$\boldsymbol{W}_H \boldsymbol{W} \boldsymbol{X} \boldsymbol{X}^\top \boldsymbol{W}^\top = \boldsymbol{Y} \boldsymbol{X}^\top \boldsymbol{W}^\top$$

Taking transposes, we have

$$\boldsymbol{W} \boldsymbol{X} \boldsymbol{X}^\top \boldsymbol{W}^\top \boldsymbol{W}_H^\top = \boldsymbol{W} \boldsymbol{X} \boldsymbol{Y}^\top \tag{12}$$

Right multiplying by $\boldsymbol{W}_H$ and using the fact that $\boldsymbol{W}_H^\top \boldsymbol{W}_H = \boldsymbol{I}_m$ gives

$$\boldsymbol{W} \boldsymbol{X} \boldsymbol{X}^\top \boldsymbol{W}^\top = \boldsymbol{W} \boldsymbol{X} \boldsymbol{Y}^\top \boldsymbol{W}_H \tag{13}$$

Combining Equations 11 and 13, we get

$$\boldsymbol{W} \boldsymbol{X} \boldsymbol{X}^\top \boldsymbol{W}^\top \boldsymbol{W} \boldsymbol{X} \boldsymbol{X}^\top = \boldsymbol{W} \boldsymbol{X} \boldsymbol{Y}^\top \boldsymbol{W}_H \boldsymbol{W}_H^\top \boldsymbol{Y} \boldsymbol{X}^\top$$

We claim that

$$\boldsymbol{W} \boldsymbol{X} \boldsymbol{Y}^\top \boldsymbol{W}_H \boldsymbol{W}_H^\top = \boldsymbol{W} \boldsymbol{X} \boldsymbol{Y}^\top$$

Indeed, since $\boldsymbol{W}_H$ is an isometry, $\boldsymbol{W}_H^\top$ is a partial isometry, so $\boldsymbol{W}_H \boldsymbol{W}_H^\top$ has a basis $\{\boldsymbol{v}_1, \ldots, \boldsymbol{v}_n\}$ such that $\boldsymbol{W}_H \boldsymbol{W}_H^\top \boldsymbol{v}_i = \boldsymbol{v}_i$ or $\boldsymbol{W}_H \boldsymbol{W}_H^\top \boldsymbol{v}_i = \boldsymbol{0}$. If the former is true, then clearly $\boldsymbol{W} \boldsymbol{X} \boldsymbol{Y}^\top \boldsymbol{W}_H \boldsymbol{W}_H^\top \boldsymbol{v}_i = \boldsymbol{W} \boldsymbol{X} \boldsymbol{Y}^\top \boldsymbol{v}_i$. If the latter is true, then we know that $\boldsymbol{W}_H^\top \boldsymbol{v}_i = \boldsymbol{0}$. But then by Equation 12 we have

$$\boldsymbol{W} \boldsymbol{X} \boldsymbol{Y}^\top \boldsymbol{v}_i = \boldsymbol{W} \boldsymbol{X} \boldsymbol{X}^\top \boldsymbol{W}^\top \boldsymbol{W}_H^\top \boldsymbol{v}_i = \boldsymbol{0}$$

Since equality holds on a basis, we conclude the two matrix products are equal, as claimed.
Thus we now have

$$\boldsymbol{W} \boldsymbol{X} \boldsymbol{X}^\top \boldsymbol{W}^\top \boldsymbol{W} \boldsymbol{X} \boldsymbol{X}^\top = \boldsymbol{W} \boldsymbol{X} \boldsymbol{Y}^\top \boldsymbol{Y} \boldsymbol{X}^\top$$

Substituting the values $\boldsymbol{M} = \boldsymbol{X} \boldsymbol{X}^\top$ and $\boldsymbol{M}_{pos} = \boldsymbol{X} \boldsymbol{Y}^\top \boldsymbol{Y} \boldsymbol{X}$,

$$\boldsymbol{W} \boldsymbol{M}_{pos} = \boldsymbol{W} \boldsymbol{M} \boldsymbol{W}^\top \boldsymbol{W} \boldsymbol{M}$$

as desired.

For the converse, suppose that $\boldsymbol{W}$ is a critical point of $\mathcal{L}_{scl}$, namely

$$\boldsymbol{W}\boldsymbol{M}_{pos} = \boldsymbol{W}\boldsymbol{M}\boldsymbol{W}^\top\boldsymbol{W}\boldsymbol{M}$$

Let $V = \ker(\boldsymbol{M}_{pos} - \boldsymbol{M}\boldsymbol{W}^\top\boldsymbol{W}\boldsymbol{M})$. Since $\boldsymbol{M}_{pos} - \boldsymbol{M}\boldsymbol{W}^\top\boldsymbol{W}\boldsymbol{M}$ is symmetric, $V^\perp$ is spanned by eigenvectors with nonzero eigenvalues. Let $\boldsymbol{v}$ be such an eigenvector with eigenvalue $\lambda \neq 0$. Then

$$\boldsymbol{0} = \boldsymbol{W}(\boldsymbol{M_{pos}} - \boldsymbol{M}\boldsymbol{W}^\top\boldsymbol{W}\boldsymbol{M})\boldsymbol{v} = \lambda\boldsymbol{W}\boldsymbol{v}$$

It follows that $\boldsymbol{W}\boldsymbol{v} = 0$, so $V^\perp \subset \ker\boldsymbol{W}$.
Set $U = (\boldsymbol{W}\boldsymbol{X}\boldsymbol{X}^\top)(V), Z = (\boldsymbol{Y}\boldsymbol{X}^\top)(V)$. Since

$$(\boldsymbol{Y}\boldsymbol{X}^\top)^\top(\boldsymbol{Y}\boldsymbol{X}^\top) = \boldsymbol{M_{pos}} = \boldsymbol{M}\boldsymbol{W}^\top\boldsymbol{W}\boldsymbol{M} = (\boldsymbol{W}\boldsymbol{X}\boldsymbol{X}^\top)^\top(\boldsymbol{W}\boldsymbol{X}\boldsymbol{X}^\top)$$

when restricted to $V$, there exists an isometry $\boldsymbol{W}'_H : U \to Z$ such that $\boldsymbol{Y}\boldsymbol{X}^\top = \boldsymbol{W}_H\boldsymbol{W}\boldsymbol{X}\boldsymbol{X}^\top$ on $V$ and $\boldsymbol{X}\boldsymbol{Y}^\top = \boldsymbol{X}\boldsymbol{X}^\top\boldsymbol{W}^\top\boldsymbol{W}_H^\top$ on $Z$. Extend $\boldsymbol{W}'_H$ to a partial isometry $\boldsymbol{W}_H : \mathbb{R}^m \to \mathbb{R}^n$ such that $\boldsymbol{W}_H|_U = \boldsymbol{W}'_H$ and $\boldsymbol{W}_H|_{U^\perp} = \boldsymbol{0}$.
Now using the fact that $\text{im}(\boldsymbol{W}^\top) = \ker(\boldsymbol{W})^\perp \subset V$, we have

$$\boldsymbol{Y}\boldsymbol{X}^\top\boldsymbol{W}^\top = \boldsymbol{W}_H\boldsymbol{W}\boldsymbol{X}\boldsymbol{X}^\top\boldsymbol{W}^\top$$

Also

$$\boldsymbol{X}\boldsymbol{Y}^\top\boldsymbol{W}_H = \boldsymbol{X}\boldsymbol{X}^\top\boldsymbol{W}^\top\boldsymbol{W}_H^\top\boldsymbol{W}_H$$

because any vector in $\mathbb{R}^m$ can be written as $\boldsymbol{u} + \boldsymbol{u}_\perp$ where $\boldsymbol{u} \in U, \boldsymbol{u}_\perp \in U^\perp$ and

$$\begin{aligned}
\boldsymbol{X}\boldsymbol{Y}^\top\boldsymbol{W}_H(\boldsymbol{u} + \boldsymbol{u}_\perp) &= \boldsymbol{X}\boldsymbol{Y}^\top\boldsymbol{W}_H\boldsymbol{u} \\
&= \boldsymbol{X}\boldsymbol{X}^\top\boldsymbol{W}^\top\boldsymbol{W}_H^\top\boldsymbol{W}_H\boldsymbol{u} \\
&= \boldsymbol{X}\boldsymbol{X}^\top\boldsymbol{W}^\top\boldsymbol{W}_H^\top\boldsymbol{W}_H(\boldsymbol{u} + \boldsymbol{u}_\perp)
\end{aligned}$$

These are the two conditions for being a critical point of $\mathcal{L}_{diet}$, completing the proof.

$\square$

We now narrow our attention from critical points to global minima. The above Lemma means that we can restrict our study to the critical points of $\mathcal{L}_{scl}$. Using this fact, we can now characterize the global minimizers of $\mathcal{L}_{diet}$ as follows:

**Theorem 7.** *Assume that $\boldsymbol{W}_H$ is an isometry. Then $(\boldsymbol{W}, \boldsymbol{W}_H)$ is global minimizer of $\mathcal{L}_{diet}$ iff the following hold*

$$\boldsymbol{W}^\top\boldsymbol{W}\boldsymbol{M} = [\boldsymbol{M}^\dagger\boldsymbol{M}_{pos}]_m$$

$$\frac{1}{N}\text{Tr}(\boldsymbol{W}\boldsymbol{X}\boldsymbol{Y}^\top\boldsymbol{W}_H^\top) = \text{Tr}[[\boldsymbol{M}^\dagger\boldsymbol{M}_{pos}]_m]$$

*Proof.* Suppose $(\boldsymbol{W}, \boldsymbol{W}_H)$ is a global minimizer of $\mathcal{L}_{diet}$ and $\boldsymbol{W}_H$ is an isometry. By Lemma 2, $\boldsymbol{W}$ is a critical point of $\mathcal{L}_{scl}$. By Theorem 6, there is a basis such that

$$\begin{aligned}
\boldsymbol{M}^\dagger\boldsymbol{M}_{pos} &= diag(\lambda_1, \dots, \lambda_r, \lambda_{r+1}, \dots, \lambda_d) \\
\boldsymbol{W}^\top\boldsymbol{W}\boldsymbol{M} &= diag(\lambda_1, \dots, \lambda_r, 0, \dots, 0) \\
\boldsymbol{W}^\top\boldsymbol{W}\boldsymbol{M}_{pos} &= diag(\lambda_1^2, \dots, \lambda_r^2, 0, \dots, 0)
\end{aligned}$$

with $\lambda_1, \dots, \lambda_d \geq 0$ and we have $r \leq \text{rank}\,\boldsymbol{W} \leq m$.

Now calculating the value of the loss

$$
\begin{aligned}
\mathcal{L}_{diet} &= \frac{1}{2}\mathbb{E}_{\mathcal{D}}[\|\boldsymbol{W}_H\boldsymbol{W}\boldsymbol{x}_i - e_{y_i}\|^2] \\
&= \frac{1}{2N}\|\boldsymbol{W}_H\boldsymbol{W}\boldsymbol{X} - \boldsymbol{Y}\|_F^2 \\
&= \frac{1}{2N}\operatorname{Tr}((\boldsymbol{W}_H\boldsymbol{W}\boldsymbol{X} - \boldsymbol{Y})^\top(\boldsymbol{W}_H\boldsymbol{W}\boldsymbol{X} - \boldsymbol{Y})) \\
&= \frac{1}{2N}\operatorname{Tr}(\boldsymbol{X}^\top\boldsymbol{W}^\top\boldsymbol{W}_H^\top\boldsymbol{W}_H\boldsymbol{W}\boldsymbol{X} - \boldsymbol{X}^\top\boldsymbol{W}^\top\boldsymbol{W}_H^\top\boldsymbol{Y} - \boldsymbol{Y}^\top\boldsymbol{W}_H\boldsymbol{W}\boldsymbol{X} + \boldsymbol{Y}^\top\boldsymbol{Y}) \\
&= \frac{1}{2N}\left(\operatorname{Tr}(\boldsymbol{W}^\top\boldsymbol{W}\boldsymbol{X}\boldsymbol{X}^\top) - 2\operatorname{Tr}(\boldsymbol{W}\boldsymbol{X}\boldsymbol{Y}^\top\boldsymbol{W}_H) + \operatorname{Tr}(\boldsymbol{Y}^\top\boldsymbol{Y})\right)
\end{aligned}
$$

Observe that

$$
\frac{1}{N}\operatorname{Tr}(\boldsymbol{W}^\top\boldsymbol{W}\boldsymbol{X}\boldsymbol{X}^\top) = \operatorname{Tr}(\boldsymbol{W}^\top\boldsymbol{W}\boldsymbol{M}) = \sum_{i=1}^r \lambda_i
$$

Also $\boldsymbol{W}^\top\boldsymbol{W}\boldsymbol{M}_{pos} = \frac{1}{N^2}\boldsymbol{W}^\top\boldsymbol{W}\boldsymbol{X}\boldsymbol{Y}^\top\boldsymbol{Y}\boldsymbol{X}^\top$ and $\frac{1}{N^2}\boldsymbol{W}\boldsymbol{X}\boldsymbol{Y}^\top\boldsymbol{Y}\boldsymbol{X}^\top\boldsymbol{W}^\top$ are diagonalizable and have the same nonzero eigenvalues, namely $\lambda_1^2, \ldots, \lambda_r^2$. Using the fact that

$$
\boldsymbol{W}\boldsymbol{X}\boldsymbol{Y}^\top\boldsymbol{W}_H\boldsymbol{W}_H^\top = \boldsymbol{W}\boldsymbol{X}\boldsymbol{Y}^\top
$$

we have

$$
(\frac{1}{N}\boldsymbol{W}\boldsymbol{X}\boldsymbol{Y}^\top\boldsymbol{W}_H)(\frac{1}{N}\boldsymbol{W}\boldsymbol{X}\boldsymbol{Y}^\top\boldsymbol{W}_H)^\top = \frac{1}{N^2}\boldsymbol{W}\boldsymbol{X}\boldsymbol{Y}^\top\boldsymbol{Y}\boldsymbol{X}^\top\boldsymbol{W}^\top,
$$

we conclude by the Spectral Theorem that

$$
\frac{1}{N}\operatorname{Tr}(\boldsymbol{W}\boldsymbol{X}\boldsymbol{Y}^\top\boldsymbol{W}_H) \le \sum_{i=1}^r \lambda_i \tag{14}
$$

Finally, note that $\operatorname{Tr}(\boldsymbol{Y}^\top\boldsymbol{Y})$ is a constant. Therefore the minimum possible value of the loss is when $\boldsymbol{W}^\top\boldsymbol{W}\boldsymbol{M} = [\boldsymbol{M}^\dagger\boldsymbol{M}_{pos}]_m$ and equality holds in equation 14 with $r = m$ and $\lambda_1, \ldots, \lambda_m$ the $m$ largest eigenvalues of $\boldsymbol{M}^\dagger\boldsymbol{M}_{pos}$. It only remains to show this value of the loss is achievable.

Indeed, it is not hard to find $\boldsymbol{W}$ such that $\boldsymbol{W}^\top\boldsymbol{W}\boldsymbol{M} = [\boldsymbol{M}^\dagger\boldsymbol{M}_{pos}]_m$ (for example take a global minimizer of $\mathcal{L}_{scl}$).

Let $\boldsymbol{W}\boldsymbol{X}\boldsymbol{Y}^\top = \boldsymbol{U}\boldsymbol{\Sigma}\boldsymbol{V}^\top$ be a singular value decomposition of $\boldsymbol{W}\boldsymbol{X}\boldsymbol{Y}^\top$. Let $\boldsymbol{W}_H : \mathbb{R}^m \to \mathbb{R}^n$ map the ith eigenvector of $\boldsymbol{U}$ to the ith eigenvector of $\boldsymbol{V}$ for $i = 1, \ldots, p$. Then

$$
\boldsymbol{W}\boldsymbol{X}\boldsymbol{Y}^\top\boldsymbol{W}_H = \boldsymbol{U}\boldsymbol{\Sigma}\boldsymbol{U}^\top.
$$

In particular, $\boldsymbol{W}\boldsymbol{X}\boldsymbol{Y}^\top\boldsymbol{W}_H$ is a positive semidefinite matrix, and

$$
\frac{1}{N^2}(\boldsymbol{W}\boldsymbol{X}\boldsymbol{Y}^\top\boldsymbol{W}_H)^2 = \frac{1}{N^2}\boldsymbol{W}\boldsymbol{X}\boldsymbol{Y}^\top\boldsymbol{Y}\boldsymbol{X}^\top\boldsymbol{W}^\top
$$

has nonzero eigenvalues $\lambda_1^2, \ldots, \lambda_r^2$, so $\frac{1}{N}\boldsymbol{W}\boldsymbol{X}\boldsymbol{Y}^\top\boldsymbol{W}_H$ has eigenvalues $\lambda_1, \ldots, \lambda_r$. Thus $\frac{1}{N}\operatorname{Tr}(\boldsymbol{W}\boldsymbol{X}\boldsymbol{Y}^\top\boldsymbol{W}_H) = \sum_{i=1}^r \lambda_i$ and $(\boldsymbol{W}, \boldsymbol{W}_H)$ as constructed achieves the minimum value of $\mathcal{L}_{diet}$. This completes the proof. $\square$

With the above two results, we obtain the desired result:

**Theorem 5.** *Suppose that Assumption 4 holds and $f$ is a parametric feature model $f(\boldsymbol{x}) = \boldsymbol{W}\phi(\boldsymbol{x})$. Then,*
- *If $(\boldsymbol{W}, \boldsymbol{W}_H)$ is a global minimizer of $\mathcal{L}_{\mathrm{DIET}}^{\mathrm{MSE}}$ and $\boldsymbol{W}_H$ is column-orthogonal, then $\boldsymbol{W}$ is a global minimizer of $\mathcal{L}_{\mathrm{SCL}}$.*
- *If $\boldsymbol{W}$ is a global minimizer of $\mathcal{L}_{\mathrm{SCL}}$, then there exists $\boldsymbol{W}_H$ such that $\boldsymbol{W}_H$ is column-orthogonal and $(\boldsymbol{W}, \boldsymbol{W}_H)$ is a global minimizer of $\mathcal{L}_{\mathrm{DIET}}^{\mathrm{MSE}}$.*

*Proof.* The first claim is immediate from Theorems 6 and 7. For the second claim, we in fact constructed the necessary $\boldsymbol{W}_H$ in the proof of Theorem 7. $\square$

### A.3 Proof of Theorem 3

We will first prove the claim about $\mathcal{L}_{diet}$. Then we will prove the claim about $\mathcal{L}_{diet-norm}^{mse}$ in a sequence of lemmas.

**Lemma 3.** *If $\boldsymbol{W}$ is a minimizer of $\mathcal{L}_{diet}^{mse}$ as defined in Equation 5, then*

$$\frac{\|\boldsymbol{W}\boldsymbol{v}_c\|}{\|\boldsymbol{W}\boldsymbol{u}_c\|} = \frac{\sigma_1^2}{\sigma_2^2} + o(1)$$

*Proof.* Since $\boldsymbol{W}_H$ is fixed, minimizing $\mathcal{L}_{diet}$ is in fact just standard linear regression. The closed form solution is well known:

$$\boldsymbol{W} = \boldsymbol{W}_H^\top \left(\frac{1}{n}\sum_{i=1}^n \mathbb{E}_A[\boldsymbol{e}_i A(\boldsymbol{x}_i)^\top]\right)\left(\frac{1}{n}\sum_{i=1}^n \mathbb{E}_A[A(\boldsymbol{x}_i)A(\boldsymbol{x}_i)^\top]\right)^{-1} \tag{15}$$

Now we calculate

$$
\begin{aligned}
\mathbb{E}_A[A(\boldsymbol{x}_i)A(\boldsymbol{x}_i)^\top] &= \mathbb{E}_A[((1+\epsilon_1)\boldsymbol{u}_{C(i)} + (1+\epsilon_2)\boldsymbol{v}_{C(i)} + \boldsymbol{\xi})((1+\epsilon_1)\boldsymbol{u}_{C(i)} + (1+\epsilon_2)\boldsymbol{v}_{C(i)} + \boldsymbol{\xi})^\top] \\
&= (1+\sigma_1^2)\boldsymbol{u}_{C(i)}\boldsymbol{u}_{C(i)}^\top + \boldsymbol{v}_{C(i)}\boldsymbol{u}_{C(i)}^\top + \boldsymbol{u}_{C(i)}\boldsymbol{v}_{C(i)}^\top + (1+\sigma_2^2)\boldsymbol{v}_{C(i)}\boldsymbol{v}_{C(i)}^\top \\
&\quad + \frac{\phi^2}{d}(\boldsymbol{I}_d - \boldsymbol{u}_{C(i)}\boldsymbol{u}_{C(i)}^\top - \boldsymbol{v}_{C(i)}\boldsymbol{v}_{C(i)}^\top)
\end{aligned}
$$

Therefore

$$
\begin{aligned}
\frac{1}{n}\sum_{i=1}^n \mathbb{E}_A[A(\boldsymbol{x}_i)A(\boldsymbol{x}_i)^\top] &= \frac{1}{n}\sum_{i=1}^n (1+\sigma_1^2)\boldsymbol{u}_{C(i)}\boldsymbol{u}_{C(i)}^\top + \boldsymbol{u}_{C(i)}\boldsymbol{v}_{C(i)}^\top(1+\sigma_2^2)\boldsymbol{v}_{C(i)}\boldsymbol{v}_{C(i)}^\top + \boldsymbol{v}_{C(i)}\boldsymbol{u}_{C(i)}^\top \\
&\quad + \frac{\phi^2}{d}(\boldsymbol{I}_d - \boldsymbol{u}_{C(i)}\boldsymbol{u}_{C(i)}^\top - \boldsymbol{v}_{C(i)}\boldsymbol{v}_{C(i)}^\top) \\
&= \frac{1}{C}\left(\sum_{c=1}^C \alpha_1 \boldsymbol{u}_c\boldsymbol{u}_c^\top + \boldsymbol{u}_c\boldsymbol{v}_c^\top + \boldsymbol{v}_c\boldsymbol{u}_c^\top + \alpha_2\boldsymbol{v}_c\boldsymbol{v}_c^\top\right) + \frac{\phi^2}{d}(\boldsymbol{I}_d - \sum_{c=1}^C \boldsymbol{u}_c\boldsymbol{u}_c^\top - \boldsymbol{v}_c\boldsymbol{v}_c^\top)
\end{aligned}
$$

where we set $\alpha_1 = 1 + \sigma_1^2 + \frac{(C-1)\phi^2}{d}, \alpha_2 = 1 + \sigma_2^2 + \frac{(C-1)\phi^2}{d}$. Taking the inverse,

$$
\begin{aligned}
\left(\frac{1}{n}\sum_{i=1}^n \mathbb{E}_A[A(\boldsymbol{x}_i)A(\boldsymbol{x}_i)^\top]\right)^{-1} &= \frac{C}{\alpha_1\alpha_2 - 1}\left(\sum_{c=1}^C \alpha_2\boldsymbol{u}_c\boldsymbol{u}_c^\top - \boldsymbol{u}_c\boldsymbol{v}_c^\top - \boldsymbol{v}_c\boldsymbol{u}_c^\top + \alpha_1\boldsymbol{v}_c\boldsymbol{v}_c^\top\right) \\
&\quad + \frac{d}{\phi^2}(\boldsymbol{I}_d - \sum_{c=1}^C \boldsymbol{u}_c\boldsymbol{u}_c^\top - \boldsymbol{v}_c\boldsymbol{v}_c^\top)
\end{aligned}
$$

Also, we have

$$
\begin{aligned}
\frac{1}{n}\sum_{i=1}^n \mathbb{E}_A[\boldsymbol{e}_i A(\boldsymbol{x}_i)^\top] &= \frac{1}{n}\sum_{i=1}^n \mathbb{E}_A[\boldsymbol{e}_i((1+\epsilon_1)\boldsymbol{u}_{C(i)} + (1+\epsilon_2)\boldsymbol{v}_{C(i)} + \boldsymbol{\xi})^\top] \\
&= \frac{1}{n}\sum_{i=1}^n \boldsymbol{e}_i(\boldsymbol{u}_{C(i)} + \boldsymbol{v}_{C(i)})^\top
\end{aligned}
$$

Now using the previously calculated expressions,

$$
\begin{aligned}
\boldsymbol{W}\boldsymbol{u}_c &= \boldsymbol{W}_H^\top \left( \frac{1}{n} \sum_{i=1}^n \mathbb{E}_A[\boldsymbol{e}_i A(\boldsymbol{x}_i)^\top] \right) \left( \frac{1}{n} \sum_{i=1}^n \mathbb{E}_A[A(\boldsymbol{x}_i) A(\boldsymbol{x}_i)^\top] \right)^{-1} \boldsymbol{u}_c \\
&= \boldsymbol{W}_H^\top \left( \frac{1}{n} \sum_{i=1}^n \mathbb{E}_A[\boldsymbol{e}_i A(\boldsymbol{x}_i)^\top] \right) \left( \frac{C\alpha_2}{\alpha_1\alpha_2 - 1} \boldsymbol{u}_c - \frac{C}{\alpha_1\alpha_2 - 1} \boldsymbol{v}_c \right) \\
&= \boldsymbol{W}_H^\top \left( \frac{1}{n} \sum_{C(i)=c} \left( \frac{C\alpha_2}{\alpha_1\alpha_2 - 1} - \frac{C}{\alpha_1\alpha_2 - 1} \right) \boldsymbol{e}_i \right) \\
&= \frac{C(\alpha_2 - 1)}{n(\alpha_1\alpha_2 - 1)} \sum_{C(i)=c} \boldsymbol{W}_H^\top \boldsymbol{e}_i \\
&= \frac{C(\sigma_2^2 + \frac{(C-1)\phi^2}{d})}{n(\alpha_1\alpha_2 - 1)} \sum_{C(i)=c} \boldsymbol{W}_H^\top \boldsymbol{e}_i
\end{aligned}
$$

Similarly, we have

$$
\boldsymbol{W}\boldsymbol{v}_c = \frac{C(\sigma_1^2 + \frac{(C-1)\phi^2}{d})}{n(\alpha_1\alpha_2 - 1)} \sum_{C(i)=c} \boldsymbol{W}_H^\top \boldsymbol{e}_i
$$

It follows that

$$
\begin{aligned}
\frac{\|\boldsymbol{W}\boldsymbol{v}_c\|}{\|\boldsymbol{W}\boldsymbol{u}_c\|} &= \frac{\frac{C(\sigma_1^2 + \frac{(C-1)\phi^2}{d})}{n(\alpha_1\alpha_2 - 1)}}{\frac{C(\sigma_2^2 + \frac{(C-1)\phi^2}{d})}{n(\alpha_1\alpha_2 - 1)}} \\
&= \frac{\sigma_1^2 + \frac{(C-1)\phi^2}{d}}{\sigma_2^2 + \frac{(C-1)\phi^2}{d}}
\end{aligned}
$$

Using the fact that $C = o(d)$, this shows that $\frac{\|\boldsymbol{W}\boldsymbol{v}_c\|}{\|\boldsymbol{W}\boldsymbol{u}_c\|} = \frac{\sigma_1^2}{\sigma_2^2} + o(1)$, as desired. $\qquad\square$

For normalized diet, we prove the result via the following lemmas. First we define some notation.
Let $C(i) \in \mathcal{C}$ represent the concept associated with $\boldsymbol{x}_i$, and set $\boldsymbol{r}_c = \sum_{C(i)=c} \boldsymbol{W}_H^\top \boldsymbol{e}_i$. Also as shorthand we write

$$
\mathcal{L}_{diet-norm}^{(i)} = \frac{1}{2} \mathbb{E}_A[\|\boldsymbol{W}_H(norm(\boldsymbol{W}(A(\boldsymbol{x}_i)))) - \boldsymbol{e}_i\|^2]
$$

$$
\mathcal{L}_{diet-norm}^{MSE} = \frac{1}{n} \sum_{i=1}^n \mathcal{L}_{diet-norm}^{(i)}
$$

**Lemma 4** (Useful facts)**.** *In the assumed setting, the following hold*

1. *If $i \neq j$, then $(\boldsymbol{W}_H^\top \boldsymbol{e}_i)^\top (\boldsymbol{W}_H^\top \boldsymbol{e}_j) = 0$*

2. *If $C(i) = c$, then $\boldsymbol{r}_c^\top \boldsymbol{W}_H^\top \boldsymbol{e}_i = h_i^2$.*

*Proof.* Since $\boldsymbol{W}_H$ is an isometry by assumption, $\boldsymbol{W}_H^\top$ is a partial isometry. Since $\boldsymbol{e}_i \perp \boldsymbol{e}_j$, the first claim follows.
For the second claim, we calculate that

$$
\begin{aligned}
\boldsymbol{r}_c^\top \boldsymbol{W}_H^\top \boldsymbol{e}_i &= \sum_{C(j)=c} (\boldsymbol{W}_H^\top \boldsymbol{e}_j)^\top \boldsymbol{W}_H^\top \boldsymbol{e}_i \\
&= (\boldsymbol{W}_H^\top \boldsymbol{e}_i)^\top \boldsymbol{W}_H^\top \boldsymbol{e}_i \\
&= h_i^2
\end{aligned}
$$

$\qquad\square$

**Lemma 5** (Step 1). *When training with $\mathcal{L}^{MSE}_{diet-norm}$, at every step in training $\boldsymbol{W}\boldsymbol{u}_c$ and $\boldsymbol{W}\boldsymbol{v}_c$ are parallel to $\boldsymbol{r}_c$, and $\boldsymbol{W}\boldsymbol{p} = 0$ for any $\boldsymbol{p}$ orthogonal to all the $\boldsymbol{u}_c$ and $\boldsymbol{v}_c$.*

*Proof.* We proceed by induction on the iteration of SGD.

The base case follows from the initialization $\boldsymbol{W} = 0$.

For the inductive step, we calculate the change due to the gradient descent update.

We first note that the inductive hypothesis implies the following useful fact: if $C(i) = c$ and $\boldsymbol{q} \in \mathbb{R}^d$ is orthogonal to $\boldsymbol{u}_c$ and $\boldsymbol{v}_c$, then $\boldsymbol{W}\boldsymbol{q} \in Span(\{\boldsymbol{r}_{c'} : c' \neq c\})$. In particular, by Lemma 4, $\boldsymbol{W}\boldsymbol{q}$ and $\boldsymbol{W}_H^\top \boldsymbol{e}_i$ are orthogonal.

Now denoting $\boldsymbol{x}_i^A = A(\boldsymbol{x}_i), \boldsymbol{z}_i^A = \boldsymbol{W}\boldsymbol{x}_i^A$, the gradient is

$$\frac{\partial \mathcal{L}^{(i)}_{diet-norm}}{\partial \boldsymbol{W}} = \mathbb{E}_A\left[\frac{1}{\|\boldsymbol{z}_i^A\|}\left(\boldsymbol{I} - \frac{1}{\|\boldsymbol{z}_i^A\|^2}\boldsymbol{z}_i^A(\boldsymbol{z}_i^A)^\top\right)\boldsymbol{W}_H^\top(\boldsymbol{W}_H\boldsymbol{z}_i^A - \boldsymbol{e}_i)(\boldsymbol{x}_i^A)^\top\right]$$

$$= \mathbb{E}_A\left[\frac{1}{\|\boldsymbol{z}_i^A\|}\left(\boldsymbol{I} - \frac{1}{\|\boldsymbol{z}_i^A\|^2}\boldsymbol{z}_i^A(\boldsymbol{z}_i^A)^\top\right)\boldsymbol{z}_i^A(\boldsymbol{x}_i^A)^\top - \frac{1}{\|\boldsymbol{z}_i^A\|}\left(\boldsymbol{I} - \frac{1}{\|\boldsymbol{z}_i^A\|^2}\boldsymbol{z}_i^A(\boldsymbol{z}_i^A)^\top\right)\boldsymbol{W}_H^\top\boldsymbol{e}_i(\boldsymbol{x}_i^A)^\top\right]$$

$$= -\mathbb{E}_A\left[\frac{1}{\|\boldsymbol{z}_i^A\|}\left(\boldsymbol{I} - \frac{1}{\|\boldsymbol{z}_i^A\|^2}\boldsymbol{z}_i^A(\boldsymbol{z}_i^A)^\top\right)\boldsymbol{W}_H^\top\boldsymbol{e}_i(\boldsymbol{x}_i^A)^\top\right]$$

Thus

$$\frac{\partial \mathcal{L}^{(i)}_{diet-norm}}{\partial \boldsymbol{W}}\boldsymbol{u}_c = -\mathbb{E}_A\left[\frac{(\boldsymbol{x}_i^A)^\top\boldsymbol{u}_c}{\|\boldsymbol{z}_i^A\|}\left(\boldsymbol{I} - \frac{1}{\|\boldsymbol{z}_i^A\|^2}\boldsymbol{z}_i^A(\boldsymbol{z}_i^A)^\top\right)\boldsymbol{W}_H^\top\boldsymbol{e}_i\right]$$

We now consider two cases. First assume $C(i) = c$. Writing $\boldsymbol{x}_i^A = (1 + \epsilon_1)\boldsymbol{u}_c + (1 + \epsilon_2)\boldsymbol{v}_c + \boldsymbol{\xi}$, and $\boldsymbol{W}((1 + \epsilon_1)\boldsymbol{u}_c + (1 + \epsilon_2)\boldsymbol{v}_c) = \alpha_c\boldsymbol{r}_c$

$$\frac{\partial \mathcal{L}^{(i)}_{diet-norm}}{\partial \boldsymbol{W}}\boldsymbol{u}_c = \mathbb{E}_A\left[\frac{1 + \epsilon_1}{\|\boldsymbol{z}_i^A\|}\left(\boldsymbol{I} - \frac{1}{\|\boldsymbol{z}_i^A\|^2}(\alpha_c\boldsymbol{r}_c + \boldsymbol{W}\boldsymbol{\xi})(\alpha_c\boldsymbol{r}_c + \boldsymbol{W}\boldsymbol{\xi})^\top\right)\boldsymbol{W}_H^\top\boldsymbol{e}_i\right]$$

Now by the symmetry of the noise distribution, we can replace $\boldsymbol{\xi}$ with $-\boldsymbol{\xi}$. By induction, $\boldsymbol{W}\boldsymbol{\xi}$ is orthogonal to $\boldsymbol{r}_c$, this does not change $\|\boldsymbol{z}_i\|$, so the above is equal to

$$= \mathbb{E}_A\left[\frac{1 + \epsilon_1}{\|\boldsymbol{z}_i^A\|}\left(\boldsymbol{I} - \frac{1}{2\|\boldsymbol{z}_i^A\|^2}((\alpha_c\boldsymbol{r}_c + \boldsymbol{W}\boldsymbol{\xi})(\alpha_c\boldsymbol{r}_c + \boldsymbol{W}\boldsymbol{\xi})^\top + (\alpha_c\boldsymbol{r}_c - \boldsymbol{W}\boldsymbol{\xi})(\alpha_c\boldsymbol{r}_c - \boldsymbol{W}\boldsymbol{\xi})^\top)\right)\boldsymbol{W}_H^\top\boldsymbol{e}_i\right]$$

$$= \mathbb{E}_A\left[\frac{1 + \epsilon_1}{\|\boldsymbol{z}_i^A\|}\left(\boldsymbol{I} - \frac{1}{\|\boldsymbol{z}_i^A\|^2}(\alpha_c^2\boldsymbol{r}_c\boldsymbol{r}_c^\top + (\boldsymbol{W}\boldsymbol{\xi})(\boldsymbol{W}\boldsymbol{\xi})^\top)\right)\boldsymbol{W}_H^\top\boldsymbol{e}_i\right]$$

Using the useful fact from above and Lemma 4, this is equal to

$$= \mathbb{E}_A\left[\frac{1 + \epsilon_1}{\|\boldsymbol{z}_i^A\|}\boldsymbol{W}_H^\top\boldsymbol{e}_i - \frac{(1 + \epsilon_1)\alpha_c^2\boldsymbol{r}_c^\top\boldsymbol{W}_H^\top\boldsymbol{e}_i}{\|\boldsymbol{z}_i^A\|^3}\boldsymbol{r}_c\right]$$

$$= \mathbb{E}_A\left[\frac{1 + \epsilon_1}{\|\boldsymbol{z}_i^A\|}\boldsymbol{W}_H^\top\boldsymbol{e}_i - \frac{(1 + \epsilon_1)\alpha_c^2 h_c^2}{\|\boldsymbol{z}_i^A\|^3}\boldsymbol{r}_c\right]$$

Now suppose $C(i) = c' \neq C$. A similar calculation shows that

$$\frac{\partial \mathcal{L}^{(i)}_{diet-norm}}{\partial \boldsymbol{W}}\boldsymbol{u}_c = \mathbb{E}_A\left[\frac{\boldsymbol{\xi}^\top\boldsymbol{u}_c}{\|\boldsymbol{z}_i^A\|}\left(\boldsymbol{I} - \frac{1}{\|\boldsymbol{z}_i^A\|^2}(\alpha_{c'}\boldsymbol{r}_{c'} + \boldsymbol{W}\boldsymbol{\xi})(\alpha_{c'}\boldsymbol{r}_{c'} + \boldsymbol{W}\boldsymbol{\xi})^\top\right)\boldsymbol{W}_H^\top\boldsymbol{e}_i\right]$$

Again using the symmetry of the noise and the useful fact, this is equal to

$$
\begin{aligned}
&= \frac{1}{2}\mathbb{E}_A\left[\frac{\boldsymbol{\xi}^\top \boldsymbol{u}_c}{\|\boldsymbol{z}_i^A\|}(\boldsymbol{I} - \frac{1}{\|\boldsymbol{z}_i^A\|^2}(\alpha_{c'}\boldsymbol{r}_{c'} + \boldsymbol{W}\boldsymbol{\xi})(\alpha_{c'}\boldsymbol{r}_{c'} + \boldsymbol{W}\boldsymbol{\xi})^\top)\boldsymbol{W}_H^\top \boldsymbol{e}_i\right.\\
&\qquad\left. + \frac{-\boldsymbol{\xi}^\top \boldsymbol{u}_c}{\|\boldsymbol{z}_i^A\|}(\boldsymbol{I} - \frac{1}{\|\boldsymbol{z}_i^A\|^2}(\alpha_{c'}\boldsymbol{r}_{c'} - \boldsymbol{W}\boldsymbol{\xi})(\alpha_{c'}\boldsymbol{r}_{c'} - \boldsymbol{W}\boldsymbol{\xi})^\top)\boldsymbol{W}_H^\top \boldsymbol{e}_i\right]\\
&= -\mathbb{E}_A\left[\frac{\boldsymbol{\xi}^\top \boldsymbol{u}_c}{\|\boldsymbol{z}_i^A\|^3}\left(\alpha_{c'}\boldsymbol{r}_{c'}(\boldsymbol{W}\boldsymbol{\xi})^\top + (\boldsymbol{W}\boldsymbol{\xi})(\alpha_{c'}\boldsymbol{r}_{c'})^\top\right)\boldsymbol{W}_H^\top \boldsymbol{e}_i\right]\\
&= -\mathbb{E}_A\left[\frac{\alpha_{c'}(\boldsymbol{\xi}^\top \boldsymbol{u}_c)(\boldsymbol{r}_{c'}^\top \boldsymbol{W}_H^\top \boldsymbol{e}_i)}{\|\boldsymbol{z}_i^A\|^3}\boldsymbol{W}\boldsymbol{\xi}\right]\\
&= -\mathbb{E}_A\left[\frac{\alpha_{c'}h_{c'}^2(\boldsymbol{\xi}^\top \boldsymbol{u}_c)}{\|\boldsymbol{z}_i^A\|^3}\boldsymbol{W}\boldsymbol{\xi}\right]
\end{aligned}
$$

Now isolating the component of $\boldsymbol{\xi}$ along $\boldsymbol{u}_c$, write $\boldsymbol{\xi} = \xi_c\boldsymbol{u}_c + \boldsymbol{\xi}'$. Again by the symmetry of the noise, we can consider replacing $\boldsymbol{\xi}'$ with $-\boldsymbol{\xi}$, so

$$
\begin{aligned}
-\mathbb{E}_A\left[\frac{\alpha_{c'}h_{c'}^2(\boldsymbol{\xi}^\top \boldsymbol{u}_c)}{\|\boldsymbol{z}_i\|^3}\boldsymbol{W}\boldsymbol{\xi}\right] &= -\mathbb{E}_A\left[\frac{\alpha_{c'}h_{c'}^2\xi_c}{2\|\boldsymbol{z}_i\|^3}\boldsymbol{W}(\xi_c\boldsymbol{u}_c + \boldsymbol{\xi}' + \xi_c\boldsymbol{u}_c - \boldsymbol{\xi}')\right]\\
&= -\mathbb{E}_A\left[\frac{\alpha_{c'}h_{c'}^2\xi_c^2}{\|\boldsymbol{z}_i\|^3}\boldsymbol{W}\boldsymbol{u}_c\right]
\end{aligned}
$$

Combining all these results, we have

$$
\begin{aligned}
\frac{\partial \mathcal{L}_{diet-norm}^{MSE}}{\partial \boldsymbol{W}}\boldsymbol{u}_c &= \frac{1}{n}\sum_{i=1}^n \frac{\partial \mathcal{L}_{diet-norm}^{(i)}}{\partial \boldsymbol{W}}\\
&= -\frac{1}{n}\sum_{C(i)=c}\mathbb{E}_A\left[\frac{1+\epsilon_1^{(i)}}{\|\boldsymbol{z}_i^A\|}\boldsymbol{W}_H^\top \boldsymbol{e}_i - \frac{(1+\epsilon_1^{(i)})(\alpha_c^{(i)})^2 h_c^2}{\|\boldsymbol{z}_i^A\|^3}\boldsymbol{r}_c\right]\\
&\qquad + \frac{1}{n}\sum_{C(i)=c'\neq c}\mathbb{E}_A\left[\frac{\alpha_{c'}^{(i)}(\xi_c^{(i)})^2 h_{c'}^2}{\|\boldsymbol{z}_i^A\|^3}\boldsymbol{W}\boldsymbol{u}_c\right]\\
&= -\mathbb{E}_A\left[\frac{1+\epsilon_1}{\|\boldsymbol{z}^A\|}\right]\boldsymbol{r}_c + \frac{1}{n}\sum_{C(i)=c}\mathbb{E}_A\left[\frac{(1+\epsilon_1^{(i)})(\alpha_c^{(i)})^2 h_c^2}{\|\boldsymbol{z}_i^A\|^3}\right]\boldsymbol{r}_c\\
&\qquad + \frac{1}{n}\sum_{C(i)=c'\neq c}\mathbb{E}_A\left[\frac{\alpha_{c'}^{(i)}(\xi_c^{(i)})^2 h_{c'}^2}{\|\boldsymbol{z}_i^A\|^3}\right]\boldsymbol{W}\boldsymbol{u}_c
\end{aligned}
$$

By the inductive hypothesis $\boldsymbol{W}\boldsymbol{u}_c$ is parallel to $\boldsymbol{r}_c$, so the change from the gradient update is parallel to $\boldsymbol{r}_c$. The same argument shows that $\boldsymbol{W}\boldsymbol{v}_c$ is parallel to $\boldsymbol{r}_c$.

Now consider any $\boldsymbol{p}$ orthogonal to all the $\boldsymbol{v}_i$ and $\boldsymbol{u}_i$. We calculate that

$$
\frac{\partial \mathcal{L}_{diet-norm}^{MSE}}{\partial \boldsymbol{W}}\boldsymbol{p} = -\frac{1}{n}\sum_{i=1}^n \mathbb{E}_A\left[\frac{(\boldsymbol{x}_i^A)^\top \boldsymbol{p}}{\|\boldsymbol{z}_i^A\|}\left(\boldsymbol{I} - \frac{1}{\|\boldsymbol{z}_i\|^2}\boldsymbol{z}_i^A(\boldsymbol{z}_i^A)^\top\right)\boldsymbol{W}_H^\top \boldsymbol{e}_i\right]
$$

Decomposing $\boldsymbol{x} = \beta\boldsymbol{p} + \gamma$, by the symmetry of the noise we can replace $\beta$ with $-\beta$. Since $\boldsymbol{W}\boldsymbol{p} = \boldsymbol{0}$ by induction $\boldsymbol{z}_i$ does not change, so we have

$$
\begin{aligned}
\frac{\partial \mathcal{L}_{diet-norm}^{MSE}}{\partial \boldsymbol{W}}\boldsymbol{p} &= -\frac{1}{2n}\sum_{i=1}^n \mathbb{E}_A\left[\frac{(\boldsymbol{x}_i^A)^\top \boldsymbol{p} - (\boldsymbol{x}_i^A)^\top \boldsymbol{p}}{\|\boldsymbol{z}_i^A\|}\left(\boldsymbol{I} - \frac{1}{\|\boldsymbol{z}_i\|^2}\boldsymbol{z}_i^A(\boldsymbol{z}_i^A)^\top\right)\boldsymbol{W}_H^\top \boldsymbol{e}_i\right]\\
&= \boldsymbol{0}
\end{aligned}
$$

Thus the change from the gradient update is $\mathbf{0}$,

This completes the induction. $\qquad\square$

**Lemma 6** (Step 2). *Assume that we train to convergence using $\mathcal{L}_{diet-norm}^{MSE}$. Then $\boldsymbol{r}_c^\top \boldsymbol{W} \boldsymbol{u}_c$, $\boldsymbol{r}_c^\top \boldsymbol{W} \boldsymbol{v}_c \neq 0$.*

*Proof.* Using the gradient calculation from the proof of step 1:

$$\boldsymbol{r}_c^\top \frac{\partial \mathcal{L}_{diet-norm}^{MSE}}{\partial \boldsymbol{W}} \boldsymbol{u}_c = -\mathbb{E}_A \left[ \frac{1+\epsilon_1}{\|\boldsymbol{z}^A\|} \right] \|\boldsymbol{r}_c\|^2 + \frac{1}{n} \sum_{C(i)=c} \mathbb{E}_A \left[ \frac{(1+\epsilon_1^{(i)})(\alpha_c^{(i)})^2 h_c^2}{\|\boldsymbol{z}_i^A\|^3} \right] \|\boldsymbol{r}_c\|^2$$

$$+ \frac{1}{n} \sum_{C(i)=c' \neq c} \mathbb{E}_A \left[ \frac{\alpha_{c'}^{(i)} (\xi_c^{(i)})^2 h_{c'}^2}{\|\boldsymbol{z}_i^A\|^3} \right] \boldsymbol{r}_c^\top \boldsymbol{W} \boldsymbol{u}_c$$

$$= -\mathbb{E}_A \left[ \frac{1+\epsilon_1}{\|\boldsymbol{z}^A\|} \right] \|\boldsymbol{r}_c\|^2 + \frac{1}{n} \sum_{C(i)=c} \mathbb{E}_A \left[ \frac{(1+\epsilon_1^{(i)})(\alpha_c^{(i)})^2 h_c^2}{\|\boldsymbol{z}_i^A\|^3} \right] \|\boldsymbol{r}_c\|^2$$

Recall from Lemma 4 that $h_c^2 \leq 1$. Also note that by definition $(\alpha_c^{(i)})^2 \leq \|\boldsymbol{z}_i\|^2$. Hence

$$\boldsymbol{r}_c^\top \frac{\partial \mathcal{L}_{diet-norm}^{MSE}}{\partial \boldsymbol{W}} \boldsymbol{u}_c = -\mathbb{E}_A \left[ \frac{1+\epsilon_1}{\|\boldsymbol{z}^A\|} \right] + \frac{1}{n} \sum_{C(i)=c} \mathbb{E}_A \left[ \frac{(1+\epsilon_1^{(i)})(\alpha_c^{(i)})^2 h_c^2}{\|\boldsymbol{z}_i^A\|^3} \right] < 0$$

This implies that $\boldsymbol{r}_c^\top \frac{\partial \mathcal{L}_{diet-norm}^{MSE}}{\partial \boldsymbol{W}} \boldsymbol{u}_c < 0$. This contradicts the fact that we have converged to a point where $\frac{\partial \mathcal{L}_{diet-norm}^{MSE}}{\partial \boldsymbol{W}} = \mathbf{0}$. $\qquad\square$

**Lemma 7** (Proof of Theorem for Normalized Diet). *Assume that we train to convergence using $\mathcal{L}_{diet}^{norm}$. Then*

$$\frac{1-\nu_1}{1+\nu_2} \leq \frac{\|\boldsymbol{W} \boldsymbol{v}_c\|}{\|\boldsymbol{W} \boldsymbol{u}_c\|} \leq \frac{1+\nu_1}{1-\nu_2}$$

*Proof.* From the previous steps, there exists $a_1, \ldots, a_C, b_1, \ldots, b_C \neq 0$ such that $\boldsymbol{W} \boldsymbol{u}_c = a_c \boldsymbol{r}_c$ and $\boldsymbol{W} \boldsymbol{v}_c = b_c \boldsymbol{r}_c$. Therefore, for a given example $\boldsymbol{x}_i$ with $C(i) = c$, the distribution $\boldsymbol{W}(A(\boldsymbol{x}_i))$ over choice of augmentation $A$ takes the form

$$(a_c + a_c \epsilon_1 + b_c + b_c \epsilon_2) \boldsymbol{r}_c + \sum_{c' \neq c} \frac{\phi^2}{d} \sqrt{a_{c'}^2 + b_{c'}^2} \xi_{c'} \boldsymbol{r}_{c'} \tag{16}$$

where $\epsilon_1 \sim \mathcal{G}_1, \epsilon_2 \sim \mathcal{G}_2$, and $\xi_{c'} \sim \mathcal{N}(0,1)$ for each $c'$. To ease notation, set $\kappa_c = a_c + a_c \epsilon_1 + b_c + b_c \epsilon_2$ and $\lambda_c = \frac{\phi^2}{d} \sqrt{a_c^2 + b_c^2} \xi_c$. Note that since the $\boldsymbol{r}_c$ are orthogonal, $\|\boldsymbol{W}(A(\boldsymbol{x}_i))\|$ follows the distribution

$$\sqrt{\kappa_c^2 \|\boldsymbol{r}_c\|^2 + \sum_{c' \neq c} \lambda_{c'}^2 \|\boldsymbol{r}_{c'}\|^2}$$

We can now treat the loss as a multivariate function in $a_1, \ldots, a_C, b_1, \ldots, b_C$. Suppose we vary $a_c$ and $b_c$ such that $a_c da_c + b_c db_c = 0$. It suffices to calculate the directional derivative induced by this variation and show that it cannot be zero if $|\frac{b}{a}| > \frac{1+\nu_1}{1-\nu_2}$ or $|\frac{b}{a}| < \frac{1-\nu_1}{1+\nu_2}$.

The loss term due to an example $\boldsymbol{x}_i$ is

$$\mathcal{L}_{diet-norm}^{(i)} = \frac{1}{2}\mathbb{E}_A[\|\boldsymbol{W}_H(norm(\boldsymbol{W}(A(\boldsymbol{x}_i)))) - \boldsymbol{e}_i\|^2]$$

$$= \frac{1}{2}\mathbb{E}\left[\left\|\frac{\boldsymbol{W}_H(\kappa_{C(i)}\boldsymbol{r}_{C(i)} + \sum_{c'\neq C(i)}\lambda_{c'}\boldsymbol{r}_{c'})}{\sqrt{\kappa_{C(i)}^2\|\boldsymbol{r}_{C(i)}\|^2 + \sum_{c'\neq C(i)}\lambda_{c'}^2\|\boldsymbol{r}_{c'}\|^2}} - \boldsymbol{e}_i\right\|^2\right]$$

$$= 1 - \mathbb{E}\left[\frac{\boldsymbol{e}_i^\top\boldsymbol{W}_H(\kappa_{C(i)}\boldsymbol{r}_{C(i)} + \sum_{c'\neq C(i)}\lambda_{c'}\boldsymbol{r}_{c'})}{\sqrt{\kappa_{C(i)}^2\|\boldsymbol{r}_{C(i)}\|^2 + \sum_{c'\neq C(i)}\lambda_{c'}^2\|\boldsymbol{r}_{c'}\|^2}}\right]$$

$$= 1 - \mathbb{E}\left[\frac{\kappa_{C(i)}\boldsymbol{e}_i^\top\boldsymbol{W}_H\boldsymbol{r}_{C(i)}}{\sqrt{\kappa_{C(i)}^2\|\boldsymbol{r}_{C(i)}\|^2 + \sum_{c'\neq C(i)}\lambda_{c'}^2\|\boldsymbol{r}_{c'}\|^2}}\right]$$

$$= 1 - \mathbb{E}\left[\frac{\kappa_{C(i)}h_c^2}{\sqrt{\kappa_{C(i)}^2\|\boldsymbol{r}_{C(i)}\|^2 + \sum_{c'\neq C(i)}\lambda_{c'}^2\|\boldsymbol{r}_{c'}\|^2}}\right]$$

Observe that by construction $d(a_c^2 + b_c^2) = 0$, which implies $d\lambda_c = 0$. Thus if $C(i) \neq c$, the change in the loss $\mathcal{L}_{diet-norm}^{(i)}$ is zero.

On the other hand, if $C(i) = c$, we now calculate the derivatives

$$\frac{\partial}{\partial a_c}\mathcal{L}_{diet-norm}^{(i)} = \frac{\partial}{\partial a_c}\mathbb{E}_A[\|\boldsymbol{W}_H(norm(\boldsymbol{W}(A(\boldsymbol{x}_i)))) - \boldsymbol{e}_i\|^2]$$

$$= -\mathbb{E}\left[\frac{h_c^2\sum_{c'\neq c}\lambda_{c'}^2\|\boldsymbol{r}_{c'}\|^2}{(\kappa_c^2\|\boldsymbol{r}_c\|^2 + \sum_{c'\neq c}\lambda_{c'}^2\|\boldsymbol{r}_{c'}\|^2)^{\frac{3}{2}}}(1 + \epsilon_1)\right]$$

$$\frac{\partial}{\partial b_c}\mathcal{L}_{diet-norm}^{(i)} = -\mathbb{E}\left[\frac{h_c^2\sum_{c'\neq c}\lambda_{c'}^2\|\boldsymbol{r}_{c'}\|^2}{(\kappa_c^2\|\boldsymbol{r}_c\|^2 + \sum_{c'\neq c}\lambda_{c'}^2\|\boldsymbol{r}_{c'}\|^2)^{\frac{3}{2}}}(1 + \epsilon_2)\right]$$

Hence

$$d\mathcal{L}_{diet-norm}^{MSE} = \sum_{C(i)=c}\frac{\partial\mathcal{L}_{diet-norm}^{(i)}}{\partial a_c}da_c + \frac{\partial\mathcal{L}_{diet-norm}^{(i)}}{\partial b_c}db_c$$

$$= \sum_{C(i)=c}-\mathbb{E}\left[\frac{h_c^2\sum_{c'\neq c}\lambda_{c'}^2\|\boldsymbol{r}_{c'}\|^2}{(\kappa_c^2\|\boldsymbol{r}_c\|^2 + \sum_{c'\neq c}\lambda_{c'}^2\|\boldsymbol{r}_{c'}\|^2)^{\frac{3}{2}}}((1 + \epsilon_1)da_c + (1 + \epsilon_2)db_c)\right]$$

First consider the case that $\frac{a}{b} > 0$. Suppose for the sake of contradiction $|\frac{a}{b}| > \frac{1+\nu_1}{1-\nu_2}$. Then

$$0 > 1 + \nu_1 - \frac{a}{b} + \frac{a}{b}\nu_2$$

$$> 1 + \epsilon_1 - \frac{a}{b} + \frac{a}{b}(-\epsilon_2)$$

$$= (1 + \epsilon_1) - \frac{a}{b}(1 + \epsilon_2)$$

In the case that $\frac{a}{b} < 0$

$$0 > 1 + \nu_1 - \left|\frac{a}{b}\right| + \left|\frac{a}{b}\right|\nu_2$$

$$> 1 - \epsilon_1 + \frac{a}{b} - \frac{a}{b}\epsilon_2$$

$$= 2 - ((1 + \epsilon_1) - \frac{a}{b}(1 + \epsilon_2))$$

$$(1 + \epsilon_1) - \frac{a}{b}(1 + \epsilon_2) > 2$$

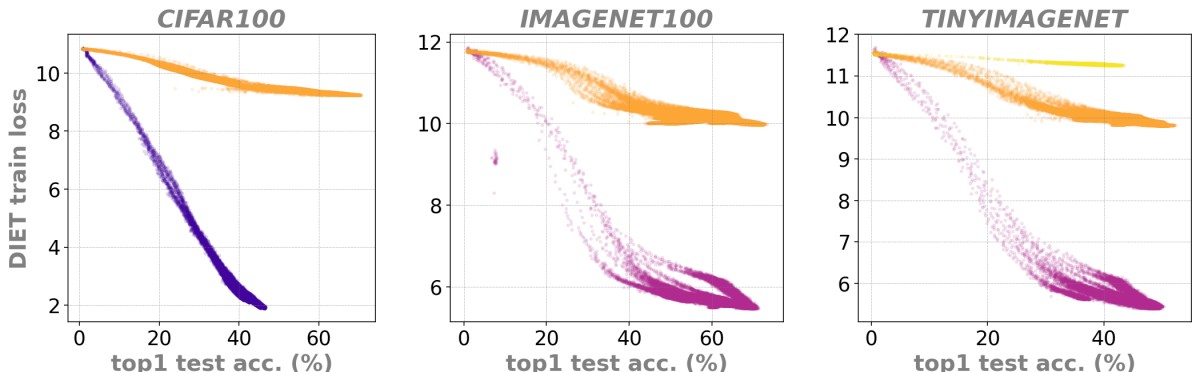

Figure 7: **DIET's training loss is indicative of downstream test performance.** We depict DIET's training loss (**y-axis**) against the online test linear probe accuracy (**x-axis**) for all the models, hyper-parameters, and training epochs. Yellow to purple correspond to different label smoothing which plays a role in DIET's convergence speed (Appx. C). For a given label smoothing parameter, there exists a strong relationship between **DIET**'s training loss and the downstream test accuracy enabling label-free quantitative quality assessment one's model.

Table 13: **DIET is competitive and works out-of-the-box across architectures.** We keep the settings of Fig. 9. Benchmarks from 1:Dubois et al. (2022), 2 :Ozsoy et al. (2022).

### Imagenet-100 (IN100)

| Resnet18 | | Resnet50 | |
|---|---|---|---|
| SimMoCo | 58.20* | | |
| MocoV2 | 60.52* | | |
| SimCo | 61.28 * | MoCo+Hyper. | 75.60 * |
| W-MSE2 | 69.06 [2] | MoCo+DCL | 76.80 * |
| ReSSL | 74.02● | MoCoV2 + Hyper. | 77.70 * |
| DINO | 74.16● | BYOL | 78.76 [2] |
| MoCoV2 | 76.48● | MoCoV2 + DCL | 80.50 * |
| BYOL | 76.60● | SimCLR | 80.70 * |
| SimCLR | 77.04[2] | SimSiam | 81.60[2] |
| SimCLR | 78.72[2] | SimCLR + DCL | 83.10 * |
| MocoV2 | 79.28[2] | | |
| VICReg | 79.40[2] | | |
| BarlowTwins | 80.38[2] | | |

| DIET | | | |
|---|---|---|---|
| *resnet18* | 64.31 | *resnet50* | 73.50 |
| *wide_resnet50_2* | 71.92 | *convnext_small* | 71.06 |
| *resnext50_32x4d* | 73.07 | *MLPMixer* | 56.46 |
| *densenet121* | 67.46 | *swin_t* | 67.02 |
| *convnext_tiny* | 69.77 | *vit_b_16* | 62.63 |

Either way $(1 + \epsilon_1) - \frac{a}{b}(1 + \epsilon_2)$ is strictly positive or strictly negative. Now write

$$(1 + \epsilon_1)da_c + (1 + \epsilon_2)db_c = \left((1 + \epsilon_1) - \frac{a}{b}(1 + \epsilon_2)\right)da_c$$

Combined with the fact that $\frac{h_c^2 \sum_{c' \neq c} \lambda_{c'}^2 \|\boldsymbol{r}_{c'}\|^2}{(\kappa_c^2 \|\boldsymbol{r}_c\|^2 + \sum_{c' \neq c} \lambda_{c'}^2 \|\boldsymbol{r}_{c'}\|^2)^{\frac{3}{2}}}$ is always nonnegative and not always zero, it follows that $d\mathcal{L}_{diet-norm}^{MSE} \neq 0$, contradicting the fact that we have converged to a local minima.

The same argument shows that $\frac{b}{a} \leq \frac{1+\nu_2}{1-\nu_1}$, giving the lower bound. $\qquad \square$

Table 14: **DIET trained on small datasets competes with Imagenet pre-trained SSL.** We also report performances for a ViT based architecture (SwinTiny) to demonstrate the ability of DIET to handle different models out-of-the-box following Fig. 9. Benchmarks from †:Yang et al. (2022), +:Ericsson et al. (2021)

| Arch. | Pretrain | Frozen | N= C= | Aircraft 6667 100 | DTD 1880 47 | Pets 2940 37 | Flower 1020 102 | CUB-200 11788 200 | Food101 68175 101 | Cars 6509 196 |
|---|---|---|---|---|---|---|---|---|---|---|
| *Resnet18* | IN100† | Yes | SimCLR | 24.19 | 54.35 | 46.46 | 75.00 | 16.73 | - | - |
| | | | +CLAE | 25.87 | 52.12 | 43.55 | 76.82 | 17.58 | - | - |
| | | | +IDAA | 26.02 | 54.97 | 46.76 | 77.99 | 18.15 | - | - |
| | None | No | DIET | 37.29 | 50.62 | 64.06 | 72.01 | 33.03 | 62.00 | 42.55 |
| *Resnet50* | IN-1k+ | Yes | InsDis | 36.87 | 68.46 | 68.78 | 83.44 | - | 63.39 | 28.98 |
| | | | MoCo | 35.55 | 68.83 | 69.84 | 82.10 | - | 62.10 | 27.99 |
| | | | PCL. | 21.61 | 62.87 | 75.34 | 64.73 | - | 48.02 | 12.93 |
| | | | PIRL | 37.08 | 68.99 | 71.36 | 83.60 | - | 64.65 | 28.72 |
| | | | PCLv2 | 37.03 | 70.59 | 82.79 | 85.34 | - | 64.88 | 30.51 |
| | | | SimCLR | 44.90 | 74.20 | 83.33 | 90.87 | - | 67.47 | 43.73 |
| | | | MoCov2 | 41.79 | 73.88 | 83.30 | 90.07 | - | 68.95 | 39.31 |
| | | | SimCLRv2 | 46.38 | 76.38 | 84.72 | 92.90 | - | 73.08 | 50.37 |
| | | | SeLav2 | 37.29 | 74.15 | 83.22 | 90.22 | - | 71.08 | 36.86 |
| | | | InfoMin | 38.58 | 74.73 | 86.24 | 87.18 | - | 69.53 | 41.01 |
| | | | BYOL | 53.87 | 76.91 | 89.10 | 94.50 | - | 73.01 | 56.40 |
| | | | DeepClusterv2 | 54.49 | 78.62 | 89.36 | 94.72 | - | 77.94 | 58.60 |
| | | | Swav | 54.04 | 77.02 | 87.60 | 94.62 | - | 76.62 | 54.06 |
| | None | No | DIET | 44.81 | 51.75 | 67.08 | 73.32 | 41.03 | 71.58 | 55.82 |
| *SwinTiny* | None | No | DIET | 33.15 | 51.88 | 58.06 | 70.78 | 32.11 | 68.86 | 47.12 |
| *Convnext-S* | None | No | DIET | 43.13 | 49.52 | 61.72 | 67.72 | 31.44 | 69.84 | 40.63 |

## A.4 Gradient Analysis

Consider a batch of $B$ representations $\{z_i\}_{i=1}^{B}$, with class probabilities $\{y_i\}_{i=1}^{B}$ for $N$ classes and a set of $N$ class prototypes $\{w_k\}_{k=1}^{N}$. The cross-entropy loss for a batch is given by:

$$\mathcal{L}_{\text{CE}} = -\frac{1}{B} \sum_{i=1}^{B} \sum_{k=1}^{N} y_{i,k} \log p(k|z_i),$$

where

$$p(k|z_i) = \frac{\exp(w_k^\top z_i)}{\sum_j \exp(w_j^\top z_i)}.$$

To compute the derivative with respect to $z_i$, we proceed as follows:

$$\frac{\partial \mathcal{L}_{\text{CE}}}{\partial z_i} = -\frac{1}{B} \sum_{k=1}^{N} y_{i,k} \frac{\partial}{\partial z_i} \log p(k|z_i).$$

Since

$$\log p(k|z_i) = w_k^\top z_i - \log \sum_j^N \exp(w_j^\top z_i),$$

we have

$$\frac{\partial}{\partial z_i} \log p(k|z_i) = w_k - \sum_j^N p(j|z_i) w_j.$$

Substituting back, we get:

$$\frac{\partial \mathcal{L}_{\text{CE}}}{\partial z_i} = -\frac{1}{B} \sum_{k=1}^{N} y_{i,k} \left( w_k - \sum_j^N p(j|z_i) w_j \right).$$

Table 15: **GPU Memory Usage** in MiB for s-DIET, DIET, and other SSL methods with a batch size of 256. OOM indicates out-of-memory on an Nvidia A40 GPU, which has 46068 MiB of memory. s-DIET reduces the memory requirements of DIET by more than 2x on large datasets such as TinyImageNet, and make it up to 2.2x more memory efficient than CL methods.

| Method | CIFAR | | ImageNet-100 | Tiny-Imagenet |
| | ResNet-18 | ResNet-50 | ResNet-50 | ResNet-50 |
|---|---|---|---|---|
| Barlow Twins | 4026 | 17090 | 44698 | 4532 |
| BYOL | 4512 | 17296 | (OOM) | 4842 |
| SimCLR | 3896 | 16408 | 40322 | 4352 |
| Simsiam | 3964 | 16562 | 45264 | 4390 |
| DIET | 2556 | 9720 | 31164 | 6676 |
| s-DIET | **2312** | **7770** | **23634** | **2976** |

Due to $\sum_{k=1}^{N} y_{i,k} = 1$, we simplify to:

$$\frac{\partial \mathcal{L}_{\text{CE}}}{\partial \boldsymbol{z}_i} = -\frac{1}{B} \sum_{k=1}^{N} \left( y_{i,k} - p(k|\boldsymbol{z}_i) \right) \boldsymbol{w}_k = -\frac{1}{B} \left( \sum_{k=1}^{N} y_{i,k} \boldsymbol{w}_k - \sum_{k=1}^{N} p(k|\boldsymbol{z}_i) \boldsymbol{w}_k \right)$$

To compute the derivative of $\mathcal{L}_{\text{CE}}$ with respect to $\boldsymbol{w}_k$, we start with:

$$\mathcal{L}_{\text{CE}} = -\frac{1}{B} \sum_{i=1}^{B} \sum_{k=1}^{N} y_{i,k} \log p(k|\boldsymbol{z}_i) = -\frac{1}{B} \sum_{i=1}^{B} \sum_{k=1}^{N} y_{i,k} \boldsymbol{w}_k^T z_i + \frac{1}{B} \sum_{i=1}^{B} \sum_{k=1}^{N} y_{i,k} \log \sum_{j=1}^{N} \exp(\boldsymbol{w}_j^T \boldsymbol{z}_i),$$

$$\mathcal{L}_{\text{CE}} = -\frac{1}{B} \sum_{i=1}^{B} \sum_{k=1}^{N} y_{i,k} \log p(k|\boldsymbol{z}_i) = -\frac{1}{B} \sum_{i=1}^{B} \sum_{k=1}^{N} y_{i,k} \boldsymbol{w}_k^T \boldsymbol{z}_i + \frac{1}{B} \sum_{i=1}^{B} \log \sum_{k=1}^{N} \exp(\boldsymbol{w}_j^T \boldsymbol{z}_i),$$

We then apply the partial derivation operator,

$$\frac{\partial \mathcal{L}_{\text{CE}}}{\partial \boldsymbol{w}_k} = -\frac{1}{B} \sum_{i=1}^{B} y_{i,k} \boldsymbol{z}_i + \frac{1}{B} \sum_{i=1}^{B} p(k|\boldsymbol{z}_i) \boldsymbol{z}_i,$$

$$\frac{\partial \mathcal{L}_{\text{CE}}}{\partial \boldsymbol{w}_k} == -\frac{1}{B} \sum_{i=1}^{B} (y_{i,k} - p(k|\boldsymbol{z}_i)) \boldsymbol{z}_i = -\frac{1}{B} \left( \sum_{i=1}^{B} y_{i,k} \boldsymbol{z}_i - \sum_{i=1}^{B} p(k|\boldsymbol{z}_i) \boldsymbol{z}_i \right),$$

In summary, the gradients of the DIET objectives according to class prototypes and representations are defined by:

$$\mathcal{L}_{\text{CE}}(Y, Z, W, \delta) = -\frac{1}{B} \sum_{i=1}^{B} \sum_{k=1}^{N} y_{i,k}^{\delta} \log \frac{\exp(\boldsymbol{w}_k^\top \boldsymbol{z}_i)}{\sum_j \exp(\boldsymbol{w}_j^\top \boldsymbol{z}_i)}$$

$$\nabla_{\boldsymbol{z}_i} \mathcal{L}_{\text{CE}} = -\frac{1}{B} \sum_{k=1}^{N} y_{i,k} \boldsymbol{w}_k + \frac{1}{B} \sum_{k=1}^{N} p(k|\boldsymbol{z}_i) \boldsymbol{w}_k = -\frac{1}{B} \sum_{k=1}^{N} \left( y_{i,k} - p(k|\boldsymbol{z}_i) \right) \boldsymbol{w}_k.$$

$$\nabla_{\boldsymbol{w}_k} \mathcal{L}_{\text{CE}} = -\frac{1}{B} \sum_{i=1}^{B} y_{i,k} \boldsymbol{z}_i + \frac{1}{B} \sum_{i=1}^{B} p(k|\boldsymbol{z}_i) \boldsymbol{z}_i = -\frac{1}{B} \sum_{i=1}^{B} (y_{i,k} - p(k|\boldsymbol{z}_i)) \boldsymbol{z}_i$$

## B  Extended Related Works

Despite DIET's simplicity, we could not find an existing method that considered it perhaps due to the common belief that dealing with hundreds of thousands of classes ($N$ in Fig. 6, the training set size) would not produce successful training. As such, the closest method to ours is *Exemplar CNN* Alexey et al. (2015) which extracts a few patches from a given image dataset, and treats each of them as their own class; this way the number of classes is the number of extracted patches, which is made independent from $N$. A more recent method, *Instance Discrimination* Wu et al. (2018) extends this by introducing inter-sample discrimination. However, they do so

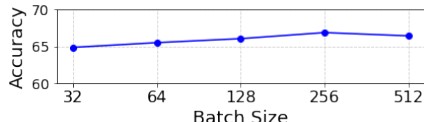

Figure 8: **Linear Probe Performance** vs batch size of S-DIET on CIFAR-100 with ResNet-18. Performance is consistent, even for small batch sizes.

using a non-parametric softmax, *i.e.*, by defining a learnable bank of centroids to cluster training samples; for successful training those centroids must be regularized to prevent representation collapse. Lastly, methods such as *Noise as Targets* Bojanowski & Joulin (2017) and DeepCluster Caron et al. (2018) are quite far from DIET as (i) they perform clustering and use the datum's cluster as its class, *i.e.*, greatly reducing the dependency on $N$; and (ii) they perform clustering in the output space of the model $f_{\boldsymbol{\theta}}$ being learned which brings multiple collapsed solutions that force those methods to employ complicated mechanisms to ensure training to learn non-trivial representations. We note that while the added complexity enables those methods to scale to large datasets, it also greatly increases the performance sensitivity to the training hyper-parameters.

### B.1 The Effect of Projection Head in DIET

This section discusses how the results from Xue et al. (2024) can be applied to analyze the benefits of using a projection head with DIET.

Given an $l$ layer linear model $f(\boldsymbol{x}) = \boldsymbol{W}_l \dots \boldsymbol{W}_1 \boldsymbol{x}$, Xue et al. (2024) all show that when training with gradient flow with any loss function and initialization $\boldsymbol{W}_i(0)\boldsymbol{W}_i(0)^\top = \boldsymbol{W}_{i+1}(0)^\top \boldsymbol{W}_{i+1}(0)$, then

$$\boldsymbol{W}_i(t)\boldsymbol{W}_i(t)^\top = \boldsymbol{W}_{i+1}(t)^\top \boldsymbol{W}_{i+1}(t) \tag{17}$$

holds at all training times $t$. We note that if weight decay is used, then Equation 17 holds for all $1 \le i \le l-1$ regardless of initialization when taking $t \to \infty$.

Now Equation 17 implies that the singular values of each layer are equal, and hence weighting of features by the model change exponentially as we go deeper into the model. As a result, intermediate layers learn more balanced and less specialized representations.

The setup from § 4 can easily be translated to a two layer model $l = 2$ with $\boldsymbol{W}_1 = \boldsymbol{W}$ the weight matrix of the linear model and $\boldsymbol{W}_2 = \boldsymbol{W}_H$ the classifier head. This can also be generalized to the case where we explicitly include a projection head along with the classifier head. Specifically, given a $j$ layer linear backbone model and a $k$ layer projection head, this equates to a setup with $l = j + k + 1$ where the linear model is represented by $\boldsymbol{W}_j \dots \boldsymbol{W}_1$, the projection head is represented by $\boldsymbol{W}_{j+k} \dots \boldsymbol{W}_{j+1}$, and the classifier head is represented by $\boldsymbol{W}_{j+k+1}$.

## C Additional Experimental Details: DIET

### C.1 DIET Pseudocode and setup

### C.2 Training dynamics

To understand feature learning in DIET, we compare its learning dynamics to other SSL methods, which exhibit step-wise learning dynamics (Zimmermann et al., 2021b; Rusak et al., 2024; von Kügelgen et al., 2021; Reizinger et al., 2024) , i.e., with small-scale initialization, the eigenvalues of the learned representations evolve in discrete steps rather than continuously (Simon et al., 2023). We observe the same for DIET. In Fig. 11 and Fig. 12, we show that training a ResNet18 on CIFAR100 and TinyImageNet using DIET leads to a step-wise increase of the embedding's singular values, similarly to other SSL methods. Additionally, we observe that the range of the singular values drops substantially for DIET, much more than for SimCLR and VicReg methods. DIET representations are thus high-rank embeddings compared to other SSL methods.

**Algorithm 1** DIET's algorithm and dataset loader.

```
# take any preferred DNN e.g. resnet50
# see Alg. 2 for other examples
f = torchvision.models.resnet50() # f_θ

# f comes with a classifier so we remove it
K = f.fc.in_features
f.fc = nn.Identity()

# define DIET's linear classifier and XEnt
W = nn.Linear(K, N, bias=False) # W_H in Equation (1)
XEnt = nn.CrossEntropyLoss(label_smoothing=0.8)

# define dataset and train (Fig. 6)
train_dataset = DatasetWithIndices(train_dataset)
train_loader = DataLoader(train_dataset, ...)

for x, n in train_loader:
    loss = XEnt(W(f(x)), n) # Equation (1)
    # backprop/optimizer/scheduler
```

```
from torch.utils.data import Dataset,
    DataLoader
from torchvision.datasets import CIFAR100

class DatasetWithIndices(Dataset):
    def __init__(self, dataset):
        self.dataset = dataset
    def __getitem__(self, n):
        # disregard the labels
        x, _ = self.dataset[n]
        return x, n
    def __len__(self):
        return len(self.dataset)

# example with CIFAR100
C100 = CIFAR100(root)
C100_w_ind = DatasetWithIndices(C100)
```

---

**DIET's experimental setup:**
- Official Torchvision architectures (no changes in init./arch.), only swapping the classification layer with DIET's one (right of Fig. 6), no projector DNN
- Same DA pipeline ($\mathcal{T}$ in Fig. 6) across datasets/architectures with batch size of 256 to fit on 1 GPU
- AdamW optimizer with linear warmup (10 epochs) and cosine annealing learning rate schedule, XEnt loss (right of Fig. 6) with *label smoothing of* 0.8
- *Learning rate/weight-decay* of 0.001/0.05 for non transformer architectures and 0.0002/0.01 for transformers

Figure 9: In underlined are the design choices directly ported from standard supervised learning (not cross-validated for DIET), in *italic* are the design choices cross-validated for DIET but held constant across this study unless specified otherwise. Batch-size sensitivity analysis is reported in Tab. 16 and Fig. 15 showing that performances do not vary when taking values from 32 to 4096. XEnt's label smoothing parameter plays a role into DIET's convergence speed, and is cross-validated in Fig. 14 and Tab. 16; we also report DA ablation in Fig. 15 and Tab. 16.

Table 16: Ablation studies indicate that **DIET benefits from longer training and stronger data augmentation while being robust to architecture and batch-size changes**. We report top1 test accuracy on CIFAR100 with varying training epochs (**top left**), on TinyImagenet with varying DA pipelines (Alg. 3), and on TinyImagenet with 3k training epochs and with varying batch-size (**bottom**) with learning rate $0.001\frac{\text{bs}}{256}$; additional comparisons on MedMNIST Tab. 17.

| Epochs | 50 | 100 | 200 | 500 | 1000 | 5000 | 10000 |
|---|---|---|---|---|---|---|---|
| resnet18 | 33.46 | 42.94 | 48.24 | 54.54 | 58.81 | 62.63 | 63.29 |
| resnet50 | 37.71 | 47.86 | 54.04 | 60.23 | 64.24 | 69.51 | 69.91 |
| resnet101 | 34.03 | 46.59 | 54.3 | 60.8 | 64.71 | 70.56 | 71.39 |

| DA strength | 1 | 2 | 3 |
|---|---|---|---|
| resnet18 | 31.48 | 43.62 | 43.88 |
| resnet34 | 32.93 | 45.60 | 45.75 |
| resnet50 | 40.24 | 48.80 | 50.81 |
| resnet101 | 40.07 | 49.74 | 50.76 |

| batch-size | 8 | 16 | 32 | 64 | 128 | 256 | 512 | 1024 |
|---|---|---|---|---|---|---|---|---|
| resnet18 | 32.9 | 37.9 | 42.7 | 43.4 | 43.3 | 43.7 | 43.7 | 42.6 |

## C.3 Impact of Training Time and Label Smoothing

In Figure 14 we show the performance of DIET on CIFAR100 across three label smoothing settings. We find higher values of label smoothing speed up convergence, although in this setting all cases greatly benefit from longer training schedules; final linear probe performances are reported in Tab. 16.

## C.4 Impact of Mini-Batch Size

We show in 15 ablations for TinyImagenet using DIET. In addition we show DIET's robustness to batch size by conducting an additional ablation by varying the batch size for the Derma MedMNIST dataset with batch sizes as low as 8. As shown in Table 17, we see DIET performs well even with very small batch sizes.

---

**Algorithm 2** Get the output dimension and remove the linear classifier from a given torchvision model (Pytorch used for illustration).

```python
model = torchvision.models.__dict__[architecture]()

# CIFAR procedure to adjust to the lower image resolution
if is_cifar and "resnet" in architecture:
    model.conv1 = torch.nn.Conv2d(3, 64, kernel_size=3, stride=1, padding=2, bias=False)
    model.maxpool = torch.nn.Identity()

# for each architecture, remove the classifier and get the output dim. (K)
if "alexnet" in architecture:
    K = model.classifier[6].in_features
    model.classifier[6] = torch.nn.Identity()
elif "convnext" in architecture:
    K = model.classifier[2].in_features
    model.classifier[2] = torch.nn.Identity()
elif "convnext" in architecture:
    K = model.classifier[2].in_features
    model.classifier[2] = torch.nn.Identity()
elif "resnet" in architecture or "resnext" in architecture or "regnet" in architecture:
    K = model.fc.in_features
    model.fc = torch.nn.Identity()
elif "densenet" in architecture:
    K = model.classifier.in_features
    model.classifier = torch.nn.Identity()
elif "mobile" in architecture:
    K = model.classifier[-1].in_features
    model.classifier[-1] = torch.nn.Identity()
elif "vit" in architecture:
    K = model.heads.head.in_features
    model.heads.head = torch.nn.Identity()
elif "swin" in architecture:
    K = model.head.in_features
    model.head = torch.nn.Identity()
```

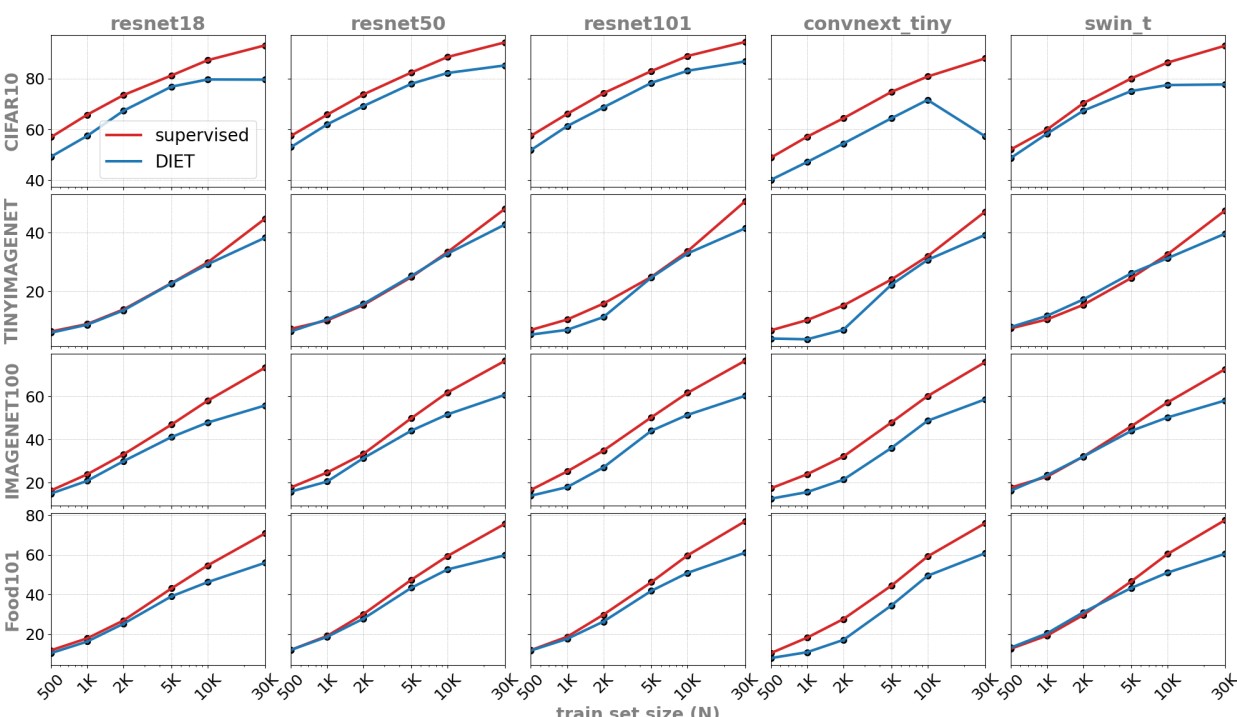

Figure 10: Reprise of Appx. C.2 on additional datasets depicting how DIET is able to compete with supervised learning for in-distribution generalization in very small dataset regime.

**Batch-size does not impact DIET's performance.** One important question when it comes to training a method with low resources is the ability to employ (very) small batch sizes. This is in fact one reason hindering

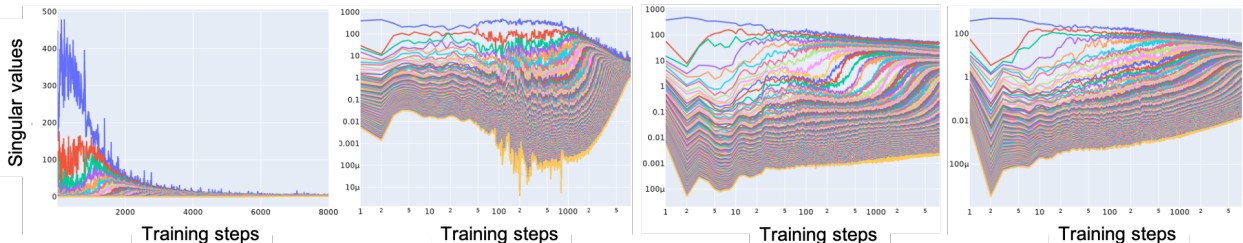

Figure 11: **Dynamics of SSL:** (left) Step dynamics similar to (Simon et al., 2023), we find that the embeddings' singular values increase in a sequential and step-wise fashion. (right) Top 200 singular values across the first 2000 training steps for DIET (left), SimCLR (middle), VicReg (right) on TinyImageNet. We find that for DIET the range of values taken by the singular values drops during training.

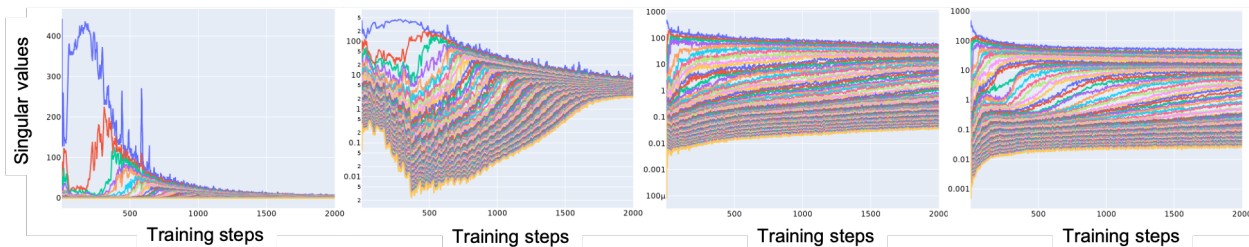

Figure 12: **Dynamics of SSL:** (left) Step dynamics similar to (Simon et al., 2023), we find that the embeddings' singular values increase in a sequential and step-wise fashion. (right) Top 200 singular values across the first 2000 training steps for DIET (left), SimCLR (middle), VicReg (right) on CIFAR100. We find that for DIET the range of values taken by the singular values drops during training.

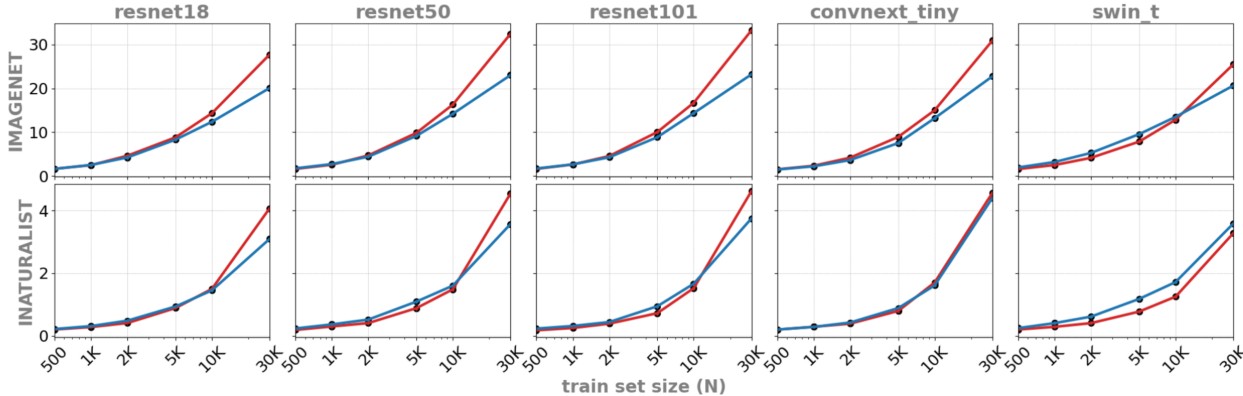

Figure 13: **DIET matches supervised learning on datasets with only a few samples per class.** Depiction of DIET's downstream performances (blue) against supervised learning (red) controlling training set size (**x-axis**); evaluation is performed over the original full evaluation set. DIET is able to learn highly competitive representations when the dataset is small with only a few samples per classes. See Fig. 10 for additional datasets.

the deployment of SSL methods which require quite large batch sizes to work (256 is a strict minimum in most cases). Therefore, we perform a small sensitivity analysis in Tab. 16 where we vary the batch size from 8 to 2048 without any hyper-parameter tuning other than the standard learning rate scaling used in supervised learning: $lr = 0.001 \frac{\text{bs}}{256}$. We observe small fluctuations of performances (due to a sub-optimal learning rate) but no significant drop in performance, even for batch size of 32. When going to 16 and 8, we observe slightly lower performances, likely due to batch-normalization Ioffe & Szegedy (2015) which is known to behave erratically below a batch size of 32 Ioffe (2017).

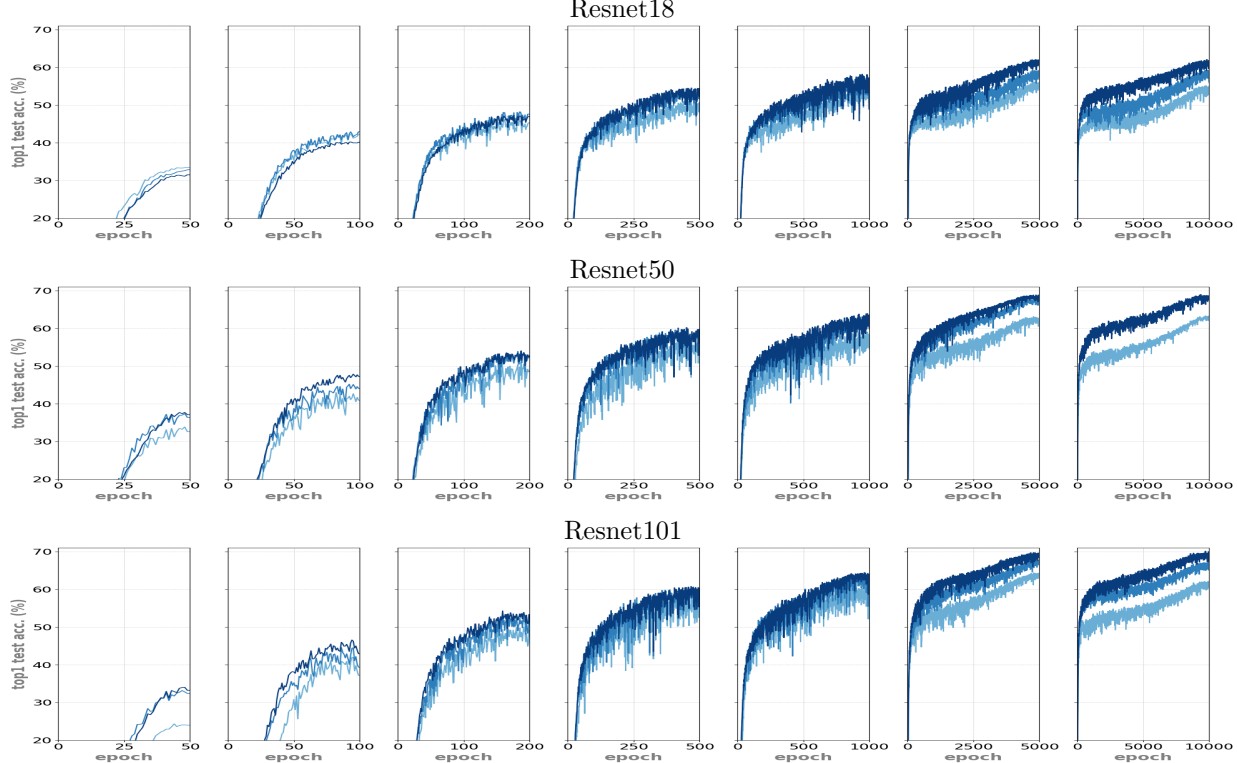

Figure 14: Depiction of the evolution of linear top1 accuracy throughout epochs on CIFAR100 with three Resnet variants and three label smoothing parameters represented by the different **shades of blue** going from light to dark shades with values of 0.1, 0.4, and 0.8 respectively.

| Batch Size | 8 | 32 | 64 | 128 | 512 |
|---|---|---|---|---|---|
| DIET | 71.87 | 72.52 | 73.07 | 74.36 | 71.02 |
| MoCov2 | 66.88 | 64.64 | 66.73 | 66.88 | 61.40 |
| SimCLR | 63.14 | 66.43 | 66.83 | 66.88 | 66.83 |
| VICReg | 65.84 | 60.45 | 64.79 | 66.78 | 66.88 |

Table 17: Reprise of Tab. 16: DIET's performance across varying batch sizes on the Derma MedMNIST dataset with all other hyperparameter fixed demonstrating the stability of DIET do that hyper-parameter and across training iterations. All models are trained for 500 epochs.

## C.5   Impact of Data-Augmentation

To further study the effect of data augmentation in DIET we study varying data augmentation strengths for TinyImageNet in Fig. 15. We also examine the effect of weaker data augmetnations for smaller medical images using PathMNIST in Tab. 18.

**Data-Augmentation sensitivity is similar to SSL.** When using DA, DIET is able to perform on par with highly engineered state-of-the-art methods. Yet, knowing which DA to employ is not trivial, e.g., many data modalities have no obvious DA. One natural question is, thus, concerning the sensitivity of DIET's performance to the employed DA. To that end, we propose three DA regimes, one only consistent of random crops and horizontal flips (**strength:1**), which could be considered minimal in computer vision, one which adds color jittering and random grayscale (**strength:2**), and one last which further adds Gaussian blur and random erasing Zhong et al. (2020) (**strength:3**); the exact parameters for those transformations are given in Alg. 3. We observe on TinyImagenet and with a Resnet34 the following performances $32.93\pm 0.6$, $45.60\pm 0.2$, and $45.75\pm 0.1$ respectively over 5 independent runs, details and additional architectures provided in Fig. 15

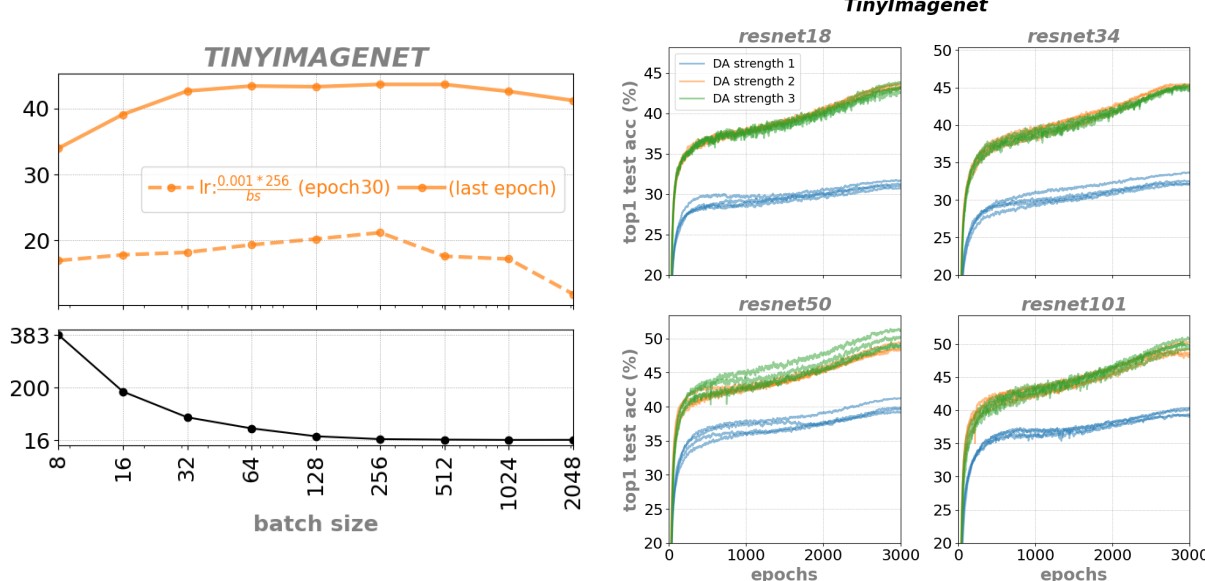

Figure 15: **Left:**TinyImagenet with fixed number of epochs and a single learning rate which is adjusted for each case using the LARS rule therefore per batch-size learning cross-validation can only improve performances, see Tab. 16, , the per-epoch time includes training, testing, and checkpointing. **Right:** TinyImagenet, see Tab. 16 for table of results, and the specific DAs can be found in Alg. 3.

---

**Algorithm 3** Custom dataset to obtain the indices ($n$) in addition to inputs $\boldsymbol{x}_n$ and (optionally) the labels $y_n$ to obtain `train_loader` used in Appx. C.1 (Pytorch used for illustration).

```
transforms = [
    RandomResizedCropRGBImageDecoder((size, size)),
    RandomHorizontalFlip(),
]
if strength > 1:
    transforms.append(
        T.RandomApply(
            torch.nn.ModuleList([T.ColorJitter(0.4, 0.4, 0.4, 0.2)]), p=0.3
        )
    )
    transforms.append(T.RandomGrayscale(0.2))
if strength > 2:
    transforms.append(
        T.RandomApply(
            torch.nn.ModuleList([T.GaussianBlur((3, 3), (1.0, 2.0))]), p=0.2
        )
    )
    transforms.append(T.RandomErasing(0.25))
```

and Tab. 16 in the Appendix. We thus observe that while DIET greatly benefit from richer DA (strength:1 $\mapsto$ 2), it however does not require heavier transformation such as random erasing.

## C.6 Impact of Label Smoothing

**Label smoothing helps.** One important difference in training behavior between supervised learning and SSL is in the number of epochs required to see the quality of the representation plateau. Due to the different loss used in DIET, one might wonder about the differences in training behavior. We observe that DIET takes more epochs than SSL until the loss converges. However, by using large values of label smoothing, *e.g.*, 0.8, it is possible to obtain faster convergence. We provide a sensitivity analysis in Fig. 14 and Tab. 16 in the Appendix. In fact, one should recall that within a single epoch, only one of each datum/class is observed, making the convergence speed of the classifier's $\boldsymbol{W}_H$ matrix the main limitation; we aim to explore improved training strategies in the future as discussed in § 8.

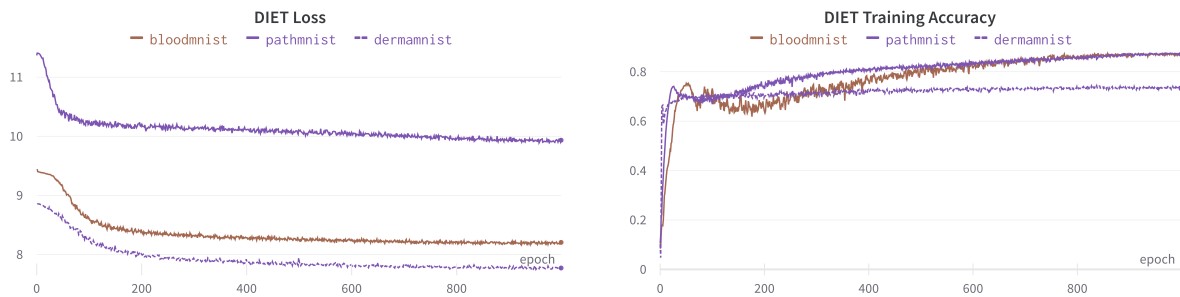

Figure 16: DIET MedMNIST training loss curves for the DIET criterion (left) and training accuracy (right) with a ResNet18 backbone.

### C.7 DIET compared to supervised learning

**DIET matches supervised learning on datasets with only a few samples per class.** In Appx. C.2 we directly compare DIET with supervised learning on a variety of models and datasets but with controlled training size. We clearly observe that for small dataset, *i.e.*, for which we only use a small part of the original training set (less than 30 images per class), DIET's learned representation is as efficient as the supervised one for the in-distribution classification downstream task.

**DIET works with scattering network architectures** As an additional test, scattering networks (Oyallon et al., 2018; Gauthier et al., 2022) hard-code part of the model parameters to be wavelet filter-banks. That specification naturally makes such scattering networks very competitive for small data regimes since the number of degrees of freedom is reduced. We therefore performed two additional experiments: Training a hybrid scattering network in a supervised setting Training a hybrid scattering network with DIET and then learning a linear probe on top (keeping the hybrid scattering frozen) We perform both cases above on the full CIFAR10 training set and on a reduced training set of 5000 (10% of the training data) samples. Supervised training of the scattering network results in 72.1% (58.2%) test set accuracy, whereas unsupervised DIET pretraining followed by a linear probe results in 77.64% (62.8%) for the same architecture. From that experiment we obtain two novel insights. First, DIET works out-of-the-box on DNs such as the hybrid scattering network, with a reduced number of parameters. Second, even in that regime, DIET provides strong performances.

### C.8 Additional Results for MedMNIST

In Figure 16 we show training curves for DIET with a ResNet18 architecture. We perform additional experiments with DIET using a vision transformer architecture (ViT-Small with patch size 4) based on the architecture from `https://github.com/lucidrains/vit-pytorch/blob/main/vit_pytorch/vit_for_small_dataset.py`. We find DIET achieves good performance on the same MedMNIST datasets with this ViT architecture without additional hyperparameter tuning as shown in Table 19 and in comparison to all three baseline SSL methods in Table 18.

We find evidence of the default augmentations for PathMNIST being too aggressive and confirm DIET's performance improves with the use of weaker augmentations in Table 18. Surprisingly, we find DIET performs quite well with no augmentations at all, a setting in which most standard SSL methods would be impossible to train.

## D Additional Experimental Details: S-DIET

### D.1 S-DIET Pseudocode

```
1  """
2  Uppercase variables stored on disk
3  Lowercase variables stored in memory
4
5  X: train data
6  H: classifier head
7  M: first moment for classifier head
```

|  | bloodmnist | | dermamnist | | pathmnist | |
|---|---|---|---|---|---|---|
|  | train | test | train | test | train | test |
| **DIET** | 77.65 | 81.85 | 71.03 | 68.88 | 56.37 | 21.27 |
| **SimCLR** | 82.48 | 79.45 | 69.13 | 32.37 | 69.45 | 21.80 |
| **VICReg** | 86.71 | 81.03 | 69.89 | 46.33 | 82.94 | 12.76 |
| **MoCov2** | 62.76 | 51.01 | 66.78 | 63.39 | 72.9 | 41.75 |

| DIET | PathMNIST | |
|---|---|---|
| Augmentation | train | test |
| Default | 56.37 | 21.27 |
| Weak | 44.90 | 48.95 |
| None | 44.65 | 45.67 |

Table 18: **Top:**DIET performance across the three MedMNIST datasets using a transformer (ViT-S) architecture with patch size 4 in comparison to standard SSL baselines with the same ViT architecture. **Bottom:**Comparing DIET's performance across data augmentations for PathMNIST using a transformer (ViT-S) architecture with patch size 4. Weak augmentation corresponds to only random resized cropping and horizontal flipping.

Table 19: DIET performance across the three MedMNIST datasets using a transformer (ViT-S) architecture with patch size 4. In the first row we show the performance of a baseline SimCLR model with the default ResNet18 encoder for comparison.

| dataset | bloodmnist | | dermamnist | | pathmnist | |
|---|---|---|---|---|---|---|
|  | train | test | train | test | train | test |
| DIET | 77.65 | 81.85 | 71.03 | 68.88 | 56.37 | 21.27 |

Table 20: **DIET is competitive and works out-of-the-box across architectures.** We keep the settings of Fig. 9. Benchmarks from 1:Dubois et al. (2022), 2 :Ozsoy et al. (2022)

**TinyImagenet**

| Resnet18 | |
|---|---|
| SimSiam | 44.54 [‡] |
| DIET | 45.07 |
| SimCLR | 46.21[‡] |
| BYOL | 47.23[‡] |
| MoCo | 47.98 [‡] |
| SimCLR | 48.70 [1] |
| DINO | 49.20 [1] |

| Resnet50 | |
|---|---|
| SimSiam | 46.76 [2] |
| SimCLR | 48.12 [2] |
| DIET | 51.66 |

| DIET | | | |
|---|---|---|---|
| resnet18 | 45.07 | resnet50 | 51.66 |
| resnet34 | 47.04 | convnext_tiny | 50.88 |
| resnet101 | 51.86 | convnext_small | 50.05 |
| wide_resnet50_2 | 50.03 | MLPMixer | 39.32 |
| resnext50_32x4d | 52.45 | swin_t | 50.80 |
| densenet121 | 49.38 | vit_b_16 | 48.38 |

**Imagenet-100 (IN100)**

| Resnet18 | |
|---|---|
| SimMoCo | 58.20* |
| MocoV2 | 60.52* |
| SimCo | 61.28 * |
| W-MSE2 | 69.06 [2] |
| ReSSL | 74.02• |
| DINO | 74.16• |
| MoCoV2 | 76.48• |
| BYOL | 76.60• |
| SimCLR | 77.04[2] |
| SimCLR | 78.72[2] |
| MocoV2 | 79.28[2] |
| VICReg | 79.40[2] |
| BarlowTwins | 80.38[2] |

| Resnet50 | |
|---|---|
| MoCo+Hyper. | 75.60 * |
| MoCo+DCL | 76.80 * |
| MoCoV2 + Hyper. | 77.70 * |
| BYOL | 78.76 [2] |
| MoCoV2 + DCL | 80.50 * |
| SimCLR | 80.70 * |
| SimSiam | 81.60[2] |
| SimCLR + DCL | 83.10 * |

| DIET | | | |
|---|---|---|---|
| resnet18 | 64.31 | resnet50 | 73.50 |
| wide_resnet50_2 | 71.92 | convnext_small | 71.06 |
| resnext50_32x4d | 73.07 | MLPMixer | 56.46 |
| densenet121 | 67.46 | swin_t | 67.02 |
| convnext_tiny | 69.77 | vit_b_16 | 62.63 |

```
8   V: second moment for classifier head
9
10  indices: indices for the current batch
11  """
12  def train_step(X, H, M, V, indices, model, criterion, optimizer):
13      # Load data, head weights, and head optimizer state into memory
14      inputs, head, optimizer_m, optimizer_v = X[indices], H[indices], M[indices], V[indices]
15      labels = [0, 1, ..., len(indices)−1]
16
```

```
17      # Forward and backward pass
18      outputs = head(model(inputs))
19      loss = criterion(outputs, labels)
20      optimizer.zero_grad()
21      loss.backward()
22      optimizer.step()
23      head, m, v = perform_multistep_adamw_head_update(head, m, v)
24
25      # Save head weights and head optimizer state
26      # Done asynchronously
27      H[indices], M[indices], V[indices] = head, m, v
28
29
30  def perform_multistep_adamw_head_update(head, m, v):
31      g = head.grad
32
33      # first step
34      head = (1 - lr * weight_decay) * head
35      m = beta1 * m + (1 - beta1) * g
36      v = beta2 * v + (1 - beta2) * g * g
37      head = head - lr * m / (sqrt(v) + eps)
38
39      # all other steps
40      mu = beta1 / sqrt(beta2)
41      alpha1 = (1 - lr * weight_decay) ** (t - 1)
42      alpha2 = (alpha1 * lr * mu - lr * (mu ** t)) / (1 - lr * weight_decay - mu)
43
44      head = alpha1 * head - alpha2 * m / (sqrt(v) + eps)
45      m = (beta1 ** (t - 1)) * m
46      v = (beta2 ** (t - 1)) * v
```

Listing 1: Pseudocode for a S-DIET training step

## D.2 Handling Stateful Optimizers in s-DIET

**AdamW.** Stateful optimizers provide a further opportunity. Considering the AdamW update rules (Loshchilov & Hutter, 2019):

$$\boldsymbol{m}_t \leftarrow \beta_1 \boldsymbol{m}_{t-1} + (1 - \beta_1)\boldsymbol{g}_t, \; \boldsymbol{v}_t \leftarrow \beta_2 \boldsymbol{v}_{t-1} + (1 - \beta_2)\boldsymbol{g}_t^2, \; \boldsymbol{\theta}_t \leftarrow (1 - \eta\lambda)\boldsymbol{\theta}_{t-1} - \eta\psi_t \frac{\boldsymbol{m}_t}{\sqrt{\boldsymbol{v}_t} + \epsilon}$$

where $\psi_t = \frac{\sqrt{1 - \beta_2^t}}{1 - \beta_1^t}$. For simplicity, we replace $\psi_t = 1$. For default settings $\beta_1 = 0.9, \beta_2 = 0.999$, this can be interpreted as learning rate warmup. As the optimizer may update the weights even if their gradient at the current step $\boldsymbol{g}_t$ is zero. Thus it would require loading the entire $\boldsymbol{W}_H$ even with batch cross entropy. The $i$-th row of $\boldsymbol{W}_H$ is used only with sample $i$ in the batch; otherwise the corresponding batch cross entropy gradient is zero. Since not each batch contains sample $i$, when it does, we perform $t$ optimizer steps on the $i$-th row of $\boldsymbol{W}_H$. As we do not exactly know $t$, i.e. is the number of steps until the sample $i$ is again in the batch, we approximate $t = \frac{N}{B}$. This yields what we call the *multistep update formula* for AdamW. Assuming we have dropped the $\psi_t$ term and $\epsilon$ is negligible, if $\boldsymbol{g}_t = 0$ for all $t$, the above update formulas for AdamW become an inhomogeneous linear recurrence relation which has a closed form solution:

$$\boldsymbol{m}_t = \beta_1^t \boldsymbol{m}_0, \qquad \boldsymbol{v}_t = \beta_2^t \boldsymbol{v}_0, \qquad \boldsymbol{\theta}_t = (1 - \eta\lambda)^t \boldsymbol{\theta}_0 + \frac{(1 - \eta\lambda)^t \eta\mu - \eta\mu^{t+1}}{1 - \eta\lambda - \mu} \frac{\boldsymbol{m}_0}{\sqrt{\boldsymbol{v}_0} + \epsilon}. \tag{18}$$

where $\mu$ denotes the ratio $\frac{\beta_1}{\sqrt{\beta_2}}$. In summary, at each step we only update the weights and optimizer state of rows of $\boldsymbol{W}_H$ that were selected for batch cross entropy at that step. We perform the update by first taking one step with $\boldsymbol{g}_t$ as the calculated gradient, and then apply the multistep update given by Eq. 18 for $t = \frac{N}{b} - 1$.

**SGD with Momentum.** Recall the update rule of SGD with learning rate $\eta$, momentum $\mu$, dampening $\tau$, weight decay $\lambda$:

$$\boldsymbol{m}_t \longleftarrow \mu\boldsymbol{m}_{t-1} + \tau\boldsymbol{g}_t, \qquad \boldsymbol{\theta}_t \longleftarrow (1 - \eta\lambda)\boldsymbol{\theta}_{t-1} - \eta\boldsymbol{m}_t \tag{19}$$

If $\boldsymbol{g}_t = 0$ for all $t$, the above update formulas for SGD with momentum become an inhomogeneous linear recurrence relation which has an exact solution:

$$\boldsymbol{m}_t = \mu^t \boldsymbol{m}_0, \qquad \boldsymbol{\theta}_t = (1 - \eta\lambda)^t \boldsymbol{\theta}_0 + \frac{(1 - \eta\lambda)^t \eta\mu - \eta\mu^{t+1}}{1 - \eta\lambda - \mu} \boldsymbol{m}_0 \tag{20}$$

### D.3 SSL Methods

Pretrained models for SSL methods are obtained using the solo-learn library (da Costa et al., 2022). We use the batch size and augmentations as specified in the previous section, and change the precision to 32-bit for consistency. All other hyperparameters are left unchanged.

### D.4 Toy Dataset

We instantiate the scenario from Section 4.2 with a more realistic training setup:

- we make the classifier head $\boldsymbol{W}_H$ trainable from random initialization.

- instead of taking the expectation over all augmentations, we sample a single random augmentation of the input at each step.

- We also choose $\mathcal{G}_1, \mathcal{G}_2$ to follow normal distributions (this eliminates the requirement of bounded feature noise).

We set $C = 4, d = 16, m = 4, n = 32, \sigma_1 = 0.01, \sigma_2 = 0.1, \phi = 0.004$. We train for 5000 steps using the Adam optimizer with learning rate 0.1 and cosine learning rate schedule. We reset the state of the Adam optimizer after the first step to eliminate the effect of gradient blowup from normalizing zero vectors, see Appx. A.1.5 for details.

### D.5 Synthetic Dataset

For the synthetic dataset described in Section 7.1, we modify the first convolutional layer of the ResNet model to take 4 input channels instead of 3. For MNIST augmentations, we replace random horizontal flip and random grayscale with gaussian blur. We also modify the random cropping to keep at least 0.75 of the area of the original image. We train for 500 epochs. All other hyperparameters are set as described above.

### D.6 Equivalence of DIET and Spectral Contrastive Learning for Ideal Encoders

A common line of study is to analyze the minima of loss functions assuming an ideal encoder, namely one that can realize any output configuration, since such analysis depends only on the loss function and not how exactly the encoder is parameterized. Thm. 5 directly covers this case, as an ideal encoder can be parameterized as a fixed feature map $\phi$ which maps the inputs to a linearly independent set and then applying a linear encoder. In this case, the global minima of the spectral contrastive loss is any encoder for which

1. All augmentations of an example are collapsed to a single unit vector

2. Embeddings of different examples are orthogonal.

We note that this recovers a result from Johnson et al. (2023), which, after applying the rescaling discussed in Appx. A.1.2 states that the global minima of spectra is achieved when

$$K_{\boldsymbol{\theta}}(\boldsymbol{x}_1, \boldsymbol{x}_2) = f(\boldsymbol{x}_1)^\top f(\boldsymbol{x}_2) = \delta_{y_1, y_2}$$

Meanwhile, minimizing the MSE DIET loss requires that the outputs of the classifier be exactly equal to the specified targets. Assuming that the classifier head is an isometry, we exactly recover the previous two conditions on the learned embeddings.

### D.7 Comparison Between DIET and CL

In Figure 17, we compare t-SNE visualizations (van der Maaten & Hinton, 2008) of test embeddings produced by s-DIET and SimCLR (Chen et al., 2020) on CIFAR-10 with ResNet-50. We observe that the high level structure of the embeddings is remarkably similar for both methods.

### D.8 s-DIET experiments with ViT backbone

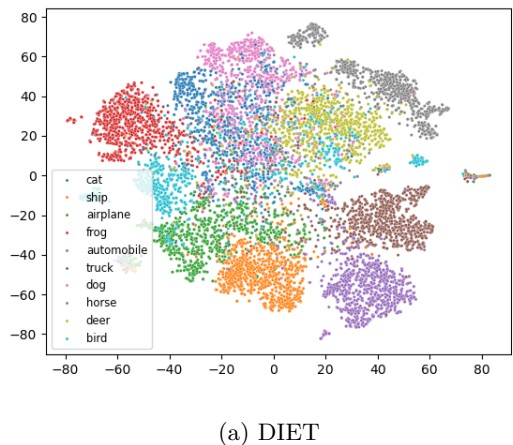

(a) DIET

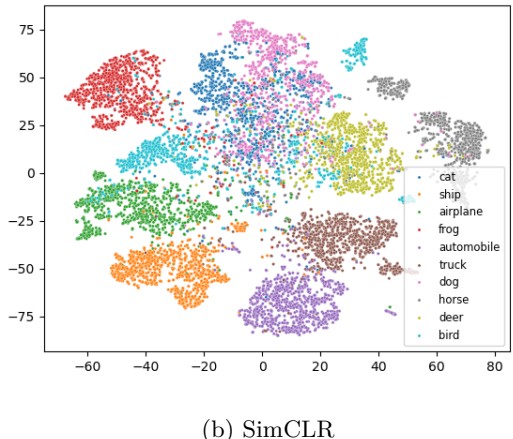

(b) SimCLR

Figure 17: TSNE of embeddings produced by DIET and SimCLR on CIFAR-10 using ResNet-50.

Table 21: **Linear Probe Accuracy** of ViT trained with DIET and s-DIET. Again s-DIET provides significant performance gains over DIET.

| Method | ImageNet-100 | TinyImageNet |
|--------|-------------|--------------|
| DIET   | 62.63       | 48.38        |
| s-DIET | **74.04**   | **55.82**    |

# E   Acronyms

**CL** Contrastive Learning

**DIET** Datum IndEx as its Target

**MSE** Mean Squared Error

**PID** parametric instance discrimination

**s-DIET** Scaled DIET

**SCL** Spectral Contrastive Learning

**SSL** self-supervised learning

