# OpenReview forum: "Occam’s Razor for SSL: Memory-Efficient Parametric Instance Discrimination"
_TMLR — Accepted by TMLR_

### Review · Reviewer_WEha · 2025-10-23

**Summary Of Contributions:**

Introduce a simple parametric instance discrimination approach (DIET) that learns data representations without relying on complex techniques employed by state-of-the-art (SOTA) methods, such as gradient stopping or negative sampling, to prevent representation collapse.

Strength:\
1-	The problem is explained well. \
2-	The paper’s motivation to reduce the complexity and resource consumption of self-supervised learning presents an interesting aspect.

Weakness:\
1-The baseline comparisons presented in the paper lack consistency and clarity. Specifically, Table 1 includes only SimCLR as a baseline for ImageNet-100 with a ResNet18 backbone. In contrast, Table 2 adds MoCo-v2 and VICReg, while Table 3 further expands the comparison to include BYOL, SimSiam, Barlow Twins, and SimCLR. Table 4 then reverts to only SimCLR and VICReg. This inconsistency creates confusion regarding the chosen baseline.\
2- SOTA approaches are typically trained for 800 to 1000 epochs, whereas DIET is trained for 5000 epochs. This raises concerns regarding the time complexity of training DIET. \
3- SOTA approaches typically demonstrate their robustness across a variety of downstream tasks. In contrast, the authors evaluate their method solely through linear evaluation. \
4-The fundamental objective of self-supervised learning (SSL) is to eliminate the reliance on manual data annotations by enabling models to learn representations directly from large-scale unlabelled datasets. Consequently, many contemporary SSL frameworks, such as DINO-v2 and DINO-v3 [1,2], emphasise scalability to increasingly extensive datasets. However, as noted on page 8, paragraph 2, the author indicates that the DIET approach becomes infeasible when applied to large datasets. This limitation appears to contradict the central aim of SSL, thereby constraining the scalability and reducing the overall significance of the proposed method.

[1] Oquab, M., Darcet, T., Moutakanni, T., Vo, H., Szafraniec, M., Khalidov, V., Fernandez, P., Haziza, D., Massa, F., El-Nouby, A., et al. (2023). Dinov2: Learning robust visual features without supervision. arXiv preprint arXiv:2304.07193.

[2] Siméoni, O., Vo, H. V., Seitzer, M., Baldassarre, F., Oquab, M., Jose, C., Khalidov, V., Szafraniec, M., Yi, S., Ramamonjisoa, M., et al. (2025). Dinov3. arXiv preprint arXiv:2508.10104.

**Audience:**

Yes

**Audience Explanation:**

The idea of reducing the complexity of SSL approaches is interesting and might inspire researchers to build on it.

**Claims And Evidence:**

No

**Claims Explanation:**

The author claims that the proposed approach can compete with SOTA without the complexity of the SOTA approaches. The author needs to conduct variant experiments across different downstream tasks to demonstrate that the proposed approaches perform competitively with SOTA methods.

**Requested Changes:**

1- It is recommended that the authors clearly define the baseline method and maintain consistency across all tables. Furthermore, top-performing approaches such as MoCo-v2 and SimSiam should be consistently included, as these methods demonstrate superior performance with the batch size of 256 used by the authors.\
2- It is recommended that the authors report the training time (in hours) or computational cost (e.g., FLOPs) to enable a fair comparison and to assess the efficiency of their proposed approach.\
3- It is recommended that the authors conduct additional downstream experiments, such as object detection and semi-supervised learning, to assess the quality and generalizability of the learned representations more comprehensively.\
4- It is recommended to explicitly provide practical scenarios in which the DIET approach is preferable over state-of-the-art SSL methods.\
5- Finally, I recommend that the author organise the ablation study into distinct sections or subsections, separate from the main experimental results, to enhance clarity and readability.

---

> ### Author Response · Authors · 2025-11-22
>
> Thank you for reviewing our work. We appreciate the detailed suggestions for improvement; we believe that the revised paper is stronger because of these revisions, which are summarized as (we detail our response below) :
>
> **Rebuttal summary:**
> - We have added comparisons on memory and compute requirements (Fig. 3., Tab. 4) of the tested SSL methods
> - We have reorganized our experimental results and ablations, to make our comparisons clear w.r.t. other SSL methods. We have unified our results tables (Tabs. 1-3), and also included MoCo-v2 and SimSiam.
> - We have clarified that the primary goal of introducing DIET was to provide a simple SSL method that performs well on small-scale datasets, offering a practical solution for practitioners working with specialized datasets (e.g., medical imaging datasets).
>
>
> **[Baselines, Results Presentation]** Thank you for the useful feedback! We reorganized the experimental section to ensure consistency across baselines:
> - Table 1,2,3 now include SimCLR, MoCov2, VICReg, SimSiam, and BYOL as baselines for experiments where models are trained from scratch or experiments using ImageNet-1k pre-trained models. Note that ImageNet-100 checkpoints across baselines are no (longer) publicly available.
> - BarlowTwins, PCP, and MoCov1 were removed from Tables 1,2,3 to improve consistency, as none of these baselines showed the strongest results across tables.
> - Table 3: We are currently actively working on adding the VICReg and MoCov2 baseline for the ImageNet-100 and TinyImageNet datasets.
> - Table 4: We are currently actively working on improving consistency on this table by adding the MoCov2, SimSiam, and BYOL baselines to this table.
>
> **Additional baselines did not change our conclusions for Table 1&3**. For Table 2, we observe that BYOL is a strong baseline on the PathMNIST dataset—the largest medical dataset in our evaluation (~90k training samples)—achieving performance that even surpasses supervised pretraining. We additionally train our sDIET method, which we show improves the DIET performance for larger datasets, and find that it outperforms all other baselines on PathMNIST, including BYOL, consistent with the results reported in Table 3.
>
> **[Computational and Memory Cost]** We now compare both memory and training time requirements. Figure 3 and Table 4 (see included below for convenience) provide a clear overview of the compromise: s-DIET is more memory efficient, but compromises on the training time.
>
> | Method | Dataset | Model | Training Time (hr)  |
> |--------|---------|--------|---------|
> | sDIET  | CIFAR   | R18    | 16.2  |
> | sDIET  | CIFAR   | R50    | 51.0  |
> | SimCLR | CIFAR   | R18    | 3.9  |
> | SimCLR | CIFAR   | R50    | 11.5  |
>
> | Method | Dataset | Model | Training Time Per Epoch (s)   |
> |--------|---------|--------|---------|
> | sDIET  | CIFAR   | R18    | 11.680  |
> | sDIET  | CIFAR   | R50    | 36.720  |
> | SimCLR | CIFAR   | R18    | 13.907  |
> | SimCLR | CIFAR   | R50    | 41.455  |
>
> **[Object Detection]** Thank you for the suggestion. Exploring additional downstream tasks, such as object detection, is certainly an interesting direction for future work. However, our contribution is to show that DIET offers a simple alternative to more complex SSL methods (even when those are pretrained on large datasets), for small datasets of interest. Extending our analysis to show that DIET provides more universal representations using object detection benchmarks would hence fall outside of the scope of this work.
>
> **[Practical Scenarios]** We have clarified that, for small datasets, s-DIET offers better downstream performance over other SSL methods—even when those baselines are pretrained on large-scale datasets such as ImageNet-1k. This directly addresses the needs of practitioners working with specialized small datasets (e.g., medical datasets) who wish to leverage SSL to improve the performance of their models. Our results show that DIET offers a simple alternative to existing SSL methods and their available checkpoints that meet these practical requirements.
>
> **[Paper Structure]** Thank you for the feedback. We have created a separate section for the ablations in the revised manuscript.
>
> Thank you again for the detailed review! We would like to highlight the contributions of our work:
> - We propose DIET, a simplified SSL pipeline, and implement it in a way that is more memory efficient than existing SSL methods.
> - We provide a theoretical justification for how our simplified method can achieve competitive performance by equating the global minima of DIET with those of existing SSL methods
> - DIET performs on par with SOTA on small datasets. Additionally, we propose s-DIET, which scales more effectively to larger datasets.
> - While SSL traditionally focuses on performance on larger datasets, we argue that the low-data regime cannot be overlooked when considering applications where data is hard to collect or requires careful curation, such as the medical domain.

---

> > ### Comment · Reviewer_WEha · 2025-11-23
> >
> > I would like to thank the authors for the updates provided in the revised manuscript and for the clarifications offered during the rebuttal phase, which have addressed most of my initial concerns. The proposed approach demonstrates promising results compared to state-of-the-art self-supervised learning (SSL) methods. However, the claim that this technique can serve as a general alternative to existing SOTA approaches appears overly broad.
> >
> > This limitation primarily stems from the scalability constraints of DIET, as well as the slow convergence in s-DIET. Furthermore, the generalisation capability of the method has not been sufficiently evaluated across a broader range of downstream tasks. In addition, the training time required is substantially higher than that of current SOTA methods.
> >
> > Given these factors, the approach seems best positioned for small-scale datasets where memory efficiency is a priority and training time is less of a constraint—for example, scenarios involving limited GPU resources. In such cases, the method offers a practical and efficient SSL solution despite its increased computational time.

---

> ### Author Response · Authors · 2025-11-24
>
> Thank you for your response to our rebuttal!
>
> We agree that DIET and s-DIET provide a clear advantage for small-scale datasets, as emphasized in our contributions and results. This directly addresses questions from practitioners aiming to apply SSL to their own, often limited, datasets. To make this even more explicit, we have refined both the introduction and conclusion, as reflected in the updated manuscript. We believe our contribution is now more clearly articulated, and appreciate your constructive feedback.
>
> We are glad that the changes introduced during the rebuttal have addressed your initial concerns, and we hope that these additional clarifications, together with the strong results on small-scale datasets, convinced you that our _"claims are supported by accurate, convincing, and clearly presented evidence"_.

---

### Review · Reviewer_ADzt · 2025-11-06

**Summary Of Contributions:**

This work presents a representation learning approach based on a self-supervised instance discrimination method, which replaces the need of labeled training data by an instance classification escenario. This approach also does not require having negative training instances. The proposed idea seems to be efficient, to obtain competitive performance on small datasets, and to be robust against changes in values of training hyperparameters.

**Audience:**

Yes

**Audience Explanation:**

- The idea is novel and interesting.
- The proposed model does not require labels nor instances from the negative class.
- The document is well organized and written.
- The introduction section clearly identifies the research gap.

Mainly, the proposed approach presents the opportunity for rich latent representations, that could be later refined and exploited for mainstream tasks.

**Broader Impact Concerns:**

No concern about ethics.

**Claims And Evidence:**

Yes

**Claims Explanation:**

The explanation of the model confirms that it does not require labels o negative instances.

**Requested Changes:**

- The authors present their analysis using a MSE function, instead of using categorical cross entropy (CCE), as it is common for multi-class classification problems. The authors justify this decision stating that MSE allows to perform a tractable analysis. However, CCE is also tractable. This claim must be justified better.
- It is unclear why there is an assumption of orthonormality mentioned in section 4.1? Is it needed? What is its impact?
- The setup in section 6 needs to indicate the number of neural units in the hidden layers.
- The setup in section 6, talks about the same data augmentation approach. Please, indicate the number of total random samples that are generated.
- In page 6, "can introduce risks such data poisoning" --> "can introduce risks such as data poisoning".
- Although the idea is interesting, and elegant, the numeric results indicate that the method provides performance either below or comparable with previous methods. This makes it difficult to assess the actual contribution of this work. Is the model smaller in terms of parameters, or does it train much faster with respect to other models?
- Indicate which metrics are reported in all tables. I guess it is accuracy. also indicate whether they correspond to the train, validation, or test set.
- The purpose of the experiment that combines the MNIST and CIFAR-10 datasets is unclear.
- Avoid inline tables. All inserted objects must be numbered and cited though the main document.

---

> ### Author Response · Authors · 2025-11-22
>
> Thank you very much for the detailed feedback, which improved our submission. To address your questions, we have made the following changes (we detail our responses below)
>
> Rebuttal summary
> - **MSE vs CCE**: we have extended our theoretical result to the cross entropy case
> - We have added **more experimental details** (hidden units, data augmentation protocol), method comparisons (training time, memory size) and state the reported metrics explicitly, while also clarifying the hypothesis we are testing with each experiment (Sec. 6 and 7)
> - We have **clarified that our main contribution** was to demonstrate that a simple SSL method such as DIET can achieve the same or comparable performance, i.e., all the bells and whistles of SSL are not strictly necessary
>
> We hope to address all the comments in the revised version of the paper.
> 1. **[MSE vs CCE]** Thank you for the feedback! In the revised version, we have added Theorem 1, which derives a connection between CE loss and InfoNCE loss for unit-normalized embedding functions, complementing the existing correspondence between MSE loss and spectral contrastive loss.
> 2. Good question! The **orthonormality** assumption is a common assumption in the literature based on the idea that in a high-dimensional feature space, even two random vectors are likely to be almost orthogonal [1,2,3,4].
> 3. **[Architectures and Hidden Units] **We use ResNet-18, ResNet-50, and Vit-B/16 have embedding dimensions 512, 2048, and 768, respectively. We have clarified the setup in Sec. 6.
> 4. **[Data Augmentation]** Good question, data augmentation consists of random crops, random horizontal flips, color jittering, grayscaling, and gaussian blur inspired from popular data augmentation pipelines used in SSL method like SimCLR. A fresh augmentation is sampled and applied to an example whenever it is included in a training batch. That is, for a batch of B images, there are B augmented samples
> 5. Thanks for catching this. We have fixed it in the revised version.
> 6. **[Contribution: simple SSL performs comparably]** Our main goal by proposing DIET was to demonstrate that many practices used in SSL are not strictly necessary to achieve good performance. That is, our claim was not about achieving superiority. We realize our formulation was not clear enough. In the updated manuscript, we emphasize that our main contribution is proposing a simple SSL pipeline which requires limited hyper-parameter tuning. However, DIET (and its scaled version, s-DIET) has a memory advantage over Barlow Twins, SimSiam, SimCLR, and BYOL on CIFAR-10/100 and ImageNet-100, and s-DIET is also more memory efficient than the others on TinyImageNet, which we show in our Fig. 3. We added emphasis to this difference to more clearly distinguish our contributions.
> 7. **[Metrics] **Thank you for pointing it out. Yes, it is indeed linear probe accuracy–we added this to the captions.
> 8. **[Combined MNIST+CIFAR] **This experiment follows the practice of earlier works  [5,6] - We also added references to the paper to contextualize this experiment, as it has been widely used to study feature suppression. The synthetic dataset that combines MNIST and CIFAR-10 is designed to mimic the data generating process from Section 4.2, i.e., to simulate feature suppression when two features vary differently in the dataset. The MNIST digit represents the low noise content feature, while the CIFAR digit represents the high noise style feature. Our experiment corroborates the theory by showing that normalization allows DIET to learn both features. [Tables] Thank you for the suggestion. We labeled the tables and cite them in the manuscript in Sec. 7.
>
> [1]: Zixin Wen and Yuanzhi Li. Toward understanding the feature learning process of self-supervised contrastive learning. 2021.
>
> [2]: Difan Zou, Yuan Cao, Yuanzhi Li, and Quanquan Gu. Understanding the generalization of adam in learning neural networks with proper regularization, 2021. URL
>
> [3]: Yihao Xue, Siddharth Joshi, Eric Gan, Pin-Yu Chen, and Baharan Mirzasoleiman. Which features are learnt by contrastive learning? on the role of simplicity bias in class collapse and feature suppression, 2023.
>
> [4]: Yongqiang Chen, Wei Huang, Kaiwen Zhou, Yatao Bian, Bo Han, and James Cheng. Understanding and improving feature learning for out-of-distribution generalization, 2023
>
> [5]: Chen, Ting, Calvin Luo, and Lala Li. "Intriguing properties of contrastive losses." Advances in Neural Information Processing Systems 34 (2021): 11834-11845.
>
> [6]: Shah, Harshay, et al. "The pitfalls of simplicity bias in neural networks." Advances in Neural Information Processing Systems 33 (2020): 9573-9585.

---

> > ### Comment · Reviewer_ADzt · 2025-11-28
> > **Response to rebuttal**
> >
> > Authors have attended to all of my comments from the previous review. The new version contains the details that were previously missing, along with appropriate justification where needed. Thank you for the effort in the improved version. From my side, the document is ready for publication.

---

### Review · Reviewer_fK7E · 2025-11-08

**Summary Of Contributions:**

The paper under submission proposes a simple parametric instance discrimination (PID) method, called DIET, for self-supervised learning and analyzes the theoretical properties of DIET. Numerical experiments are also conducted to demonstrate the empirical advantage of DIET, compared to available benchmarks. In particular, from what I read, one important claimed advantage of DIET is that it bypasses negative sampling, which may lead to suboptimal results due to the biased comparison within batches. Instead, DIET compares every pair of samples, leading to exhaustive comparisons. Despite the seemingly large number of comparisons needed, DIET has a fast algorithm by using a batch-wise update of the gradient. The theoretical results are somewhat preliminary, but the overall methodology is interesting and may be of interest to certain areas of AI, particularly when data is difficult to collect.

**Additional Comments:**

NA.

**Audience:**

Yes

**Audience Explanation:**

Since self-supervised learning or contrastive learning is quite popular among practicing AI researchers, the proposed method should be of interest to those interested in applying self-supervised learning or understanding its theoretical underpinnings. The method proposed is also related to representation learning, which may be of interest to the general audience of TMLR.

**Broader Impact Concerns:**

None.

**Claims And Evidence:**

Yes

**Claims Explanation:**

Yes. I think the theoretical results are generally sound, which are further corroborated by empirical evidence.

**Requested Changes:**

Major comments:

1. I recommend that more details be added in certain cases. For example, in equation (2), $\mathcal{D} = (\mathbf{x}, y)$. Is $y$ the index label $i$? At least it is not immediately clear to me why $y$ suddenly appears and what it should correspond to.

2. I recommend that more intuition be given in the paper. For example, the advantage of the normalization part, to me, is simply coming from the possibility that $\mathbf{u}_{c}$ and $\mathbf{v}_{c}$ are at different scales, if I am not mistaken. Although they are hidden variables, if the data-generating process is really the additive model given by the first equation in Section 4.1, the benefit of normalization seems hardly surprising.

3. I do not quite follow Theorem 3. Why $(\mathbf{W}, \mathbf{W}_{H})$ is a global minimizer of $\mathcal{L}^{\rm MSE}_{\rm DIET}$? Is this a typo?

4. Since the theoretical results are very preliminary, please consider adding some discussion on the limitations of the theoretical results.

4. The essential idea of the paper is to maximize the prediction discrepancy between different samples in the training data, which makes perfect sense when the dataset contains diverse information. However, what if the sample is somewhat homogeneous? Suppose that in the dataset, the class "dog" contains many pictures of one single dog, with minor differences in its position or angle, but the images are largely similar otherwise. Will the proposed method still work well in this case? It will be helpful to discuss the potential caveat of using the proposed method in applications.

Minor comments:

1. Some of the references are mixed with the texts, making the paper less readable than it should have been. For example, in the first sentence of Section 2, it reads "SimCLR Chen et al. (2020) ...", which should be corrected to "SimCLR (Chen et al. 2020) ...".

2. The first sentence on page 3: "on performance(Grill et al., 2020a..." misses a space.

3. On page 4: "By studying the standard (2) and normalized (2) MSE DIET losses" to "By studying the standard (2) and normalized (3) MSE DIET losses".

4. Page 7: "differ more significantly" to "differ significantly".

---

> ### Author Response · Authors · 2025-11-22
>
> We thank the reviewer for taking the time to provide an insightful review of our paper. Your feedback was invaluable for us to improve the quality of our work. To this nbd, we made the following changes (We detail our response below).
>
> Rebuttal summary
> - We clarified the role and relationship between datum index $n$ and the corresponding label $y_n$
> - We have updated our theoretical results in Sec. 4., extending our result to the cross entropy loss, discussing its limitations, and refining the intuition. We also addressed the case of duplicate or very similar samples
>
>
> We have updated Section 4 of our paper to provide more context for our theoretical results, and have added a result correlating DIET with the InfoNCE objective for unit-normalized embedding functions. Our theory offers a first-of-its-kind framework for analyzing SSL losses from the perspective of better-understood instance-based losses, and it serves as strong motivation for the development of our DIET method.
>
> We also hope to address each of your comments below.
>
> 1. Yes, $y$ refers to the index label, which we now explicitly denote in equation (1), stating that for sample $x_n$ the corresponding label $y_n$ equals the datum index, i.e., $n$. We hope the presentation is clearer in the revised version of Section 4.
> 2. The advantage of the normalization theory lies in the different scales of the noise across content and style features. Informally, in the unnormalized model, the larger noise from the style feature introduces a larger loss, so to minimize the loss the model ends up focusing primarily on the lower noise content feature. In contrast, normalization introduces an additional dependency between the directions, which has the effect of balancing the learning between the directions. The goal of our theory is to rigorously establish that the intuition holds in a minimal setting. We have added this intuition in the revised manuscript.
> 3. The statement in Thm. 3. is correct. There are two paradigms at play here:
> - in contrastive learning, we take the output of the embedding function f, apply the contrastive loss, and then optimize over parameters W.
> - With DIET, we first apply the embedding function f, then apply the classifier head W_H, and finally calculate the loss. Here, we optimize over the parameters W from the embedding function and the classifier head W_H.
> - Summary: The core content of the theorem is that the parameters W learned in each method are equivalent (under some conditions on W_H). The result comes from a detailed analysis of the critical points of each loss function—the full proof is provided in Appendix A.
> 4. We have added more context around our theoretical results in the revised version of Section 4, including extending our results to the CE/InfoNCE that is used in practice. Our theoretical results indicate that the simplified DIET pipeline can match the performance of existing SSL methods, which we verify in our experiments.
> 5. That is an excellent question! The challenge of homogeneous samples is also present in existing SSL methods, such as contrastive learning. For example, if the dataset contains many images of the same dog, then these separate images would be treated as negative pairs and their representations would be pushed apart. There have been methods proposed to tackle this, such as relaxing the notion of positive pairs to include nearest neighbors [1]. Importantly, DIET addresses this issue automatically via calculating the inner product between the embedding of a sample and the rows of the classifier head (instead of taking the inner product between two embeddings). Indeed, this has been shown to imply thoeretically that for duplicated samples, the corresponding row vectors of W will be the same, i.e., the degenerate case of trying to distinguish samples that are basically the same is mitigated by algorithmic design in DIET [2]. Intuitively, one can think of the row vectors of W as (semantic) “cluster centroids” such that semantically similar samples (e.g., pictures of the same dog from slightly different angles) will be close to the same centroid.
>
> Minor comments: Thanks for catching these! We will fix these in the revised version.
>
> [1] Dwibedi, Aytar, Tompson, Sermanet, & Zisserman. “With a Little Help from My Friends: Nearest-Neighbor Contrastive Learning of Visual Representations.” (2021).
>
> [2] Patrik Reizinger, Alice Bizeul, Attila Juhos, Julia E. Vogt, Randall Balestriero, Wieland Brendel, and David Klindt. Cross-Entropy Is All You Need To Invert the Data Generating Process. October 2024.

---

### Decision · Action_Editor_g2BK · 2025-11-28

**Recommendation:** Accept as is

**Additional Comments:**

I kindly ask the authors to double-check that all promised changes to the reviewers have been incorporated into the camera-ready version upon submission, as I will verify this myself again at that point.

**Audience:**

Yes

**Audience Explanation:**

The topic of this paper will interest most of TMLR's audience.

**Claims And Evidence:**

Yes

**Claims Explanation:**

The proposed method called DIET is a simplified SSL method that avoids common complexities such as negative sampling and gradient stopping. The reviewers agreed that the paper is suitable for publication, considering that the authors made some additions and expanded their experiments during rebuttal.

The key claims that make this paper suitable for TMLR are:

a) Simplicity and Theoretical Contribution: the method provides a streamlined SSL pipeline with a theoretical framework linking its loss to established objectives (MSE and InfoNCE).  Authors provided additional results for cross-entropy. They also added some additional baseline methods.

b) Empirical Performance and Memory Efficiency: DIET achieves competitive performance on small-scale datasets compared to state-of-the-art methods, while being more memory-efficient. Authors added detailed comparisons of memory and compute requirements, showing DIET and its scaled variant (s-DIET) outperform baselines in memory usage, which is valuable for resource-constrained settings.